# Sequential inversion of GOCE satellite gravity gradient data and terrestrial gravity data for the lithospheric density structure in the North China Craton

Yu Tian[1,2,3], Yong Wang[2,3]

[1]Ocean College, Minjiang University, 350108 Fuzhou, China

[2]State Key Laboratory of Geodesy and Earth's Dynamics, Institute of Geodesy and Geophysics, Chinese Academy of Sciences, 430077 Wuhan, China

[3]University of Chinese Academy of Sciences, 100049 Beijing, China

*Correspondence to*: Yu Tian (tybgys455429145@163.com)

**Abstract.** The North China Craton (NCC) is one of the oldest cratons in the world. Currently, the destruction mechanism and geodynamics of the NCC remain controversial. All of the proposed views regarding the issues involve studying the internal density structure of the NCC lithosphere. Gravity field data are among the most important data in regard to investigating the lithospheric density structure, and gravity gradient data and gravity data each possess their own advantages.

Given the different observational plane heights between the on-orbit GOCE satellite gravity gradient and terrestrial gravity and the effects of the initial density model on the inversion results, sequential inversion of the gravity gradient and gravity are divided into two integrated processes. By using the preconditioned conjugate gradient (PCG) inversion algorithm, the density data are calculated using the preprocessed corrected gravity anomaly data. Then, the newly obtained high-resolution density data are used as the initial density model, which can serve as constraints for the subsequent gravity gradient inversion. Several essential corrections are applied to the four gravity gradient tensors ( $T_{xx}$ , $T_{xz}$ , $T_{yy}$ , $T_{zz}$ ) of the GOCE satellite, after which the corrected gravity gradient anomalies ( $T'_{xx}$ , $T'_{xz}$ , $T'_{yy}$ , $T'_{zz}$ ) are used as observations. The lithospheric density distribution result within the depth range of 0-180 km in the NCC is obtained.

This study clearly illustrates that GOCE data are helpful in understanding the geological settings and tectonic structures in the NCC with regional scale. The inversion results show that in the crust, the eastern NCC is affected by lithospheric thinning with obvious local features. In the mantle, the presented obvious negative density areas are mainly affected by the high heat flux environment. In the eastern NCC, the density anomaly in the Bohai Bay area is mostly attributed to the extension of the Tancheng-Lujiang major fault at the eastern boundary. In the western NCC, the crustal density anomaly distribution of the Qilian block is consistent with the northwest-southeast strike of the surface fault belt, whereas such an anomaly distribution experiences a clockwise rotation to a nearly north-south direction upon entering the mantle.

## 1 Introduction

The North China Craton (NCC, Fig. 1) is an outstanding example of a craton that has undergone both reconstruction and destruction. Studies on the NCC have enabled us to understand the formation, evolution, stabilization and destruction of the ancient continent. Two dominant NCC destruction mechanisms have been proposed, namely, delamination (Gao et al., 2009) and thermal erosion (Zhu et al., 2012). However, both of these mechanisms involve internal tectonic deformation and substance distribution in the lithosphere of the NCC, which highlights the importance of obtaining the high-resolution density structure of the lithosphere. Gravity field data play an important role in determining and interpreting the lithospheric density structure and state of motion. For studies on the density structure of the NCC, Fang (1996) managed to invert the density distribution of the lithosphere in the North China area, with the constrained least square method using the Bouguer anomaly. Additionally, using the Bouguer gravity anomaly, Wang et al. (2014) obtained the three-dimensional density structure of the NCC lithosphere through the algebraic reconstruction inversion method, which was constrained by seismic travel time. Based on gravity, geoidal surface and topography data, Xu et al. (2016) calculated the crustal density and depth of the interface between the lithosphere and asthenosphere using the rapid integrated inversion method. Using the preprocessed data of the GOCE satellite gravity gradient anomaly, Tian and Wang (2018) constructed a three-dimensional density structure within the depth range of 0-120 km in the NCC lithosphere, during which the density variation induced by the temperature differences was incorporated. Previous studies on the NCC only adopted gravity or gravity gradient measurements, both of which have their own unique advantages as first-order and second-order derivatives of the gravity potential, respectively. In the frequency domain, the high frequencies of the deep structures are strongly attenuated due to the distance, which is masked by the signals of closer masses. The gravity data are mainly used to provide mid-low frequency information about the deep structure. The amplitude of the gravity gradient data declines rapidly with increasing depth in the field source, which demonstrates that the gravity gradient anomaly is applicable to the high frequency signal information of the shallow structure, characterized by short wavelengths. By using both the gravity and gravity gradient data as the observation quantities, both the low frequency signal information and high frequency signal information of the gravity anomaly data can expand the frequency of the gravity field data. Sequential inversions of the gravity and gravity gradient data are able to achieve mutual supplementation, which is favorable to enhance the reliability of the inversion result and obtain more reasonable analyses of the inversion solutions.

Currently, most studies are based on the joint inversion of gravity and gravity gradient data instead of sequential inversions. Zhdanov et al. (2004) introduced the concept of the curvature of gravity to carry out a joint inversion of gravity and gravity gradient data and applied the proposed method to the existing models. Wu et al. (2013) inverted gravity and gravity gradients by transforming the formulas of the gravity and gravity gradients with the target body treated as a mass point. Given the varied decline rates of the kernel functions of the gravity and gravity gradient data, Capriotti and Li (2014) balanced the two decline rates using the density matrix to conduct a joint inversion of the gravity and gravity gradient data, after which the validity of the proposed method was confirmed with the published SEG model. Qin et al. (2016) developed an integrated

focusing inversion algorithm for gravity and gravity gradient data and applied this method to gravity and gravity gradient data from aerial surveys in the Vinton salt dome area. Li et al. (2017) inverted the lithospheric three-dimensional density structure of the Qinghai-Tibet Plateau and its adjacent area within a depth range of 0-120 km using the gravity gradient data measured by GOCE L2 together with the vertical gravity calculated by EGM2008. For small research areas, studies involving the gravity and gravity gradient joint inversion focused on applications of existing models or aerial gravity and

gravity gradient data at the same height. With respect to previous studies covering a large area, researchers often adopted gravity or gravity gradient data that were directly computed by the gravity field model. Although the gravity field model is able to rapidly calculate the gravity and gravity gradient data for large research areas worldwide, the resulting data are based on the spherical harmonic coefficients rather than actual measurements. In addition, directly using observations rather than the gravity field models is advantageous because it avoids the global average effect during gravity field modeling (Pavlis et

al., 2012). Thus, compared with the calculated data based on the gravity field model, the high precision gravity and gravity gradient data obtained from measurements with high resolution possess the same importance (Liu et al., 2003; Li et al., 2011).

We selected GOCE satellite gravity gradient data along the orbit and gravity data of the terrestrial survey as the

80 measurements. The GOCE satellite data at the mean orbital height only reflect large structures of the earth. To highlight the high frequency information of shallow abnormal bodies and the detailed information of structural features, it is necessary to move the GOCE satellite data from the mean orbital height downward to the near surface in the NCC area. However, given the feasibility of the downward continuation result of the gravity gradient data from the GOCE satellite, the height of the observation plane after the downward continuation should be located outside the topographic mass unit (Sebera et al., 2014;

Li et al., 2017; Tian and Wang, 2018), while each observation location for gravity data acquired through the survey is always located on the topographic surface. Therefore, an inconsistency problem in the observation plane height is anticipated between the processed GOCE satellite gravity gradient data and the gravity data obtained by the terrestrial survey. Furthermore, although gravity and gravity gradient data are favored by higher resolutions and sensitivity to density, the inversion method is characterized by a strong nonuniqueness of solutions. To constrain the gravity and gravity gradient

inversion, seismic data are often input into the inversion after a transformation based on empirical formulas to suppress the nonuniqueness of solutions. In this regard, the effects of the initial model on the inversion results should be considered. For the two aforementioned aspects, the sequential inversions of gravity and gravity gradient are divided into two processes. First, the density data converted from the seismic wave velocity are used as the initial density model. The preprocessed corrected gravity anomaly data of the NCC terrestrial survey are collected as the observation quantity, and then, the density

anomaly within a depth range of 0-180 km is calculated using the preconditioned conjugate gradient (PCG) algorithm in the first process. The obtained inversion results are used as the new initial density model and serve as constraints. The four high accuracy (Rummel et al., 2011; Yi et al., 2013) GOCE satellite gravity gradient anomaly tensors ($T_{xx}$, $T_{xz}$, $T_{yy}$, $T_{zz}$) in the

NCC are collected as the original observation data for downward continuation, topographic effect correction, underground interface undulation effect correction and long wavelength correction. The preprocessed corrected gravity gradient anomaly data ( $T'_{xx}$, $T'_{xz}$, $T'_{yy}$, $T'_{zz}$ ) are used as the new observation quantity, and the resulting density anomaly distribution within the depth range of 0-180 km in this area is obtained using the same PCG algorithm in the second process.

By considering the features of the gravity anomaly and gravity gradient anomaly data, we used the corrected gravity data as the initial measurements rather than gravity gradient data, which can be used first to determine the major and deep structures of the lithosphere based on the inversion results. Then, the inversion results of the corrected gravity gradient data are applied to identify the fine structures of the lithosphere. This presented method can be exempt from the limitation imposed by the observational plane height, which is able to sufficiently exploit and utilize the available actual gravity and gravity gradient measurements. The corrected initial model with the effect of the gravity inversion results can provide more reliable and effective initial density models for the following gravity gradient inversion. Compared with the inversion results based on either the gravity or gravity gradients in the NCC, the integrated inversion results offer both regional detailed information and a density structure model penetrating deeper underground, which is favorable for discussions and analyses of data regarding the crust and mantle in the same area; consequently, the phenomenon and origin of the NCC destruction mechanism can be investigated.

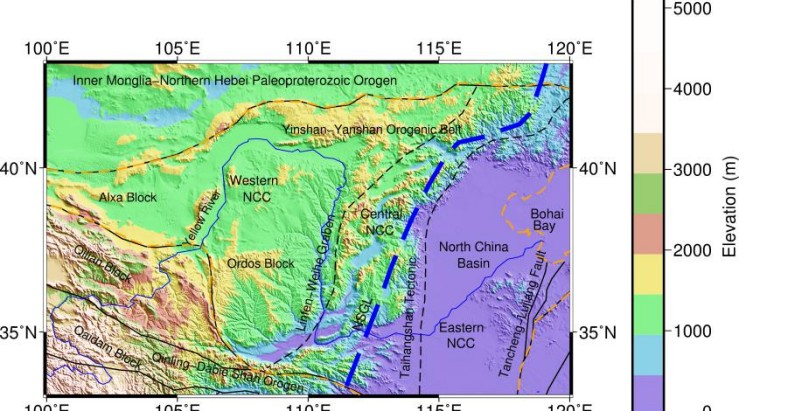

**Figure 1** Topography and main tectonics of the study area. NCC - North China Craton.

**2 Methods**

## 2.1 Kernel function calculation

Under the Cartesian coordinate system, according to the geological body with the residual density of $\rho$ and the volume of V, the first and second derivatives of the gravitational potential at any point $P_1$ $(x_0, y_0, z_0)$ in the outer space can be expressed as follows (Nagy et al. 2000):

$$T_z(P_1) = G\rho\iiint \frac{(z-z_0)}{\left[(x-x_0)^2+(y-y_0)^2+(z-z_0)^2\right]^{3/2}}dxdydz$$

$$T_{xx}(P_1) = G\rho\iiint \frac{2(x-x_0)^2-(y-y_0)^2-(z-z_0)^2}{\left[(x-x_0)^2+(y-y_0)^2+(z-z_0)^2\right]^{5/2}}dxdydz$$

$$T_{xz}(P_1) = G\rho\iiint \frac{3(x-x_0)(z-z_0)}{\left[(x-x_0)^2+(y-y_0)^2+(z-z_0)^2\right]^{5/2}}dxdydz$$

$$T_{yy}(P_1) = G\rho\iiint \frac{2(y-y_0)^2-(x-x_0)^2-(z-z_0)^2}{\left[(x-x_0)^2+(y-y_0)^2+(z-z_0)^2\right]^{5/2}}dxdydz \quad , \tag{1}$$

$$T_{zz}(P_1) = G\rho\iiint \frac{2(z-z_0)^2-(x-x_0)^2-(y-y_0)^2}{\left[(x-x_0)^2+(y-y_0)^2+(z-z_0)^2\right]^{5/2}}dxdydz$$

where G is the Newtonian gravitational constant. By dividing the underground space of the detection area into k prisms with the same size, each cube is given a particular residual density, and the analytical expression of Eq. (1) can be expressed as follows:

$$\boldsymbol{T}_z(P_1) = \sum_{i=1}^{k} G\rho\left\{(x-x_0)ln[r+(y-y_0)]+(y-y_0)ln[r+(x-x_0)-(z-z_0)arctan\frac{(x-x_0)(y-y_0)}{(z-z_0)r}]\right\}\Big|_{x_i}^{x_{i+1}}\Big|_{y_i}^{y_{i+1}}\Big|_{z_i}^{z_{i+1}} = \boldsymbol{G}_z^{(P_1)}\boldsymbol{\rho}$$

$$T_{xx}(P_1) = \sum_{i=1}^{k} G\rho_i[-tan^{-1}\frac{(y-y_0)(z-z_0)}{(x-x_0)r}]\Big|_{x_i}^{x_{i+1}}\Big|_{y_i}^{y_{i+1}}\Big|_{z_i}^{z_{i+1}} = \boldsymbol{G}_{xx}^{(P_1)}\boldsymbol{\rho}$$

$$T_{xz}(P_1) = \sum_{i=1}^{k} G\rho_i\, ln(y-y_0+r)\Big|_{x_i}^{x_{i+1}}\Big|_{y_i}^{y_{i+1}}\Big|_{z_i}^{z_{i+1}} = \boldsymbol{G}_{xz}^{(P_1)}\boldsymbol{\rho}$$

$$T_{yy}(P_1) = \sum_{i=1}^{k} G\rho_i[-tan^{-1}\frac{(z-z_0)(x-x_0)}{(y-y_0)r}]\Big|_{x_i}^{x_{i+1}}\Big|_{y_i}^{y_{i+1}}\Big|_{z_i}^{z_{i+1}} = \boldsymbol{G}_{yy}^{(P_1)}\boldsymbol{\rho} \quad , \tag{2}$$

$$T_{zz}(P_1) = \sum_{i=1}^{k} G\rho_i[-tan^{-1}\frac{(x-x_0)(y-y_0)}{(z-z_0)r}]\Big|_{x_i}^{x_{i+1}}\Big|_{y_i}^{y_{i+1}}\Big|_{z_i}^{z_{i+1}} = \boldsymbol{G}_{zz}^{(P_1)}\boldsymbol{\rho}$$

where $\boldsymbol{G}_z^{(P_1)}, \boldsymbol{G}_{xx}^{(P_1)}, \boldsymbol{G}_{xz}^{(P_1)}, \boldsymbol{G}_{yy}^{(P_1)}, \boldsymbol{G}_{zz}^{(P_1)}$ represents the corresponding kernel function matrix at point $P_1$ and $\boldsymbol{\rho} = (\rho_1, \rho_2, \ldots, \rho_k)^T$ represents the density of each cube. The gravity gradient anomalies generated by all n points on the observation surface meet the linear relationship from the k discrete prisms in the underground space. The relationship can be expressed as the following equation:

$$T_{ij} = G_{ij}\rho \ ,$$
$$T_{ij} = [T_{ij}^{(P_1)}, T_{ij}^{(P_2)}, \cdots\cdots, T_{ij}^{(Pn)}]^{\mathrm{T}} \ ,$$
$$G_{ij} = [G_{ij}^{(P_1)}, G_{ij}^{(P_2)}, \cdots\cdots, G_{ij}^{(P_n)}]^{\mathrm{T}} \ , \ ij = z,\ xx,\ xz,\ yy,\ zz \tag{3}$$

where $T_{ij}$ represents the observation and $G_{ij}$ represents the corresponding kernel function matrix for the observation surface.

## 2.2 Inversion method

Essentially, the inversion of the gravity and gravity gradient tensors is a process of solving a system of linear equations. The quantity of the unknown m greatly exceeds the acquired data vector, and the solutions of the equations are nonunique. Moreover, inversion is an ill-conditioned problem, and appropriate constraints on the objective function are required to narrow the range of solutions. Therefore, in the linear inversion theory, the objective function mostly consists of the data fitting function and the model objective function (Constable et al., 1987). Under such circumstances, solving the inversion

problem is equivalent to finding a model vector $m$ that can minimize the objective function while satisfying the data fitting function condition.

The objective function can be expressed as follows:
$$\text{minimize:} \phi = \phi_d + \mu\phi_m \ , \tag{4}$$

where $\mu$ represents the regularization parameter, which represents the weight factor that balances the data fitting function $\phi_d$ and the model objective function $\phi_m$.

The data fitting function is defined as follows:
$$\phi_d = \sum_{i=1}^{N}\left(\frac{\Delta d_i - G_i\Delta m}{\sigma_i}\right)^2 = \left\|W_d(\Delta d - G\Delta m)\right\|^2 \ , \tag{5}$$

$$W_d = diag\left\{1/\sigma_1, 1/\sigma_2, ..., 1/\sigma_N\right\}, \tag{6}$$

where $\Delta m = m - m_0$ is the correction between the model parameter vector $m$ and the initial model $m_0$; $G$ is the kernel function, namely, the linear projection operator from the model element to the observation (for gravity data, $G = G_z$, while for gravity gradient components, $G = [G_{xx}, G_{xz}, G_{yy}, G_{zz}]^{\mathrm{T}}$); $\Delta d$ is the correction of the corresponding measurement; and $W_d$ is a diagonal matrix, with $\sigma_i$ representing the standard deviation of the $i$-th data. The objective function of the model

is constructed according to the minimization model function.

The Lagrangian multiplier is used as the regularization parameter in the PCG inversion algorithm. In the process of solving a large-scale matrix, the calculation of the Lagrangian multiplier is complex; therefore, an empirical value is often adopted as the relative optimal value for the regularization parameter. Since the regularization parameter serves to balance the data fitting function and the model fitting function, an excessively large value will result in substantial differences between the inverse results response and the observation, while an overwhelmingly small value leads to ineffectiveness of the model fitting function.

Given these problems, the L-curve method was developed for the selection of regularization parameters in the solution for ill-posed problems (Hansen, 1992). The L-curve is a criterion that is based on a comparison between the actual data fitting function and the model objective function, which is applicable to solving large-scale problems. The value corresponding to the inflection point of the L-curve is assigned to the regularization parameter. The effectiveness of this method has been validated in previous studies (Tian et al., 2018; 2019). The curvature of the L-curve can be expressed as follows (Hansen, 1992):

$$k = \frac{\hat{\rho}'\hat{\eta}'' - \hat{\rho}''\hat{\eta}'}{\left[\left(\hat{\rho}'\right)^2 + \left(\hat{\eta}'\right)^2\right]^{3/2}}, \tag{7}$$

where $\hat{\rho} = \log(\phi_d)$ and $\hat{\eta} = \log(\phi_m)$; and the superscripts ' and " represent the first-order and second-order derivatives of the function, respectively. Accordingly, during the inversion process, the algorithm seeks the maximum curvature of the L-curve based on the function constructed by actual data.

To constrain the spatial structure of the model and achieve a continuous variation in the inversion image along the three axis directions, a roughness matrix is introduced into the model objective function (Constable et al., 1987), with reference to the minimization model function. The three-dimensional model vector $R$ is the quadratic sum of the first-order partial difference of the model vector $m$ along the x, y and z directions.

$$R = \left\|\partial_x \Delta m\right\|^2 + \left\|\partial_y \Delta m\right\|^2 + \left\|\partial_z \Delta m\right\|^2 = \int\left(\frac{\partial \Delta m}{\partial x}\right)^2 dv + \int\left(\frac{\partial \Delta m}{\partial y}\right)^2 dv + \int\left(\frac{\partial \Delta m}{\partial z}\right)^2 dv, \tag{8}$$

by meshing the model and replacing the partial differential form with the finite difference form, the above equation is converted into the following matrix:

$$R = \Delta m (R_x^{\mathrm{T}} R_x + R_y^{\mathrm{T}} R_y + R_z^{\mathrm{T}} R_z) \Delta m, \tag{9}$$

where $R_x$, $R_y$ and $R_z$ are the roughness matrices of the model along the x, y and z directions, respectively.

Because the gravity data and gravity gradient data have no fixed depth resolution, the kernel function declines rapidly with increasing depth, and the inversion results are limited near the surface, which results in difficulties in capturing the true position of the anomaly. By introducing the depth weighting function into the model objective function, the kernel function is optimized to reflect the true weighted value of the anomaly element at each depth. The depth weighting function designed by Li and Oldenburg (1996), especially for the inversion of gravity data and gravity gradient data, is adopted as follows:

$$W(z) = \frac{1}{(Z + Z_0)^{\beta/2}},$$
(10)

where $Z$ is the burial depth of the center of the grid cell and $Z_0$ and $\beta$ are constants. For gravity data, these values are used to counteract the decline in the kernel function $G$, with $\beta$ often set to 2 and the function written as $(Z+Z_0)^{-1}$, while for the gravity gradient data, these values are used to compensate for the decline in the kernel function $G$, with $\beta$ often set to 3 and the function written as $(Z+Z_0)^{-3/2}$.

In accordance with the minimization model function, the model objective function can be constructed as shown below, in reference to the roughness matrix and depth weighting function:

$$\phi_m(\boldsymbol{m}) = \alpha_s \int_V (\partial W(z)\Delta\boldsymbol{m})^2 dv + \alpha_x \int_V \left(\frac{\partial W(z)\Delta\boldsymbol{m}}{\partial x}\right)^2 dv + \alpha_y \int_V \left(\frac{\partial W(z)\Delta\boldsymbol{m}}{\partial y}\right)^2 dv + \alpha_z \int_V \left(\frac{\partial W(z)\Delta\boldsymbol{m}}{\partial z}\right)^2 dv,$$
(11)

The model objective function can be converted into a matrix form by replacing the differential form with the finite difference method:

$$\phi_m(\boldsymbol{m}) = \Delta\boldsymbol{m}^T (W_S^T W_S + W_x^T W_x + W_y^T W_y + W_z^T W_z)\Delta\boldsymbol{m} = \Delta\boldsymbol{m}^T W_i^T W_i \Delta\boldsymbol{m},$$
(12)

$$W_i = \alpha_i R_i D, \quad i = s, x, y, z,$$
(13)

where $\alpha_i$ is the weight coefficient for each term in the objective function; $R_i$ is the difference operator for each component; and $D$ is the discretized depth weighting function matrix. Substituting the model objective function into the objective

function yields the following expression:

$$\phi = (\Delta\boldsymbol{d} - G\Delta\boldsymbol{m})^T (\Delta\boldsymbol{d} - G\Delta\boldsymbol{m}) + \mu \Delta\boldsymbol{m}^T W_i^T W_i \Delta\boldsymbol{m},$$
(14)

Eq. (11) can be rearranged into the following matrix form:

$$\begin{bmatrix} G \\ \sqrt{k}W_i \end{bmatrix} \Delta\boldsymbol{m} = \begin{bmatrix} \Delta\boldsymbol{d} \\ 0 \end{bmatrix}$$
(15)

by replacing the condition matrix of the objective function with $A = \begin{bmatrix} G, \sqrt{k}W_i \end{bmatrix}^T$ and defining $\boldsymbol{b} = \begin{bmatrix} \Delta\boldsymbol{d}, 0 \end{bmatrix}^T$, Eq. (15) can

be simplified as follows:

$$A\Delta m = b,$$ (16)

For the gravity data, we take $T_z$ as the observation, where $A = \begin{bmatrix} G_z, \sqrt{\mu}W_i \end{bmatrix}^T$ and $b = \begin{bmatrix} T_z, 0 \end{bmatrix}^T$. For the gravity gradient data, we simultaneously selected four processed components $T'_{xx}$, $T'_{xz}$, $T'_{yy}$ and $T'_{zz}$ as the observations, which implies the Jacobian matrix, where $A = \begin{bmatrix} G_{xx}, G_{xz}, G_{yy}, G_{zz}, \sqrt{\mu}W_i \end{bmatrix}^T$ and $b = \begin{bmatrix} T'_{xx}, T'_{xz}, T'_{yy}, T'_{zz}, 0 \end{bmatrix}^T$. The contribution of each component and its differences from the joint inversion can be found in our previous studies (Tian et al., 2019).

Additionally, $A^T A = (G^T G + k^{-1} W_i^{\ T} W_i)$. Because the condition number of the Jacobian matrix is normally very large, to increase the convergence speed and the solution stability, Eq. (16) is rewritten as follows:

$$S A^T A \Delta m = S A^T b,$$ (17)

where $S$ is the preconditioned factor, which is usually approximated as $(A^T A)^{-1}$. Thus, the eigenvalues of Eq. (17) will be concentrated along the diagonal, and the condition number will be improved so that the iteration efficiency is improved (Pilkington, 2009).

The PCG method is an algorithm for solving linear or nonlinear equations through an iterative calculation process and is mainly used to solve the quadratic equation $\phi = \frac{1}{2} m^T A m - m^T b$, with a solution of $m = A^{-1} b$, which is obviously equivalent to Eq. (16). The PCG method (Pilkington, 1997; 2009) is adopted for the inversion calculation.

**2.3 Effects of the initial density model**

Two models (Fig. 2 and Fig. 3) are designed to test the effects of the initial model on the inversion results. Model one (Fig. 2) consists of two independent cubes equipped with density values. The densities are different at each layer to study the role of the initial values in the inversion. Model two (Fig. 3) is composed of an upright cuboid and a trapezoid structure of multiple anomalies. This model test can be used to illustrate the initial model effects on the shape and magnitude of the inversion results. Since this study aims to capture the initial model effects upon the final inversion results, carrying out the integrated inversion for the four components ($T_{xx}, T_{xz}, T_{yy}, T_{zz}$) is required.

As shown in Fig. 2 and Fig. 3, the inversion has been carried out for identical density anomaly bodies (Fig. 2a and Fig. 3a), during which the gravity gradient measurement, kernel function and algorithm parameters are all the same, except that the

varied values are assigned to the initial model. In Model 1, the anomalies are set to 0 g/cm$^3$ (Fig. 2b); 0.4 g/cm$^3$ (Fig. 2c); 0.1 g/cm$^3$ (1$^{st}$ layer), 0.2 g/cm$^3$ (2$^{nd}$ and 3$^{rd}$ layers) and 0.3 g/cm$^3$ (4$^{th}$ layer) in Fig. 2d; and 0.2 g/cm$^3$ (1$^{st}$ layer), 0.4 g/cm$^3$ (2$^{nd}$ and 3$^{rd}$ layers) and 0.6 g/cm$^3$ (4$^{th}$ layer) in Fig. 2e. Then, in Model 2, the initial densities are set to 0 g/cm$^3$ (Fig. 3b), 0.2 g/cm$^3$ (Fig. 3c), 0.5 g/cm$^3$ (Fig. 3d) and -0.2 g/cm$^3$ (Fig. 3e).

In terms of the inversion of Model 1, the iterations are 10, 7, 6 and 3, and the inversion results are illustrated in Fig. 2b-e. The detailed results of every inversion test are summarized in Table 1.

**Table 1** Conditions and results for initial density model one.

| | True density at each layer (g/cm$^3$) (1$^{st}$ to 4$^{th}$ layer) | Initial density at each layer (g/cm$^3$) (1$^{st}$ to 4$^{th}$ layer) | Iterations (k) | RMS (E) | Maximum density at each layer (g/cm$^3$) (1$^{st}$ to 4$^{th}$ layer) |
|---|---|---|---|---|---|
| Test 1 | (0.4, 0.7, 0.7, 1.0) | (0, 0, 0, 0) | 10 | $7.35\times10^{-11}$ | (0.34, 0.42, 0.53, 0.55) |
| Test 2 | (0.4, 0.7, 0.7, 1.0) | (0.4, 0.4, 0.4, 0.4) | 7 | $6.94\times10^{-12}$ | (0.51, 0.55, 0.66, 0.58) |
| Test 3 | (0.4, 0.7, 0.7, 1.0) | (0.1, 0.2, 0.2, 0.3) | 6 | $3.50\times10^{-12}$ | (0.36, 0.64, 0.67, 0.64) |
| Test 4 | (0.4, 0.7, 0.7, 1.0) | (0.2, 0.4, 0.4, 0.7) | 3 | $2.31\times10^{-13}$ | (0.38, 0.72, 0.76, 0.95) |

| | Minimum density at each layer (g/cm$^3$) (1$^{st}$ to 4$^{th}$ layer) | Average density at each layer (g/cm$^3$) (1$^{st}$ to 4$^{th}$ layer) |
|---|---|---|
| Test 1 | (0.23, 0.27, 0.34, 0.29) | (0.31, 0.34, 0.41, 0.35) |
| Test 2 | (0.42, 0.43, 0.51, 0.48) | (0.47, 0.50, 0.58, 0.55) |
| Test 3 | (0.32, 0.46, 0.33, 0.55) | (0.34, 0.55, 0.59, 0.58) |
| Test 4 | (0.33, 0.52, 0.57, 0.83) | (0.35, 0.59, 0.65, 0.91) |

Fig. 2b-e indicates that as the initial model becomes more precise, the inversion results become more reliable, with the values of the anomalies becoming more similar to the true model. With no defined initial model (Fig. 2b), it is difficult to capture the layered structure of the anomaly, and the resulting average density is only 0.38 g/cm$^3$, which is far different from the true model; however, in this case, the inversion results capture the position of the anomaly, its depth distribution, and its shape as two upright cuboids. The whole inversion process is relatively time consuming and finally reaches the convergence

criterion after 10 iterations. We also compare the cases based on Model 1 with one another. Compared with Fig. 2d, although Fig. 2c presents smaller differences in the density anomaly values than the true model, the layered structure of the anomaly is difficult to capture from Fig. 2c. In Fig. 2d, it is feasible to conclude that certain differences exist between the density anomaly distributions of the 1$^{st}$ and 2$^{nd}$ layers, which is more similar to the true model. This finding suggests that the inversion results are not only related to the density value of the initial model but also to the distribution structure of the initial

density. Therefore, a more precise initial model with a layered density value distribution favors the generation of more reliable inversion results. Notably, the optimal inversion results are attributed to the case of Fig. 2e, which produces an inverted anomaly body shape that is almost identical to the true model. Apparently, the density distribution of the anomalous

body is laminated into 3 layers, with the 1st layer having a density anomaly of 0.3-0.4 g/cm³; the 2nd and 3rd layers generally having density anomalies of 0.6-0.7 g/cm³; and the 4th layer having a density anomaly of 0.8-1.0 g/cm³.


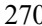

**Figure 2** Initial density model experiments. (**a**) True model. Inversion results for (**b**) Test 1, (**c**) Test 2, (**d**) Test 3, and (**e**) Test 4.

For Model 2, the inversions for each case are calculated after 8, 7, 5 and 13 iterations. The inversion results are illustrated in Fig. 3b-e, and the details of the results of each case are given in Table 2. With the aid of a more precise initial model and

more data, Fig. 3b-d demonstrates that it would be easier to determine the shape and dimensions of the density anomaly body from the inversion results. In the inversion results of Fig. 3b, it is feasible to directly capture the upright cuboid of the density anomaly on the right, and for the complicated density anomaly body on the left, the interpretation based on the inversion results is limited to this anomaly body having an irregular shape and its size growing with increasing depth. In Fig. 3c, the anomaly body on the left reveals a stair-like shape, yet it is surrounded by multiple small anomaly bodies, and the

density values of the upright cuboid on the right are closer to those of the true model. Next, the optimal inversion results are found in Fig. 3d, with anomalous body shapes that are completely consistent with those of the true model, and it is easy to discriminate the shape of the anomalous body. Meanwhile, it should also be noted that the density anomaly value of the upright cuboid on the right is higher than that of the stair-like body on the left. Finally, the inversion results in Fig. 3e make

it difficult to determine the shapes and sizes of both anomalous bodies. In addition, according to the inversion results
summary (Table 2), the average density anomaly is only 0.3276 g/cm$^3$ in this case, which is very different from that of the
true model.

Table 2 Conditions and results for initial density model two.

| | True density (Trapezoid, Cuboid) (g/cm$^3$) | Initial density (Trapezoid, Cuboid) (g/cm$^3$) | Iterations (k) | RMS (E) | Maximum density (g/cm$^3$) | Minimum density (g/cm$^3$) | Average density (g/cm$^3$) |
|---|---|---|---|---|---|---|---|
| Test 5 | (0.8, 1.0) | (0, 0) | 8 | $4.37 \times 10^{-12}$ | 0.6887 | 0.1328 | 0.4225 |
| Test 6 | (0.8, 1.0) | (0.2, 0.2) | 7 | $5.63 \times 10^{-13}$ | 0.7121 | 0.1856 | 0.4886 |
| Test 7 | (0.8, 1.0) | (0.5, 0.5) | 5 | $1.06 \times 10^{-13}$ | 0.7556 | 0.4762 | 0.6256 |
| Test 8 | (0.8, 1.0) | (-0.2, -0.2) | 13 | $8.49 \times 10^{-11}$ | 0.6399 | 0.0399 | 0.3276 |

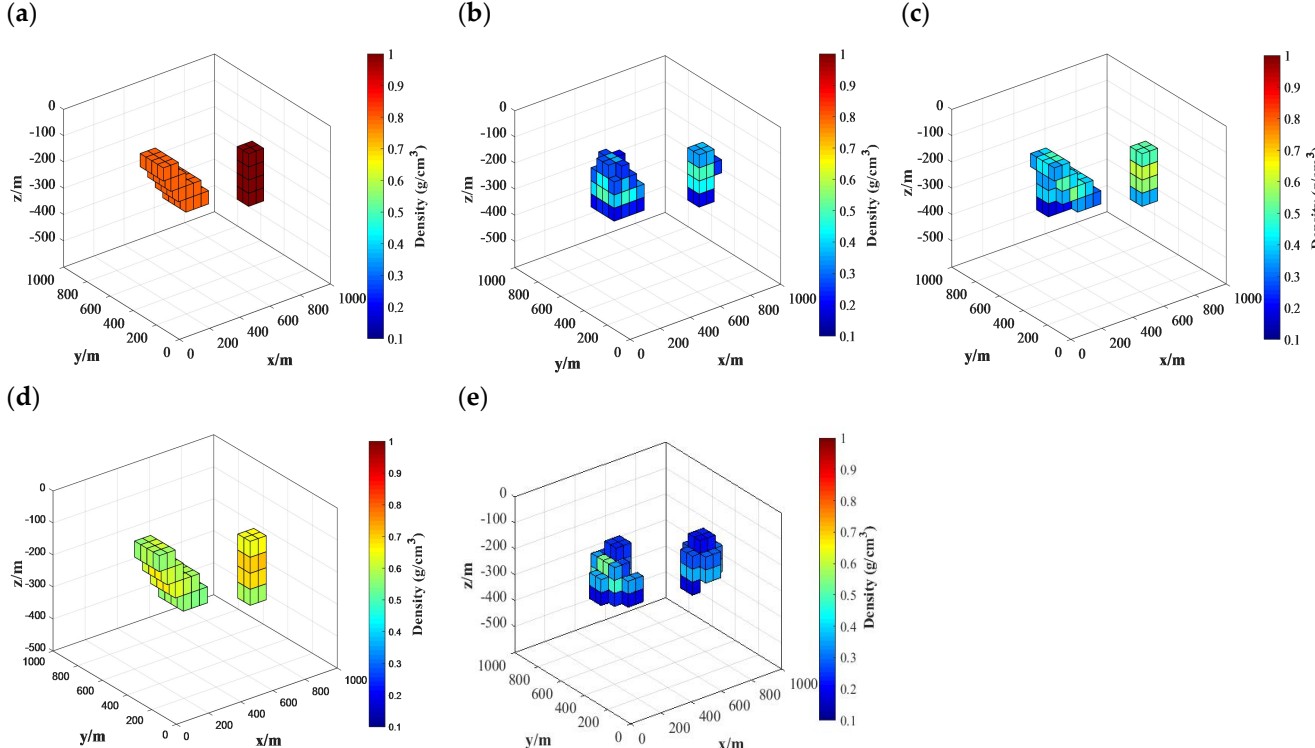

**Figure 3** Initial density model experiment two. (**a**) True model. Inversion results for (**b**) Test 1, (**c**) Test 2, (**d**) Test 3, and (**e**) Test 4.

The model tests show that the inversion results are not only related to the observation but also affected by the physical and
geometric parameters of the initial density model. Therefore, the density values for the initial model, as well as the general
area where the density anomaly body is located, should be more accurately set for the initial density model.

## 2.4 The process of the sequential inversion

The sequential inversion is realized in two steps, namely, gravity inversion and gravity gradient inversion. The detailed workflow of the whole inversion process is illustrated in Fig. 4. Given the constraint imposed on the inversion results by the initial density model, the high-precision results of the gravity inversion are used as the new initial density model for the subsequent steps, and the ultimate inversion results are somewhat constrained during the gravity gradient inversion calculation. Specifically, the density data converted from the seismic wave velocity are first used as the initial density model, and the preprocessed corrected gravity anomaly data are collected to serve as the observation quantity. Then, the kernel function corresponding to the gravity data is calculated, and the density anomaly data within the depth range of 0-180 km in the study area are calculated through the PCG algorithm. Then, the obtained gravity inversion results are used as the new initial density model, and the gravity gradient components in this area that are measured by the GOCE satellite are processed, and then, the four corrected gravity gradient anomaly components ( $T'_{xx}$, $T'_{xz}$, $T'_{yy}$, $T'_{zz}$ ) are simultaneously used as the new observations. The kernel functions corresponding to each gravity gradient component are calculated, and then, the ultimate density anomaly distribution within a depth range of 0-180 km in the study area is obtained through the same PCG algorithm.

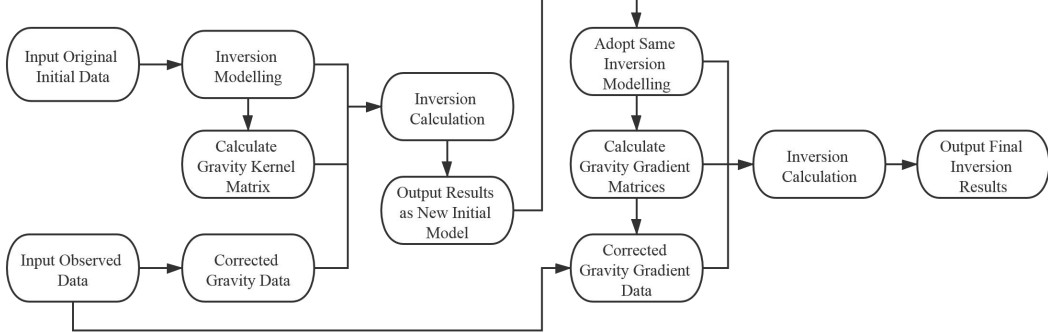

**Figure 4** The flowchart for the whole PCG inversion algorithm procedure.

## 3 Data processing

### 3.1 Inversion based on the corrected gravity anomaly data

The corrected gravity anomaly data (Wang et al., 2012) with depths of 0-180 km in the NCC have been directly collected. The corrected gravity anomaly data are based on the 5′ × 5′ free air gravity anomaly data set, which comes from the China Seismic Network. The corrected gravity data are first subjected to topographic correction and Bouguer correction, then to the underground interface undulation correction, which consists of sedimentary undulation correction and Moho undulation

correction and finally to the long wavelength correction, which corresponds to the gravity gradient effects of the 2nd-33rd order spherical harmonic coefficients. The gravity anomalies induced by the density heterogeneity of the crust and upper mantle are used as the observations for inversion. The gravity anomaly (Fig. 5a) after multiple corrections is shown in Fig. 5a. The figure shows that the gravity anomaly of the eastern NCC shows alternating distributions of positive and negative anomalies with local features. In the central NCC, most regions along the Taihang Orogenic belt and the Yinshan-Yanshan Orogenic belt are found to have negative gravity anomalies. For the western NCC, the whole Ordos block presents a prominent positive gravity anomaly distribution.

Although the gravity data are favored by their higher resolutions, the gravity inversion suffers from a strong nonuniqueness of solutions. It is effective to introduce seismic data into the gravity inversion to constrain the gravity inversion process and the nonuniqueness of the solution. Based on the 0.5°×0.5° resolution of the P-wave velocity structure obtained by seismic tomography (Tian et al., 2009), we constructed a three-dimensional initial density model for the NCC lithosphere using the empirical velocity to density conversion formula of the North China area (Eq. 16). The constructed models are divided into eight layers along the depth direction, with base depths of 10 km, 25 km, 42 km, 60 km, 80 km, 100 km, 140 km and 180 km. Along the horizontal direction, the model is meshed into grids of 0.5° × 0.5°. As shown in Fig. 5b, after 17 iterations, the curve of the defined RMS misfit tends to be horizontal, the variation is slight, and the calculated RMS misfit of $8.2×10^{-3}$ was obtained, at which point the inverse iterative calculation was complete.

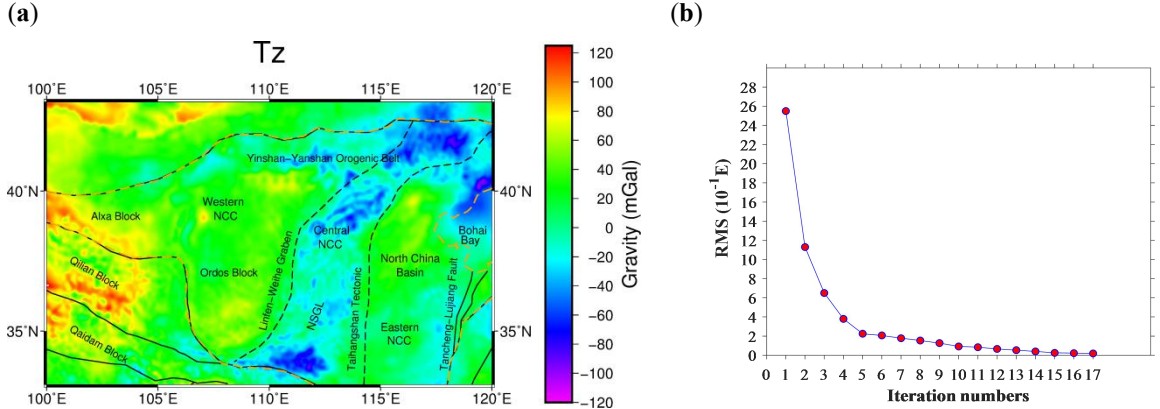

**Figure 5** (**a**) The corrected gravity anomalies after several corrections. (**b**) With the iterative calculation, residual mean square between the forward calculated theoretical gravity and the gravity measurements versus the iteration number in the PCG inversion algorithm.

The three-dimensional density anomaly distribution of the lithosphere within 0-180 km, which is meshed into 0.25° × 0.25° cells (Fig. 6), is inverted from the corrected gravity anomaly. Then, the results are used as the new initial model for gravity gradient inversion. In Sect. 5, the results of the initial and final inversion results (Fig. 6 and Fig. 15) are further analysed and

discussed. In order to highlight the density anomaly at different depths, the color scales of figures are different in Fig. 6. For further comparison of gravity inversion results (Fig. 6) and the gravity gradient inversion results (Fig. 15), the color scales are unified in Fig. 6 and Fig. 15.


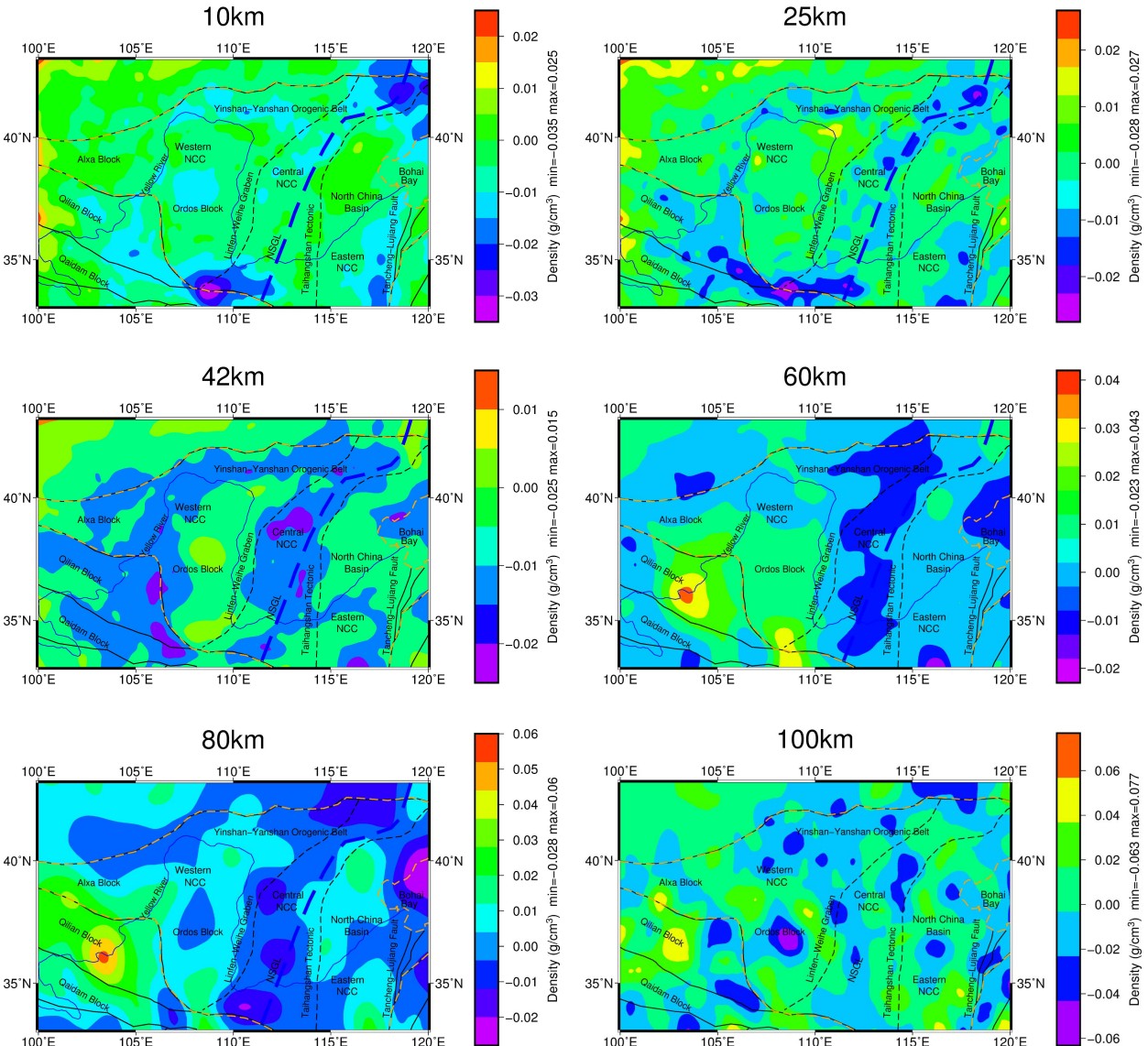

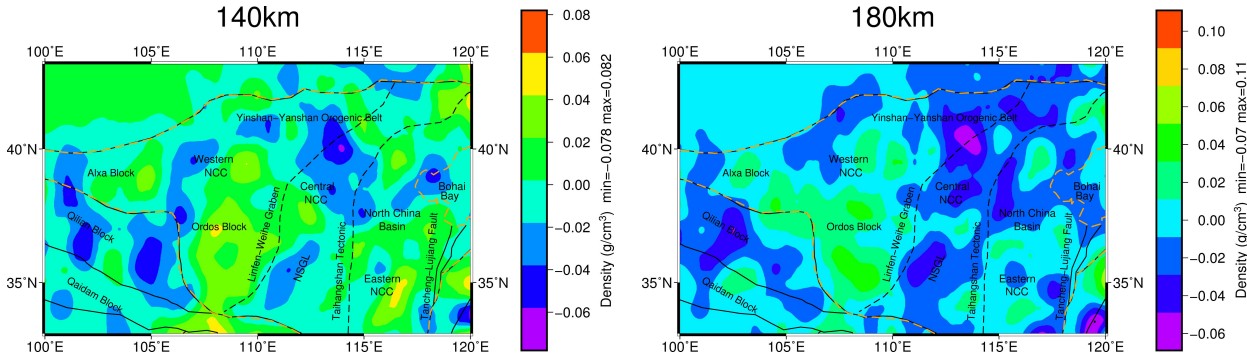

**Figure 6** Lithospheric density distribution of the NCC based on gravity values at different depths.

## 3.2 Obtain GOCE satellite data for study area

The GOCE L2 (GOCE GO_CONS_EGG_TRF_2) gravity gradient measurement product provided for users includes the
gravity gradient data along the orbit. This study directly downloads the preprocessed gravity gradient anomaly data with a
spatial resolution of 10 arc-min, acquired over 48 months from November 2009 to October 2013 (Sebera et al., 2014) from
the website GOCE+ Geoexplore II (http://http://goce.kma.zcu.cz/data.php). These data sets have undergone average orbit
height correction (Sebera et al., 2014) and normal gravity gradient correction (Mortiz, 1980; Šprlák, 2012). The geographic
area within E100°-E120° and N33°-N43° is defined as the study area (Fig. 7a-d), which is where the NCC is located.


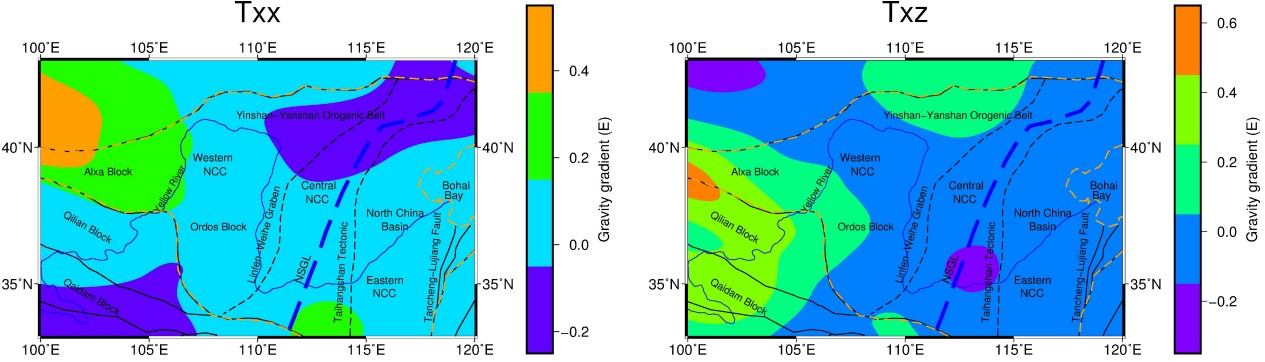

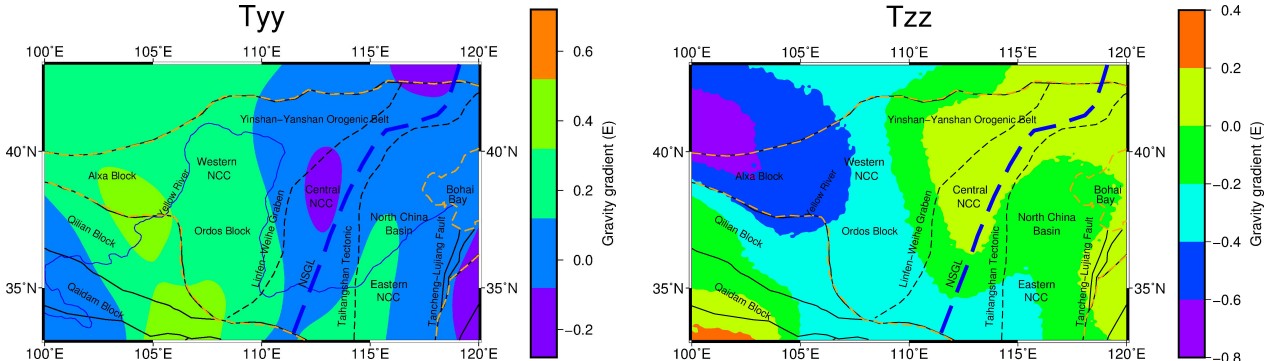

Figure 7 Gravity gradient anomaly ($T_{xx}$, $T_{xz}$, $T_{yy}$, $T_{zz}$) after average orbital correction to an altitude of 250 km.

## 3.3 Downward continuation

The data of the GOCE satellite at the average orbit height only reflect the long wavelength information and the large structural features inside the earth. To highlight the high frequency information of the shallow anomaly body and detailed information of the structural features, such as depth and shape, the GOCE satellite data at the average orbit height for the NCC have been downward continued to the near surface area (Martinec, 2014). The obtained results benefit the lithospheric structural analysis for local areas. The Tesseroids software developed by Uieda et al. (2016) is used in the topography

calculation and the underground interface undulation effects. To ensure the accuracy of the calculation, the software recommends a distance of at least 1 km between the observation plane and the mass body, while the highest point of the topography in the study area is located at approximately 5 km. Thus, the gravity gradient components continued downward to 10 km above the geoidal surface (Fig. 8a-d). The iterative continuation methods based on the Poisson approximate integration theory, specifically developed for the satellite gravity gradient components by Sebera et al. (2014), are adopted,

and the space outside the earth's surface is regarded as mass free.

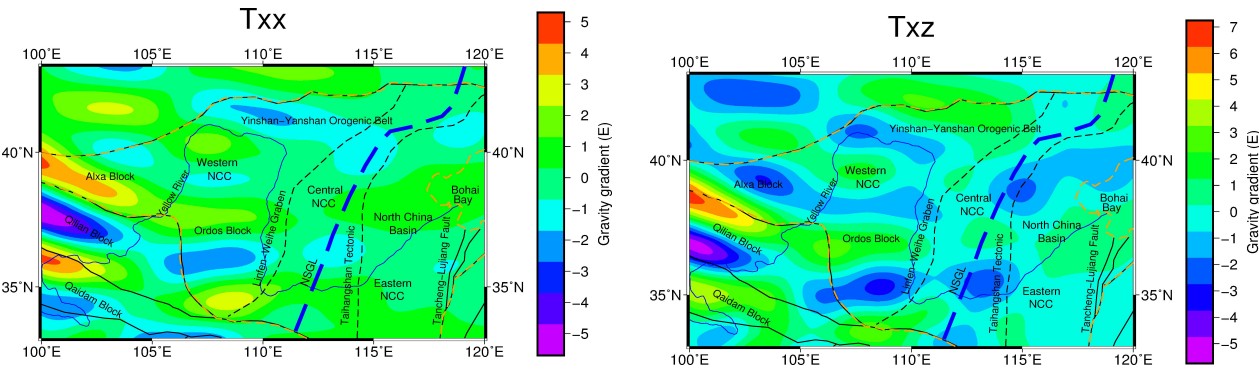

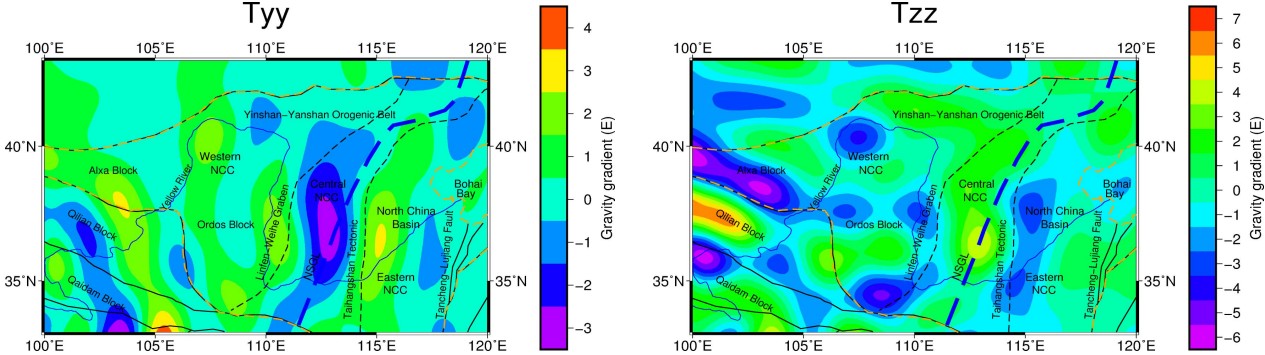

**Figure 8** Gravity gradient effect ( $T_{xx}$ , $T_{xz}$ , $T_{yy}$ , $T_{zz}$ ) after downward continuation to an altitude of 10 km.

### 3.4 Correction for the topographic effects

The gravity gradient anomaly components after downward continuation are a combination of the interface undulation and density heterogeneity. Thus, topographic correction, underground interface undulation correction and long wavelength correction should be carried out in accordance with the existing precise model. These three corrections are affected by the masses within and outside the study area, and far zone corrections outside the study area should be considered, based on previous studies (Szwillus et al., 2016), which points out that for satellite-based gravity gradients, to reduce the remaining rms distant effect to 10% of the global rms, a radius of 10° is sufficient. And for the regional test, the RMS is much more smaller than the global average, only one third of the global rms. Therefore the correction radius in the study area is extended by 10° for all corrections. The extended geographic area within E90°-E130° and N23°-N53° is defined as the calculation area (Fig. 13 e-f). The obtained results in the study area are removed from the calculation area.

First, the topographic effects are calculated using the Tesseroids software (https://tesseroids.readthedocs.io/en/latest) developed by Uieda et al. (2016), which is aimed at reducing the gravity anomaly induced by the topographic mass above the geoidal surface from the gravity gradient after continuation. The Tesseroids software can convert the spherical hexahedron (tesseroid) in spherical coordinates into the prism in Cartesian coordinates to calculate the corresponding gravity gradient effects for topography. The Tesseroids software provides a built-in module to calculate the topographic effects, for which the calculation is based on the topographic elevation data ETOPO1 with a spatial resolution of $1' \times 1'$ (Amante and Eakins 2009).

To directly reveal the correlation between the topography and calculated gravity gradient effects, the three-dimensional figure of the topographic gravity gradient effect $T_{zz}$ is shown in Fig. 9e, where the interface undulation represents the topography (Fig. 1), and the color reflects the gravity gradient effects $T_{zz}$ (Fig. 9d).

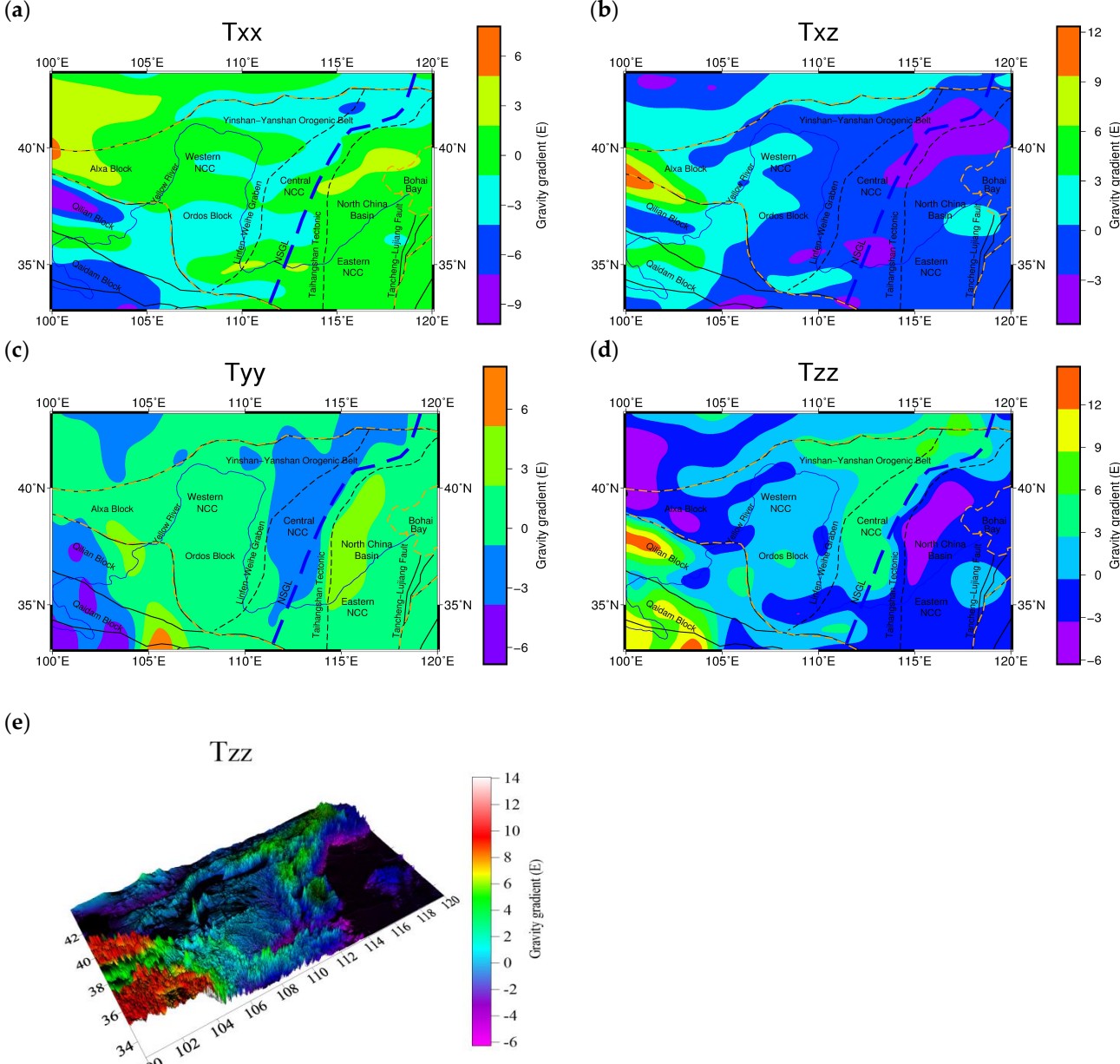

**Figure 9** Gravity gradient effect caused by topographic masses: (**a**) $T_{xx}$, (**b**) $T_{xz}$, (**c**) $T_{yy}$, (**d**) $T_{zz}$, and (**e**) the 3D gravity gradient effect $T_{zz}$ of topography.

### 3.5 Correction for the underground interface undulation effects

First, the average depth and density of the sediment in this area should be calculated for the interface undulation effects. The average depth is calculated from the depth data of the sediment in NCC (Fig. 10a) provided by CRUST 1.0 with a spatial

resolution of 1° × 1° (Laske et al., 2013), and then, the interface undulation data of the sedimentary layer (Fig. 10b) are

obtained after correcting for the average sedimentary layer depth. With respect to the calculated average depth, all the

density data corresponding to the average depth are extracted to calculate the average density corresponding to the average

sedimentary layer depth. The average sedimentary layer depth in the NCC is calculated to be 2.3 km, and the corresponding

average density is 2.45 g/cm$^3$. Based on CRUST 1.0, the actual depth and density are adopted instead of the empirical value.

Moreover, the differences in the actual depth and density at each point from the average depth and density are calculated

using the tesseroid forward modeling method based on the Tesseroids software. Consequently, the gravity gradient effects

induced by the sedimentary layer undulation in the NCC are obtained (Fig. 10c-f).

To more intuitively illustrate the correlation between the sedimentary layer interface undulation and the calculated gravity

gradient effects, in Fig. 10g, the interface undulation denotes the sedimentary layer interface undulation (Fig. 10b), and the

contour lines represent the corresponding gravity gradient effects $T_{zz}$ (Fig. 10f).

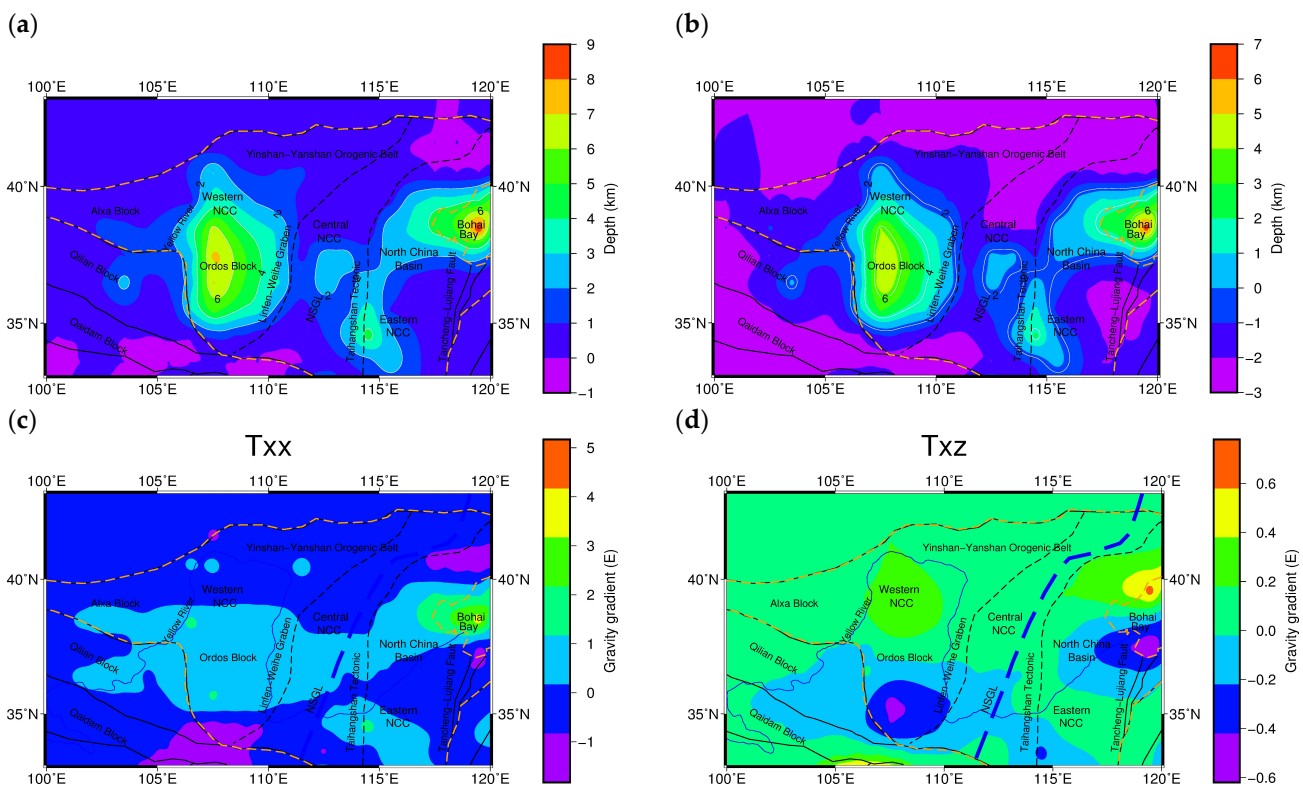

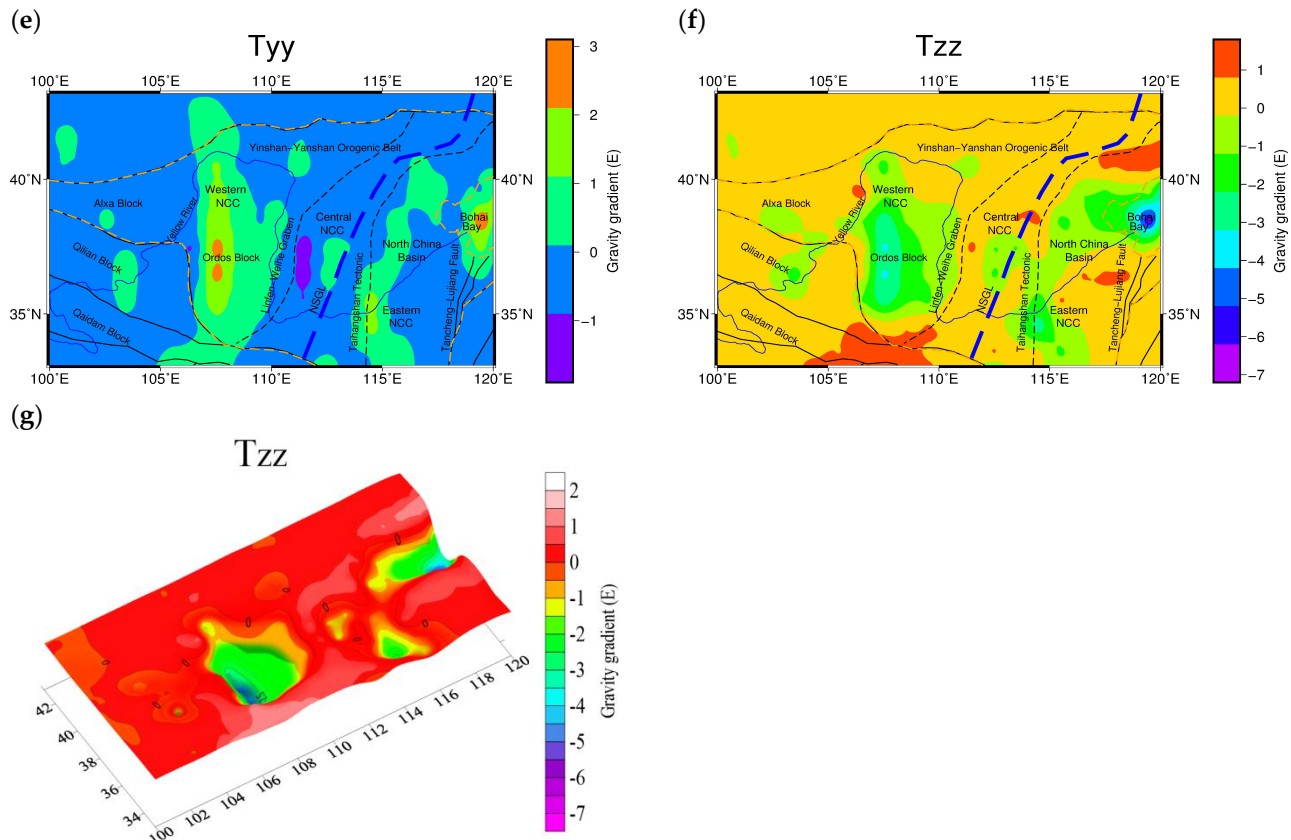

**Figure 10** (**a**) Sedimentary layer interface of the NCC. (**b**) Sedimentary layer interface undulation of the NCC. The anomalous gravity gradient component caused by the relief at the sedimentary layer interface undulation, (**c**) $T_{xx}$, (**d**) $T_{xz}$, (**e**) $T_{yy}$, and (**f**) $T_{zz}$. (**g**) The 3D gravity gradient effect $T_{zz}$ of the sedimentary layer interface undulation.

The depth of the Moho interface in the NCC provided by CRUST 1.0 is shown in Fig. 11a. The average depth calculated from the depth data of the Moho is 36.5 km, and the corresponding average density is 3.32 g/cm$^3$. The Moho undulation after correction for the average depth is illustrated in Fig. 11b. Similarly, the gravity gradient effects induced by the Moho undulation can be calculated. The gravity gradient effects induced by the Moho interface undulation based on the Tesseroids software are presented in Fig. 11c-f. The three-dimensional gravity gradient effect $T_{zz}$ induced by the Moho interface

undulation is shown in Fig. 11g, where the interface undulation reflects the Moho undulation (Fig. 11b) and the contour line represents the gravity gradient effects (Fig. 11f). The gravity gradient effect caused by the sedimentary depth uncertainty and Moho depth uncertainty has been considered in our previous studies(Tian and Wang, 2018), and the anti-noise ability of the inversion results is tested based on the underground layer depth uncertainty, the PCG algorithm has the strong robustness in anti-noise ability to ensure the reliability of the inversion results.


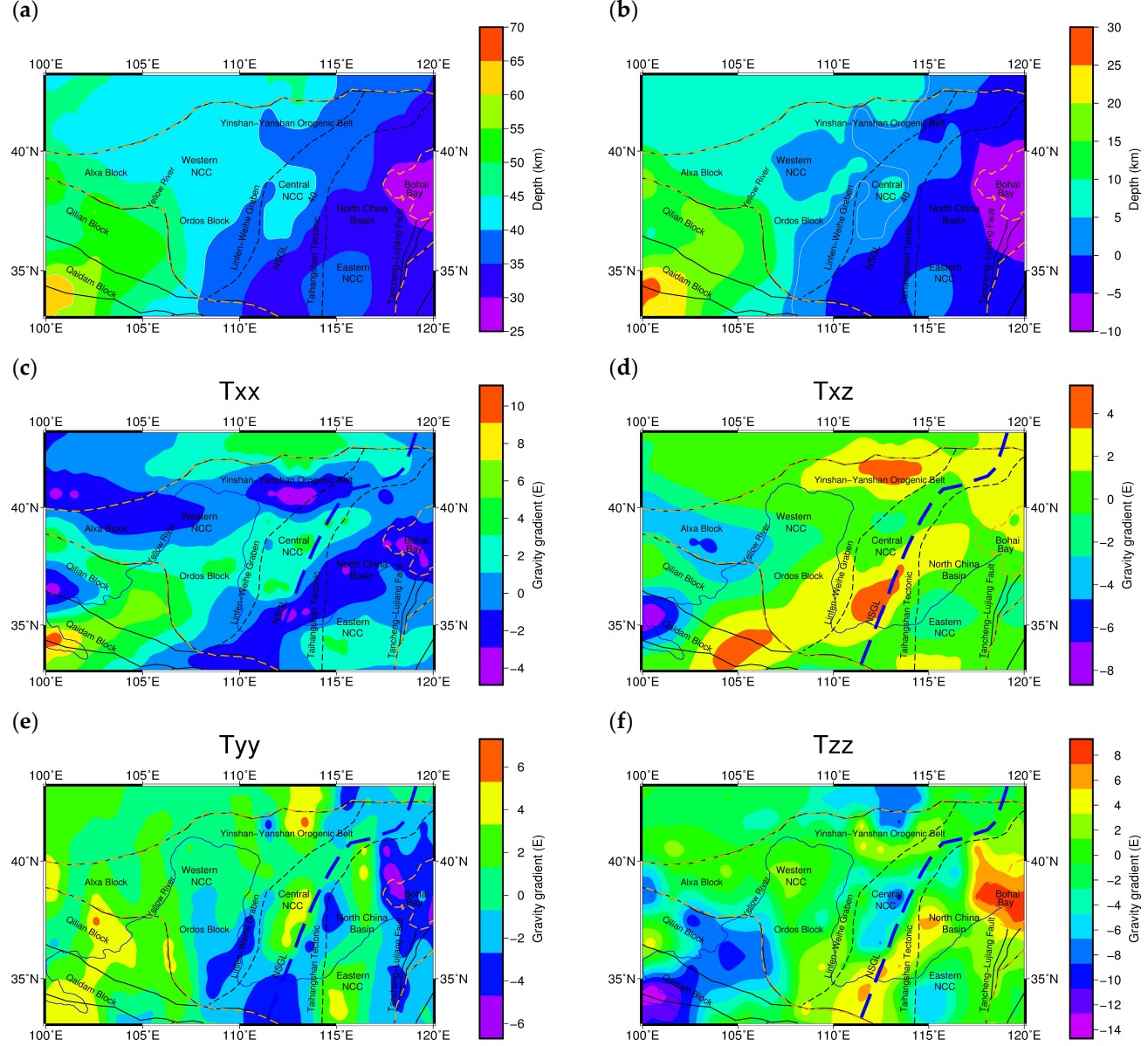

**(g)**

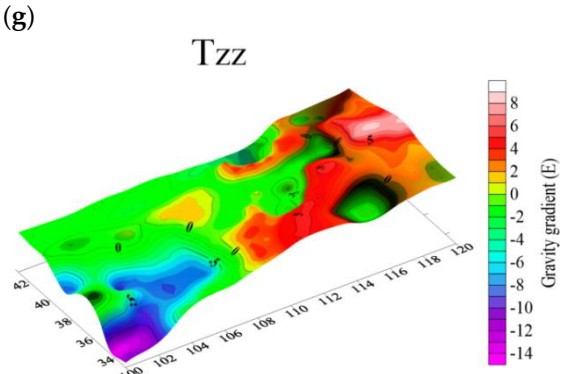

**Figure 11** (**a**) Moho interface of the NCC. (**b**) Moho interface undulation of the NCC. (**c**) The anomalous gravity gradient effect caused by the Moho interface undulation. (**c**) $T_{xx}$ , (**d**) $T_{xz}$ , (**e**) $T_{yy}$ , and (**f**) $T_{zz}$ . (**g**) The 3D gravity gradient effect $T_{zz}$ of the Moho interface undulation.

### 3.6 Correction for the gravity gradient effects of the mantle under 180km

The long wavelength gravity gradient effects are mainly induced by substance density heterogeneity in the lithosphere at depths greater than 180 km. The correlation between the depth of the field source and the order of the spherical harmonic function of the gravity potential developed by Bowin et al. (1986) is adopted as follows:

Z=R/(n-1)                                                                                                        (17)

where Z is the depth of the field source; R is the radius of the Earth; and n is the order of the spherical harmonic function.

The long wavelength gravity gradient anomaly effects corresponding to the 2$^{nd}$-33$^{rd}$ order spherical harmonic coefficients (Wang et al., 2014) in EGM2008 are calculated. The gravity gradient effects induced by the long wavelength are presented in Fig. 12.

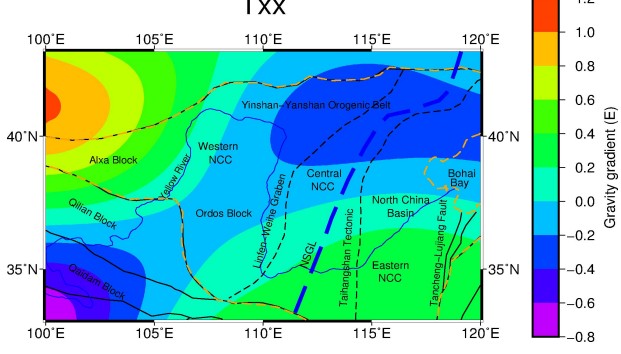

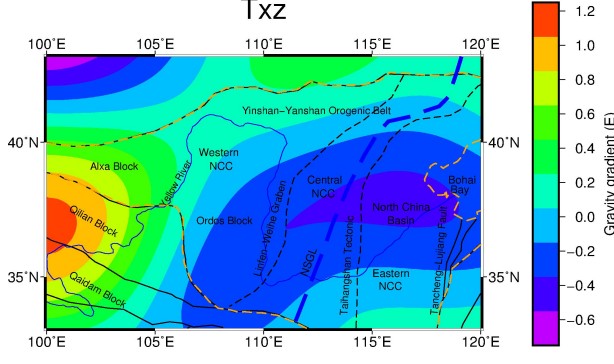

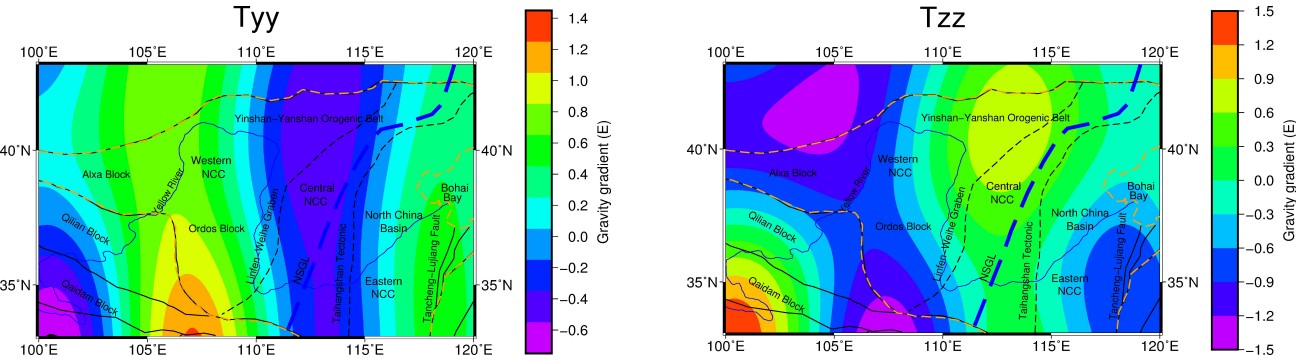

**Figure 12** The gravity gradient effect ($\boldsymbol{T}_{xx}, \boldsymbol{T}_{xz}, \boldsymbol{T}_{yy}, \boldsymbol{T}_{zz}$) caused by the lithosphere under 180 km.

### 440 3.7 The corrected gravity gradient components

Since models with different spatial resolutions are used in several corrections, the spatial resolutions of the obtained results are different. Therefore, before calculating the corrected gravity gradient component, the resolutions of the results for each calculation are homogenized to the same spatial resolution of 0.5° × 0.5°. For the higher spatial resolution of the data (e.g., gravity gradient data after downward continuation and topographic correction), we extracted data from the calculated results.
For the gravity gradient correction caused by the underground interface undulation, the common kriging interpolation method was adopted to obtain the data needed for the spatial resolution. The corrected gravity gradient anomaly components (Fig. 13) are obtained after removing the topography effects, the underground interface undulation effects and the long wavelength effects from the gravity gradient tensor data after downward continuation.

The amplitude of the topography effects and the underground interface undulation effects is obvious, but the topographic correction and underground interface undulation correction balance each other to a certain degree (Szwillus et al., 2016). This counteraction is obvious in the extended study area, as shown in Fig. 13e (topographic correction) and Fig. 13f (underground interface undulation correction). The detailed statistical amplitude for each gravity gradient correction are summarized in Table 3. To present the different corrections more intuitively, several corrections $\boldsymbol{T}_{zz}$ at different cross
sections of latitude 35° and latitude 37.5° are presented in Fig. 13 (g-h).


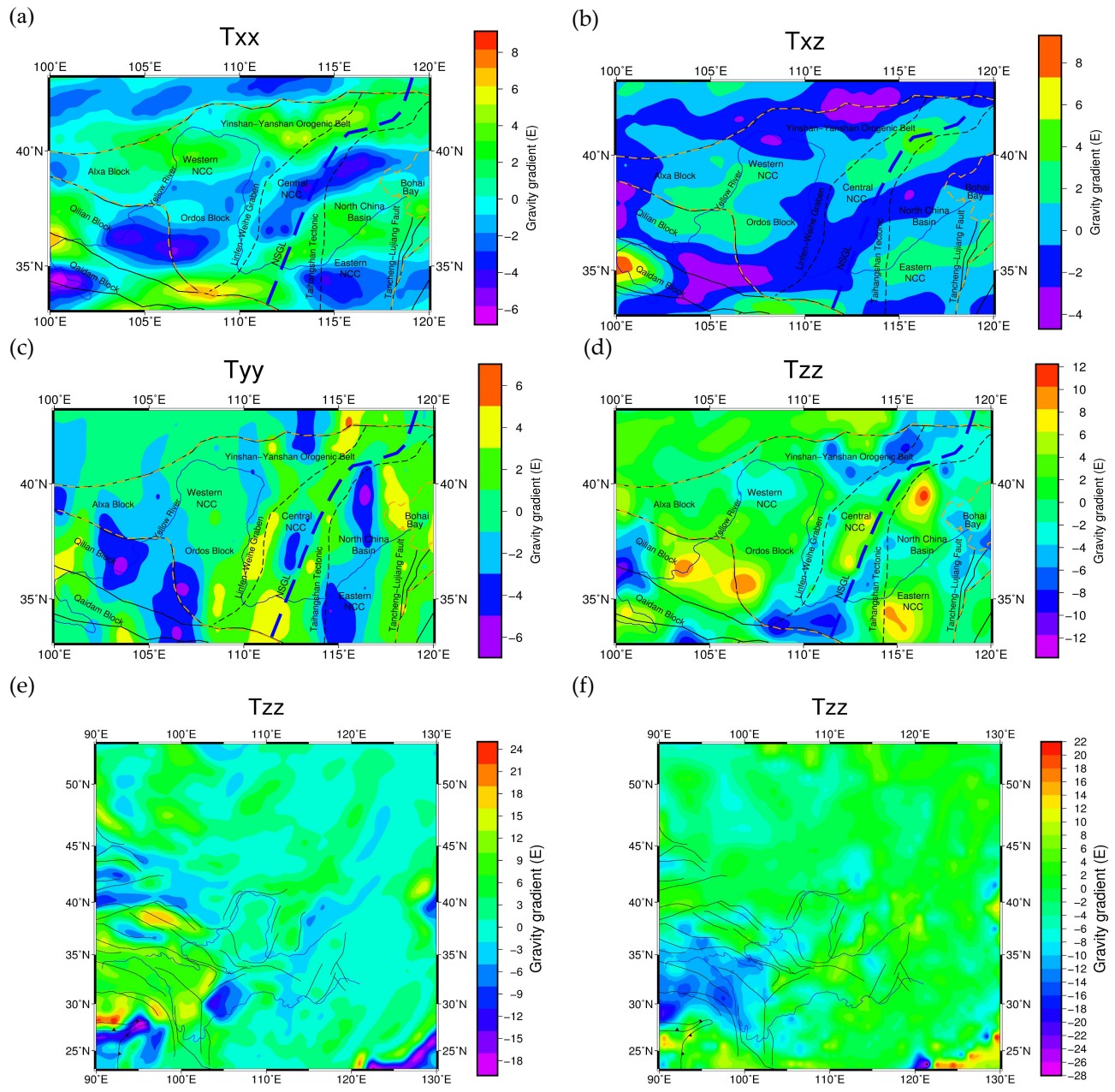

(g) (h)

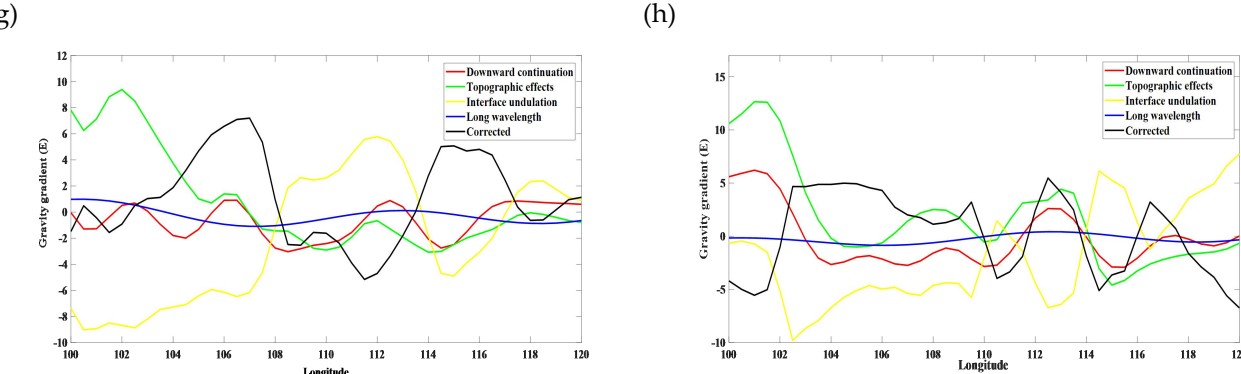

**Figure 13** The corrected gravity gradient components (a) $T_{xx}$ , (b) $T_{xz}$ , (c) $T_{yy}$ , and (d) $T_{zz}$ . Gravity gradient effects with extended area (e)

Topographic effects $T_{zz}$ . (f) Interface undulation $T_{zz}$ (Sedimentary and Moho) effects. Several corrections $T_{zz}$ at different cross

sections: (g) Latitude 35° and (h) Latitude 37.5°.

**Table 3** Statistics for each gravity gradient correction.

| Gravity Gradient Effects (E) | Downward Continuation Maximum (E) | Downward Continuation Minimum (E) | Topographic Effects Maximum (E) | Topographic Effects Minimum (E) | Interface Undulation Effects Maximum (E) | Interface Undulation Effects Minimum (E) |
|---|---|---|---|---|---|---|
| $T_{xx}$ | 5.3 | -5.7 | 6.3 | -9.4 | 10.1 | -4.9 |
| $T_{xz}$ | 7.2 | -5.6 | 11.9 | -5.6 | 5.2 | -8.9 |
| $T_{yy}$ | 4.8 | -3.6 | 7.2 | -6.8 | 6.3 | -6.7 |
| $T_{zz}$ | 7.5 | -6.5 | 13.8 | -6.3 | 7.7 | -14.1 |

| Gravity Gradient Effects (E) | Long Wavelength Effects Maximum (E) | Long Wavelength Effects Minimum (E) | Corrected Effects Maximum (E) | Corrected Effects Minimum (E) |
|---|---|---|---|---|
| $T_{xx}$ | 1.1 | -0.8 | 8.7 | -6.8 |
| $T_{xz}$ | 1.2 | -0.7 | 8.5 | -4.7 |
| $T_{yy}$ | 1.4 | -0.7 | 6.7 | -6.6 |
| $T_{zz}$ | 1.5 | -1.5 | 11.5 | -12.3 |

## 4 Results

In Fig. 14, after 8 iterations, the curve of the defined RMS misfit tends to be horizontal, and the variation is increasingly slighter between the values of the defined RMS misfit. A calculated RMS misfit of $9.0 \times 10^{-3}$ was obtained, at which point the inverse iterative computation was complete. As shown in the inversion results (Fig. 15), obvious heterogeneity is present in

both the horizontal and vertical distributions of the lithospheric density within the NCC, and this heterogeneity is specifically featured by a segmented spatial distribution of the lithospheric density. In order to highlight the density anomaly at different depths, the color scales of figures are different in Fig. 15. In comparison (Fig. 6), the inversion based on the gravity gradient provides more local and detailed information about the density anomaly distribution within the entire NCC. The maximum and minimum values of the inversion results based on the gravity gradient have a larger range, and the detailed data statistics are summarized in Table 4. The center of the anomalies is more concentrated with the regional anomaly features, which is more favorable for the discussion about the stability and destruction in different regions of the whole NCC area. In the eastern NCC, the boundary of density differences on both sides of the Tancheng-Lujiang fault belt zone is more obvious; the extreme value of low-density anomalies is continuously present in the Bohai Bay area at depths of 60 km-80 km. In the central NCC, it is easier to determine the center of the density anomaly distribution in the southern, middle and northern parts of the Taihang Orogenic belt at 42 km-80 km, as these areas have different regional block features of the density anomaly. In the western NCC, the gravity inversion results are connected overall; however, the result of gravity gradient inversion shows the southeastern trend of the Qilian block, which is more favorable for a geodynamic analysis in the western NCC.

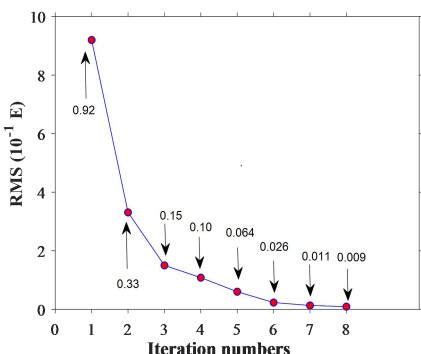

**Figure 14** For the iterative calculation, the residual mean square between the forward calculated theoretical gravity gradient and the gravity gradient measurements versus the iteration number in the PCG inversion algorithm.

**Table 4** Comparison of gravity and gravity gradient inversion results.

| Depth / Inversion Results | 10km (g/cm$^3$) | 25km (g/cm$^3$) | 42km (g/cm$^3$) | 60km (g/cm$^3$) | 80km (g/cm$^3$) | 100km (g/cm$^3$) | 140km (g/cm$^3$) | 180km (g/cm$^3$) |
|---|---|---|---|---|---|---|---|---|
| Maximum density of gravity inversion | 0.023 | 0.021 | 0.011 | 0.035 | 0.053 | 0.056 | 0.058 | 0.079 |
| Maximum density of gradient inversion | 0.025 | 0.026 | 0.015 | 0.043 | 0.058 | 0.070 | 0.077 | 0.091 |
| Minimum density of gravity inversion | -0.033 | -0.026 | -0.022 | -0.020 | -0.024 | -0.061 | -0.066 | -0.064 |

| Minimum density of gradient inversion | -0.035 | -0.028 | -0.025 | -0.023 | -0.028 | -0.063 | -0.078 | -0.069 |
|---|---|---|---|---|---|---|---|---|

## 5 Discussion

In general, obvious density anomalies at a depth of 10 km are only seen in the eastern and central areas of the NCC. At depths of 25-40 km, significant negative density residuals are present in the Taihang Orogenic belt along the central NCC and in the Qilian and Qaidam blocks along the western NCC. Significant high-density residuals exist in the Ordos block. For depths of 60-80 km, low-density residuals are found in the Bohai Bay area in the eastern NCC, while widely distributed low-density residuals are seen in the Yinshan orogenic belt. At a depth of 100 km, the entire NCC is characterized by spatially alternating positive and negative density residuals. In comparison, at depths of 140-180 km, an obvious regional differentiation in the density anomaly distribution is seen, with obvious low-density residuals in the Datong volcano at the junction of the Yinshan-Yanshan and Taihang Orogenic belts in the eastern NCC, as well as the Qilian block at the junction of the Qaidam and Ordos blocks in the western NCC.

According to the horizontal spatial distribution of the lithospheric density, the NCC is divided into three main areas, namely, the eastern NCC, which consists of the North China Basin and Bohai Bay area; the central NCC, which consists of the central transitional belts; and the western NCC, which consists of the Ordos Basin and its surrounding areas. Based on the three main local density anomaly areas in Fig. 15a-h, the density anomaly distribution in each area at different depths is discussed.

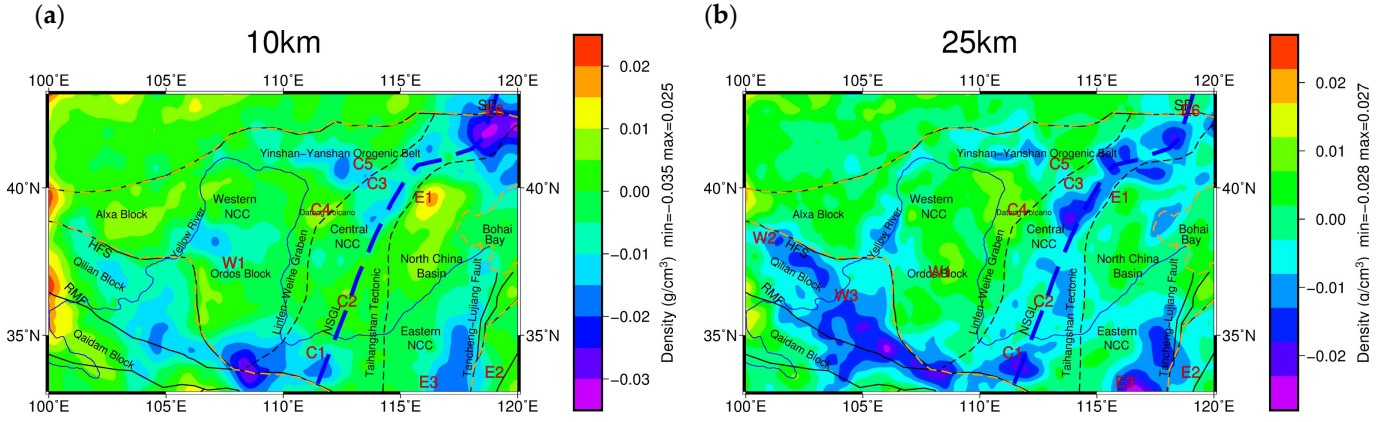

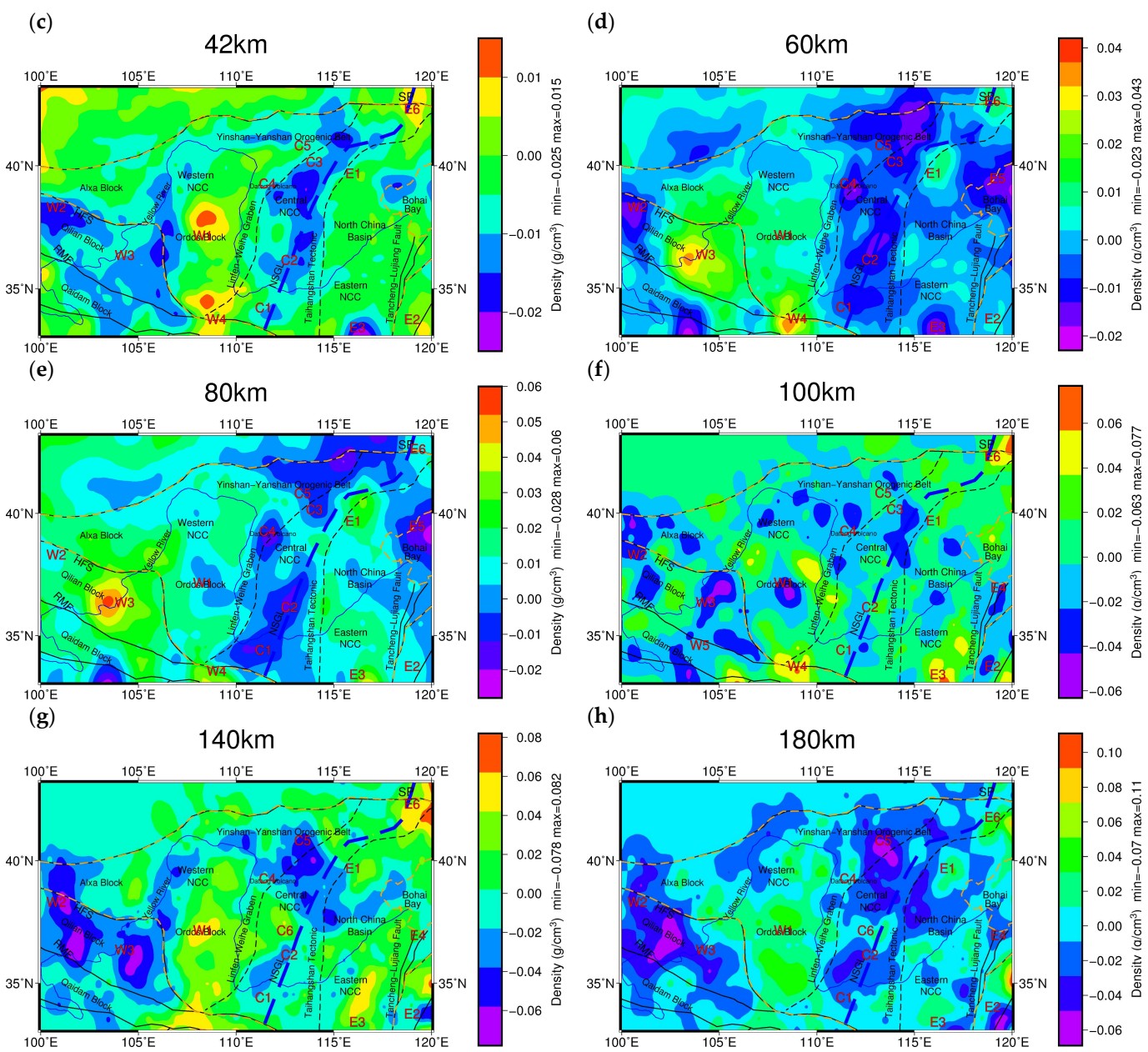

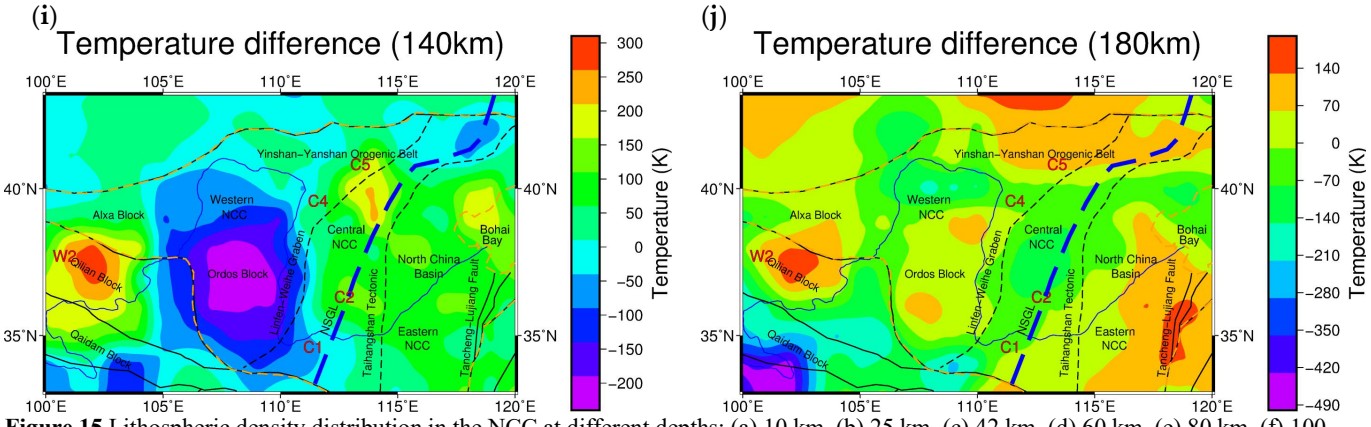

**Figure 15** Lithospheric density distribution in the NCC at different depths: (a) 10 km, (b) 25 km, (c) 42 km, (d) 60 km, (e) 80 km, (f) 100 km, (g) 140 km, and (h) 180 km. Temperature differences at (i) 140 km and (j) 180 km. E represents the eastern NCC, C represents the central NCC, and W represents the western NCC; HFS is the Haiyuan fault system, RMF is the Riyue mountain fault, and SP is the Songliao plain.

### 5.1 Eastern NCC

The eastern NCC is characterized by obvious features with connections to Bohai Bay. In the eastern NCC, the distribution of positive and negative densities is always alternately characterized within depths 0-180 km. From the Tancheng-Lujiang fault belt (E2 and E3) in the south, to Bohai Bay (E5) and North China Basin (E1) in the middle, and the Songliao Basin (E6) in the north, the density always has the features of a regional distribution instead of large-scale distribution in the central NCC. The obvious spatial distributions of the density anomalies are consistent with the theory that this area has experienced strong deformation (Tian and Zhao, 2011).

At the depth interval of 10-180 km in the eastern NCC, there are positive density anomalies in the E1 area of the North China Basin, which are mainly distributed along the Tangshan-Xingtai earthquake belt. This result is consistent with the positive P-wave velocity along the earthquake belt (Huang and Zhao, 2004; 2009), which implies a positive density distribution affected by the Tangshan-Xingtai earthquake belt.

The Tancheng-Lujiang fault belt is one of the major north-northeast fault zones in the East Asia continent. This fault belt extends more than 2400 km in China and cuts through different structural units (Huang et al., 2011). The geological structure of the Tancheng-Lujiang fault belt is complicated. The Tancheng-Lujiang fault belt in the eastern NCC is extended in the north-northeast direction, and on both sides, (the E2 area on the right and the E3 area on the left), the density anomalies are significantly different. At a depth of 42 km, significant positive density residuals are present in the E3 area as the locally extreme value, while regional positive density residuals are observed in the E2 area (Fig. 15c). At a depth of 80 km,

significant positive density residuals are present in the E3 area, while no density anomalies are observed in the E2 area (Fig.
15e). At a depth range of 100-180 km, the density anomalies are obvious (Fig. 15h), including those in the E3 area with a direction consistent with the fault belt strike and those in the E2 area with alternating distributions of positive and negative density residuals. Moreover, these positive and negative density residuals are the local extremes, reaching 0.10 g/cm$^3$ and -0.06 g/cm$^3$, respectively. On both sides of the Tancheng-Lujiang fault belt, there are no connected large-scale density anomalies, and the density anomalies are distributed along the Tancheng-Lujiang fault belt with different features. This distribution represents the density differences between different tectonic units. Therefore, the Tancheng-Lujiang fault belt is obvious as the boundary of the tectonic units. In addition, based on the study of seismic receiver functions (Chen et al., 2006; Li et al., 2011), the Tancheng-Lujiang fault belt zone and its extension are the most obvious areas of lithospheric thinning in the eastern NCC, and the thickness of the lithosphere is only 60-70 km. The significantly different density distributions on both sides of the Tancheng-Lujiang fault belt indicate that the fault belt may have penetrated the lithosphere.

In the E5 area of Bohai Bay, there are significant negative density anomalies at depths of 60-80 km. This area features a columnar shape density distribution at a depth of 80 km. However, as the depth increases from 100 km to 180 km, there are no continuous regional negative density residuals in the E5 area. In contrast, distinct low-density residuals are seen in the E4 area, which is in the extended region of the Tancheng-Lujiang fault belt. Meanwhile, density anomalies are seen in the extended region of the major fault belt in the area adjacent to the eastern boundary of the Tancheng-Lujiang fault belt. According to previous studies (Teng et al., 1997; Su et al., 2009), these density anomalies are ascribed to the extension of the Tancheng-Lujiang fault belt and the development of the mantle plume in the deep Bohai Bay area. However, the distribution of density anomalies in this region does not conform to the features of the mantle plume with continuous negative density residuals in the mantle. Based on this study, these density anomalies are mainly ascribed to the extension of the Tancheng-Lujiang fault belt. Negative density residuals (E4) continuously exist at the extension of the Tancheng-Lujiang fault belt (100-180 km), which implies that the region at the extension of the fault zone has penetrated the crust to the mantle. The cutting effect of the Tancheng-Lujiang fault belt and lithospheric thinning result in asthenospheric upwelling with negative density residuals in Bohai Bay.

In the E6 area, which is a transitional area between the Yanshan block and Songliao Basin, significant low-density residuals are present at a depth of 10 km (Fig. 15a). As the depth increases, persistent high-density residuals are seen in the upper mantle, especially in the depth range of 100-140 km, with values reaching up to 0.07-0.08 g/cm$^3$. Moreover, the density anomalies are aligned in the southwest direction at depths of 140-180 km relative to those at depths of 0-100 km. This phenomenon is attributed to the southwest extension of the high-density mantle lithosphere below the Songliao Basin, reaching the region below the Yanshan Orogenic belt.

## 5.2 Central NCC

The central NCC, formed by the central transitional zone, is characterized by significant low-density residuals with obvious segmented distributions. The Taihang Orogenic belt is generally northeast-southwest oriented, and this belt can be divided into three blocks, namely, southern block C1, middle block C2, and northern block C3. From 42 km to 100 km, C1 is connected to the high-density anomaly area of the western NCC (W4), while C3 is connected to the high-density anomaly area of the eastern NCC (E1) and the low-density anomaly area in the Yinshan-Yanshan blocks (C5). Blocks C3 and C5 are connected, forming low-density anomalies in large-scale areas. These features may indicate that the Taihang Orogenic belt has experienced different geological processes from south to north.

C4 in the northern part of the Linfen-Weihe Graben block is mainly distributed in the Datong volcanic area. As the depth increases, the density anomalies become significant high-density residuals at 25 km (Fig. 15b) and then become obvious low-density residuals from 42 km to 140 km. Although the Datong volcano is no longer active (Tian et al., 2009), its surrounding area still exhibits low-density residuals. Similarly, C5 is at the junction of Yinshan-Yanshan and north of the Taihang Orogenic belt, which features obvious low-density residuals at depths of 60-180 km. When the depth increases to 60 km (Fig. 15d), these anomalies become connected with C3 north of the Taihang Orogenic belt and with C4 at the junction of the Datong volcanic area, forming a large low-density anomaly zone. As the depth further increases, the low-density anomaly area covering C5 presents persistently negative density residuals that extend to a depth of 180 km (Fig. 15h). Moreover, the negative density anomaly value is more significant.

From the distribution of the temperature difference (An and Shi, 2007; Yang et al. 2013) in Fig. 15 i-j, C5 is located in an area with a continuous high heat environment. Based on previous studies using the magnetotelluric imaging method (Zhang et al., 2016), melting occurs in the mantle of the Datong volcanic area and north of the Taihang Orogenic belt. According to the seismic receiver function, the Poisson's ratio in the northern Taihang Orogenic belt is as high as 0.3, while in the southern Taihang Orogenic belt, the value is approximately 0.25-0.26 (Ge et al., 2011). The Poisson's ratio of the continent is generally between 0.25-0.27, and although the temperature and material composition seem to have a dominant influence on Poisson's ratio (Zandt and Ammon, 1995), it is difficult to increase the Poisson's ratio to 0.3 only by changing the material composition. Therefore, the obvious negative density residuals in this area are mainly affected by the high heat environment. The upwelling of the thermal materials from the deep asthenosphere formed the magma migration pathway, which apparently transforms the lithosphere and upper mantle.

In contrast, with the northern region, negative density residuals are observed in the central and southern parts of the Taihang Orogenic belt from 60 km to 180 km, with an extreme value of -0.048 g/cm$^3$ at a depth of 180 km. However, at the depth of the mantle, the central southern part does not show features of a high continuous heat environment (Fig. 15i-j). At a depth of

140 km, the temperature in the southern block of the Taihang Orogenic (C1) is lower than the average temperature, and the temperature is higher than the average temperature only in the middle block of the Taihang Orogenic (C2). As the depth increases, the temperatures in the southern and middle blocks of the Taihang Orogenic (C1 and C2 areas) are all lower than the average temperature at a depth of 180 km. The Poisson's ratio of this area accords with the typical continental features (Ge et al., 2011). Thermal erosion is always accompanied by a high heat flow environment, which is not consistent with this feature in the central and southern regions of the Taihang Orogenic belt. Therefore, it is inferred that the impact of temperature is limited, and the obvious negative density anomalies may be caused by delamination.

C6 in the central NCC exhibits significant positive density residuals at depths of 140-180 km, which are in contrast to the low-density residuals over a large area of the central NCC. The mantle part of C6 is connected to the positive density anomaly area in the W1 area, Ordos block, and western NCC (Fig. 15e-f). In Fig. 15i-j, the temperature in C6 is normal at depths of 140-180 km, and the temperature boundary exists between the C6 and W1 areas. Based on previous studies (Ai et al., 2019), the Taihang Orogenic belt in the mantle part experienced a blocking effect from the rigid Ordos block during the expansion of the orogenic belt. The stable Ordos block area presents continuous large-scale high-density residuals. It is inferred that the Ordos block's blocking effect creates the positive density residuals in the central NCC (C6) and connects with the Ordos block.

**5.3 Western NCC**

At a depth of 42 km, the W1 area in the central Ordos block is characterized by high-density residuals (Fig. 15c), whereas no obvious anomalies are seen in the northern part of the Ordos block. According to the receiver function (Tian and Zhao, 2011), the central part of the Ordos block is 41 km thick, which increases to 45 km in the northern part. Therefore, it can be inferred that the southern part of the Ordos block has entered the lithospheric mantle at a depth of 42 km. From 60 km to 100 km, the high-density residuals in the W1 area of the central Ordos block gradually change to low-density residuals. At the depth interval of 140-180 km, the high-density residuals still dominate the W1 area of the central Ordos block. Although Ordos is a relatively stable block with high-density features, a low-density block invasion appears at depths of 80-100 km. At a depth of 80 km, the low-density anomaly of the Ordos block (W1) on the east side is connected with the low-density anomaly in the central NCC (C2), which indicates that Ordos is affected to some extent by the destruction of the central NCC. At a depth of 100 km, the low-density anomaly of the Ordos block (W1) is connected to the Qilian block (W3) and Qaidam block (W5), and the distributions of the density anomalies are consistent with the theories that the Ordos block is affected by the northeast compression of the Qinghai-Tibet Plateau and substantial deep expansion (Sheng et al., 2015).

The whole Qilian block (W2 and W3 areas) features significant low-density residuals in the crustal part at 25-40 km, which falls in the range of the Haiyuan fault belt. The boundary of the negative density residuals is essentially consistent with the

strike of the fault belt shown on the surface. The W3 area at the junction of the Qilian and Ordos blocks is characterized by a sharp transition from low density to significant high density. At depths of 25-80 km, the density anomalies in the Qilian block present a dominant NW-SE strike; however, this strike turns clockwise to the NS direction as the depth further increases to 180 km, especially after entering the mantle. This observation is consistent with previous research results in this region. The mantle convective stress field calculated based on gravity anomalies is demonstrated to be significantly inconsistent with the crustal movement pattern (Xiong and Teng, 2002; Wang et al., 2013), which indicates obvious decoupling between the crustal and mantle materials in the Qilian block. The Qilian block is located at the junction of the northeastern Qinghai-Tibet Plateau and the Alashan and Ordos blocks in the western NCC. The Qilian block has not only been influenced by collisions and subduction of the Indian plate but also by the blocking effect of the Ordos Block, which results in a strong regional tectonic stress background in the deep part of the Qilian block with an associated clockwise rotation of the material movement.

As the depth increases in the range of 140-180 km, persistent obvious negative density residuals are increasingly dominant in the Qilian block (W2 areas) of the western NCC. According to a previous study (Teng et al., 2010), the lithosphere in the western NCC is the thickest, with an average depth of 140-150 km. However, within this depth range, obvious low-density residuals are present in the orogenic belts of the western NCC, which eliminates the possibility that lithospheric thinning is the main cause of the density anomalies at depths of 140-180 km. According to the temperature differences (Figure 15 i-j), the Qilian block in the western NCC has a high temperature at depths of 140 km-180 km. Furthermore, according to research on the terrestrial heat flux (An and Shi, 2007), the Qilian block has an average heat flux value of up to 68.340 mW/m$^2$. Therefore, it can be concluded that the high heat flux environments lead to negative density residuals in the Qilian block in the western NCC, accompanied by an upwelling of deep asthenosphere materials and the subsequent transformation of the mantle above the lithosphere.

## 5.4    The destruction mechanism of NCC

The low-density anomalies in the Qilian block and northern Taihang Orogenic belt are affected by the high heat flux environments. Low-density anomalies in the central Taihang Orogenic belt exist but are not accompanied by continuous high heat flux environments at mantle depth. However, without mantle plumes, the low-density anomalies in Bohai Bay are affected by the extension of the Tancheng-Lujiang fault belt, and mechanical extension destroys the lithosphere in this area. Based on our studies, one theory is that it is hard to explain the destruction phenomena and modes in the whole NCC. The destruction of NCC is not only affected by physical tension but also caused by thermal erosion and delamination. Previous dynamic studies have shown that (Zhu et al., 2012; Zhu, 2018) since the Mesozoic, the Pacific Plate subducted westward to the Taihang Orogenic belt in the central part of the NCC. The residual dehydration of the subducted plate in the mantle transition zone promoted an increase in the molten fluid content in the upper mantle beneath the NCC. The delamination and

thermal erosion of the lithosphere in the NCC reflect different forms of mantle convection instability. Therefore, through this study, it is believed that the destruction in the NCC is caused by several forms. Several destruction modes of the NCC coexist in different geological structural backgrounds.

## 6 Conclusions

The sequential inversion of the lithosphere density structure in the NCC is divided into two stages. The effects of the initial

density model are considered. The inversion results obtained by the inversion of the corrected gravity anomaly are used as the initial model for the inversion of the GOCE satellite gravity gradient components. The GOCE satellite gravity gradient data were processed with several corrections to obtain the corrected gravity gradient components. The density distribution with a depth range of 0-180 km in the NCC is outlined as follows: (1) In the eastern NCC, affected by lithosphere thinning, the eastern NCC has local features in the density anomaly distribution. Obvious differences in the density anomaly

distribution are observed, and the Tancheng-Lujiang fault belt in the eastern NCC penetrates through the lithosphere. The density anomaly in Bohai Bay is mainly induced by the extension of the Tancheng-Lujiang major fault at the eastern boundary. (2) In the central NCC, the Taihang Orogenic belt located in the central NCC is characterized by a segmented density anomaly distribution. (3) In the western NCC, the Qilian block in the western NCC presents a clockwise rotation of the density anomaly distribution with an increasing lithospheric depth, while the adjacent Ordos block remains continuously

stable. (4) Across the Taihang Orogenic belt in the central NCC and the Qilian-Qaidam blocks in the western NCC, stronger impacts of the orogenic belt and a high heat flux environment are observed, which results in an upwelling of the deeply buried asthenospheric substances and consequently results in a reconstruction of the lithospheric density structure distribution.

**Code and data availability.** The on-orbit GOCE gravity gradient data can be viewed and downloaded from GOCE+ Geoexplore II (http://goce.kma.zcu.cz/data.php). The "Tesseroids" software for gravity gradient topographic correction and interface undulation can be viewed and downloaded from https://tesseroids.readthedocs.io/en/latest. The global crustal model CRUST1.0 can be downloaded from https://igppweb.ucsd.edu/~gabi/crust1.html. The processed gravity gradient data and inversion results can be viewed and downloaded from https://zenodo.org/record/3545809#.XdJ5H695vIU.

**Author contributions.** Conceptualization, YW; methodology, YT; validation, YT, writing original draft preparation, YT; writing review and editing, YT; supervision, YW; funding acquisition, YW.

**Competing interests.** The authors declare no conflict of interest.

**Acknowledgements.** We are grateful to editor Prof. Mioara Mandea, and two referees Cécilia Cadio and Josef Sebera for

providing a lot of constructive comments and suggestions, which really help us a lot to make remarkable progress. We are

grateful to Prof. Xinsheng Wang in Development Research Center of China Earthquake Administration for providing

corrected gravity data. We are also grateful to Prof. Qi Lin and Prof. Bojie Yan in Minjiang University for providing a lot of

help when we prepared this paper.

**Financial support.** This research was funded by the R&D of Key Instruments and Technologies for Deep Resources

Prospecting (the National R&D Projects for Key Scientific Instruments), grant number No.ZDYZ2012-1-04.

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
