# Peer review of "Sequential inversion of GOCE satellite gravity gradient data and terrestrial gravity data for the lithospheric density structure in the North China Craton"

_Solid Earth, 2019_

## Referee Comment (RC1) · Anonymous Referee #1 · 31 Jan 2020

**Joint inversion of the lithospheric density struture in the North China Craton based on GOCE satellite gravity gradient data and surface gravity data**

Yu Tian & Yong Wang

The presented manuscript brings a joint two-step inversion of terrestrial gravity data and gravity gradiometer data from the GOCE mission. The authors start with simulated tests and continue with the applications in the area of the North China Craton. Besides the comments below, the paper is well structured and clearly follows the topic so that it can be recommended after a **major revision**. The most important issues include: i) clarifying the synthetic tests in Table 1 and 2 and settings of the colorbar spans (Figure 2 and 3), ii) the discussion of the long wavelengths with respect to the kernel (tensor component) and the far zones, iii) proof-reading by a native or an experienced speaker (some complex sentences need to be revisited).

**General comments**

- Terminology - the paper denotes the gravity anomaly/gradient data sets that were reduced for multiple effects as "remaining" gravity anomaly/gradient. Using a term that is commonly used in gravimetry would be more appropriate (e.g., "corrected" for the effect of " etc.)

- Throughout the paper: "resultant data" → "resulting data"?

- Figures - on the top of the colorbars is a slash followed with a unit (probably coming from the panels with densities). Can it be removed (it is confusing)?

- In the text, white spacing is often missing when Figures are mentioned, look for "...(Fig...)"

- IMPORTANT: Figures 2,3 have different colorbars for the input and the output. Please, update them to have the same colorbar span.

**Detailed comments** (line(s))

- 14 *Inconsistency* - do you mean a difference in the data representation (geometry)?

- 19 *performed* → applied to?

- 20 *observation quantity* → observations?

- 23-24 *high heat flux* ... could add a reference to this sentence?

- 36-38 please reformulate or leave this sentence "*The **most** direct and effective approach...*" because the following one repeats it in other words. Btw, the non-uniqueness of the gravity-based inversions make the gravity data not the easiest means for understanding the subsurface.

- 38 *variations* - this little implies time variations. Please, reformulate.

- 38 "*laws*"? Gravimetry does not study the laws behind measurements but it uses the laws to study phenomenas in the region of interest. Please reformulate

- 49 Note ... gravity data contains all the frequencies about the whole Earth, but the high frequencies of the deep structures are just strongly attenuated due to the distance. If we would substantially improve S/N of the gravimeters, we could infer hi-freqs from the very deep density contrasts - the signal is there, just masked out by the signal of the closer masses. The point is however that the data contains it all but due to the attenuation and the errors we do not have access to it.

- 55 a typo "*can expands*"

- 56 leave out the end of the sentence (repetition): "..., *which promotes ...*"

- 57 shorten the sentence (repetitions), e.g. "Joint inversion of the gravity and gravity gradient data can enhance the reliability of the inversion result."

- 60 "*rare*"? (please reformulate), search for the topic, e.g. `https://scholar. google.cz/scholar?hl=cs&as_sdt=0%2C5&q=joint+gravity+and+gradient+inversion& btnG=`

- 62 shorten "... *realized the inversion combining gravity and gravity gradient*" → inverted gravity and gravity gradient?

- 71 it should be EGM2008

- 71 "*in the case of*"

- 72 leave out "*in local areas*"

- 75 remove "*calculations*"

- 75 correct to "*coefficients*"

- 76 remove "*actual*"

- 77 correct to "*advantage*"

- 78 remove "*actual*"

- 81 rather "*GOCE satellite gravity gradients data along the orbit*" than "*on orbit GOCE*"

- 88 "... *on the topography...*" (remove "*surface of the topo...unit*")

- 98 remove "*at the first process*" ... clear from the context

- 104 remove "*at the second process*" ... clear from the context

- 104 Start a new paragraph with "*By considering ...*"

- 120 the sentence "Since" does not make sense, please update

- 121 *undetermined* is related to a number of data but you rather want to emphasize that the problem is ill-conditioned right?

- 140 "*which is expressed ...*" repeats the first part of the sentence (remove it)

- 143 "is comprehensive" not clear what you mean (please reformulate)

- 146 ?"...while a very small value may lead to a large ...the model and the data, thus providing unrealistic ..."??

- 147 introduce the L-curve method before you write it is used

- 167 Reformulate the sentence and see Martinec, Z. (2014). Mass-density Green's functions for the gravitational gradient tensor at different heights. Geophysical Journal International, 196(3), 1455-1465. ... not all gradients have the kernel with the properties as you describe (what you describe is related to the radial derivatives)

- 176 IMPORTANT. You use some damping function to revert the kernel behaviour. Why do you use the same for all the gradients if their kernels behave differently?

- 202-228 Consider leaving the whole paragraph on PCG since this method is well described and established in the community. The paper is already quite long, here is the potential to shorten it.

- 230 remove "*further*"

- 231 "..., which is relatively simple" → "equipped with density values"

- 231 sentence is bit cumbersome → e.g., "The densities are different at each layer to study the role of the initial values in the joint inversion."

- Table 1 IMPORTANT. Could you add the real densities used in the model. The table gives just the initial values but not true ones so the reader cannot compare the result with the truth.

- 266 You claim "that is almost identical to the true model". If I have not overlooked something, I did not find the true values so the claim cannot be checked by the reader. As said above, updating Table 1 with true values would help. Note "almost identical" is soft characteristics - it is more convenient to use RMS or so.

- 276 concluded → given in Table 2

- 278 ".... a more precise initial model" and/or more data, right?

- Table 2 - update with true value

- 296-288 - the last sentence might be left out, not clear what it brings.

- 300 *...decomposed into two integrated ...* → is composed of two steps?

- 317

- 319

- 321

- 331 we construct

- 335 What is meant by "*outer*" iterations. Are there any inner iterations so that these two have to distinguished in the paper?

- Figure 5: the red line is hardly visible, maybe not needed in this figure

- Figure 6: density  variation (it is not a full value right?)

- Figure 6: "" ... km is well visible from each panel, no need to repeat it in the caption.

-  ... clear from the context

- 359 IMPORTANT . The webpage is not maintained by ESA and ESA is not responsible for that.

- 359 update to *These data sets have*

- 360 cite normal gravity with "Mortiz, H. (1980). Geodetic reference system 1980 (GRS80). Bulletin Geodesique, 54".

- Figures 7,8,9: delete (a)...(d) ...clear from the panels

- 384 replace "*deducting*"

- 389 "*In order to directly...*" should continue with a second part, but the sentence is not complete probably.

- Section 3.4 Please be aware of the paper that investigates this effects: Importance of far-field topographic and isostatic corrections for regional density modelling 2016, W Szwillus, J Ebbing, N Holzrichter Geophysical Journal International 207 (1), 274-287

- Figure 9: update the caption, it is not clear what the panel (e) is showing (a 3D map or Tzz as written in the title?)

- Figure 10 the same for (g), update unit in the colorbar of this panel (g). Shorten the caption as the caption repeats all the same (you can put "anomalous gravity gradients Txx, Tyy ...etc" into one sentence)

- 421: You probably mean the gravity effect of the layer bounded by topo and Moho, right? The sentence implies to calculate the gravity effect of the Moho surface.

- Figure 11: shorten the caption - lot of repetitions.

- Section 3.6 - the text implies that this correction is due to the masses below the area of interest. This is was recently studied by Szwillus et al. (2016) showing the importance of lateral masses. Please mention this problem.

- EGM2008

- 442: what is meant by "*relatively minor*"? - the magnitudes are up to 1.5 E, which quite large

- Figure 12: remove "*(a)...(d)*" ...clear from the panels

- 445: not clear what the sentence tries to tell (what official website, why this is linked with the accuracy/reliability of the data?)

- Figure 13: caption can be shortened, gradients clear from the panels directly

- 455: "*outer iterations*"?

- Figure 14: unit is missing in the y-axis

- 470: "*negative density anomalies*" → negative density contrast/residuals?

- 481-482: sentence wording, also

- 489:  obvious

- 505: simplify (does not make sense) "... demonstrate persistently present significant differences ..."

- Figure 15: simplify the caption (km is obvious from the panels directly)

- 619: at the given website `goce.mka.zcu.cz` data cannot be browsed (how?), just it can be downloaded? Please change the sentence if true.

---

## Referee Comment (RC2) · Anonymous Referee #2 · 7 Feb 2020

Referee's comments on the manuscript se-2019-181 Title: "Joint inversion of the lithospheric density structure in the North China Craton based on GOCE satellite gravity gradient data and surface gravity data" Authors: Yu Tian and Yong Wang

In this work, the authors jointly inverse GOCE satellite gravity gradient data and surface gravity data for the distribution of their source density anomalies in the North China Craton lithosphere. The inversion method is based on the preconditioned conjugate gradient inversion algorithm and is implemented in two "independent" parts. The first one concerns the inversion of surface gravity data after corrections. The resulting

density distribution is then used in the second part as the initial density model in the preprocessed remaining gravity gradients inversion. The gravity gradients inversion results are then discussed in terms of geological structures in the North China Craton.

General comments The paper presents an interesting method allowing the combination of the gravity and gravity gradients data in the same inversion scheme, which remain rare in the community. Their inversions exploit both the high quality of gravity and gradiometric data and their mutual supplementation, which can greatly reduce the non-uniqueness of the inversion and enhance the reliability of the results. The paper is well-written in a good English, well-structured, generally clear and detailed. The quality of the figure is adequate and the number of them is sufficient.

However, I have major comments that need to be addressed before publication. One of the major issue of the paper is that the authors never discuss and quantify the impact of each correction applied on the data before the inversion on the resulting density models in terms of resolution and amplitude. The authors choose to use gravity and gravity gradient data obtained from measurements and not derived from the gravity field models in order to preserve their high precision, which is indeed relevant. However, the applied corrections imply the use of models not well constrained as the CRUST 1.0 model which highly compromise the high quality of the data and thus the resolution of the inversion results. Another major issue concerns the inversion methodology of the gravity gradients. In this paper, the authors focus on the 4-high accuracy GOCE satellite gravity gradient tensor components ($T_{xx}$, $T_{xz}$, $T_{yy}$, $T_{zz}$). The authors do not explain if their method inverts these components separately or simultaneously and do not discuss the contribution of each component in the inversion results. The last major issue regards the discussion of the results in terms of geological structures and geodynamical processes in the North China Craton. The authors should remind what is/are the fundamental question(s) in this region and in what this study brings answers or at least new constraints. This is not clearly specified in the current version of the paper. Finally, the authors interpret some density anomalies as thermal variations in

the mantle without any quantification. They can easily calculate the density anomalies produced by such thermal variations and verify if their interpretation is plausible. In a general way, in their interpretation, the authors should systematically and clearly relate the density anomaly to the geological structure or to the geodynamic process which is not the case in this current version.

This manuscript responds to the Solid Earth criteria for publication. I recommend this paper for publication once the above main concerns will be addressed as detailed as possible.

Please find attached to this message the pdf file containing the detailed revision.

Please also note the supplement to this comment:
https://www.solid-earth-discuss.net/se-2019-181/se-2019-181-RC2-supplement.pdf

[Figure]

**Supplement:**

**Referee's comments on the manuscript se-2019-181**
Title: "Joint inversion of the lithospheric density structure in the North China Craton based on GOCE satellite gravity gradient data and surface gravity data"
Authors: Yu Tian and Yong Wang

In this work, the authors jointly inverse GOCE satellite gravity gradient data and surface gravity data for the distribution of their source density anomalies in the North China Craton lithosphere. The inversion method is based on the preconditioned conjugate gradient inversion algorithm and is implemented in two "independent" parts. The first one concerns the inversion of surface gravity data after corrections. The resulting density distribution is then used in the second part as the initial density model in the preprocessed remaining gravity gradients inversion. The gravity gradients inversion results are then discussed in terms of geological structures in the North China Craton.

General comments
The paper presents an interesting method allowing the combination of the gravity and gravity gradients data in the same inversion scheme, which remain rare in the community. Their inversions exploit both the high quality of gravity and gradiometric data and their mutual supplementation, which can greatly reduce the non-uniqueness of the inversion and enhance the reliability of the results. The paper is well-written in a good English, well-structured, generally clear and detailed. The quality of the figure is adequate and the number of them is sufficient.

However, I have major comments that need to be addressed before publication. One of the major issue of the paper is that the authors never discuss and quantify the impact of each correction applied on the data before the inversion on the resulting density models in terms of resolution and amplitude. The authors choose to use gravity and gravity gradient data obtained from measurements and not derived from the gravity field models in order to preserve their high precision, which is indeed relevant. However, the applied corrections imply the use of models not well constrained as the CRUST 1.0 model which highly compromise the high quality of the data and thus the resolution of the inversion results.
Another major issue concerns the inversion methodology of the gravity gradients. In this paper, the authors focus on the 4-high accuracy GOCE satellite gravity gradient tensor components ($T_{xx}$, $T_{xz}$, $T_{yy}$, $T_{zz}$). The authors do not explain if their method inverts these components separately or simultaneously and do not discuss the contribution of each component in the inversion results.
The last major issue regards the discussion of the results in terms of geological structures and geodynamical processes in the North China Craton. The authors should remind what is/are the fundamental question(s) in this region and in what this study brings answers or at least new constraints. This is not clearly specified in the current version of the paper. Finally, the authors interpret some density anomalies as thermal variations in the mantle without any quantification. They can easily calculate the density anomalies produced by such thermal variations and verify if their interpretation is plausible. In a general way, in their interpretation, the authors should systematically and clearly relate the density anomaly to the geological structure or to the geodynamic process which is not the case in this current version.

This manuscript responds to the Solid Earth criteria for publication. I recommend this paper for publication once the above main concerns will be addressed as detailed as possible.

Specific comments

**Title**. The authors invert the data not the density structure of the lithosphere. I suggest "Joint inversion of GOCE satellite gravity gradient data and surface gravity data for the lithospheric density structure in the North China Craton".

**Methods/Kernel function.** The author should better explain what is the kernel function used in the gravity inversion and the gravity gradient inversion? They can add a reference if this function is already well explained in another paper and a small explanation here with figure or equation.

**Methods/Joint inversion (section 2.3).**
- Can we talk about joint inversion when the inversion is realized in two steps? The term of "sequential inversion" would be more appropriate here (see Lines et al., 1988 – Cooperative inversion of geophysical data).
- How the four GOCE satellite gravity gradient tensor components are inverted? Separately? Simultaneously? Which is the contribution of each component in the inversion results?
- What about the two others components not used in this study? Despite the errors on these components, they really cannot bring any useful constraints?
- How the kernel function is calculated in the gravity gradient inversion? One kernel function by component or one for the all of them?

**Data processing/remaining gravity anomaly data (section 3.1).**
- The authors should remind the origin of the gravity data used in this study.
- 5 arc-min, real resolution of measurements or resolution only after interpolation?
- The authors should specify in the text what the interface undulation correction and long wavelength correction mean for them (this is clarified later in the paper but this explanation is necessary for the reader comprehension in this paragraph).
- Which are the remaining wavelengths in the final gravity anomaly data?
- The authors say that the sources responsible for these gravity anomalies are only located in the lithosphere. Are they sure about that? The authors should explain how and why they do a such hypothesis?
- The resolution of the tomography model used for the data correction is 0.5°x0.5°. Yet the resolution of the inversion results is 0.25° x 0.25°. How is possible? The resolution of inversion results has not to be higher than the resolution of the inverted data. The author must adapt the resolution of the inversion in function of the resolution of corrected data.
- Fig. 5: The author should modify the color scale. Only the minima and maxima are clearly visible on the figure. The high resolution mentioned in the text is not perceptible. It is difficult to compare these results with the results of the gravity gradient inversion (Fig. 6).

**Data processing/correction for the underground interface undulation effects (section 3.5).**
- The resolution of the CRUST 1.0 model used for the data correction is 1°x1°. The resolution of the inversion results is 0.25° x 0.25°. I have the same question: How is possible? The resolution of inversion results has not to be higher than the resolution of the inverted data. The author should adapt the resolution of the inversion in function of the resolution of corrected data.

- What is the impact of these corrections (sedimentary layers and crust) on the final inversion results? The amplitude of these corrections is much greater than the residual signal used for the inversion. It appears crucial that the authors must clearly quantify the effects of the CRUST 1.0 uncertainties on their final results and discuss them in light of these estimates.

**Data processing/correction for the long wavelength effects (section 3.6).**
- The authors say that this effect is minor. They should directly quantify and clearly state how many it is. For the Tzz, I compute 10% of the total signal. It is not so minor.
- Why the authors use the EGM 2008 model for this correction and not a model based on the GOCE data only or even better their own data developed in spherical harmonic? I really do not understand this step of treatment.

**Results.**
- What the gravity gradient data inversion brings compared to the gravity data inversion? The author should discuss about this in terms of amplitude and distribution of the density anomalies. The difference is it significant?

**Discussion.**
- General comment: 1) The authors should remind what is/are the fundamental question(s) in this region and in what this study brings answers or at least new constraints in each study area. 2) In their interpretation, the authors should systematically and clearly relate the density anomaly to the geological structure or to the geodynamic process. For example, they should explain why a fault which penetrates in the lithosphere produces a density anomaly. The reason is not necessarily obvious for the reader. 2) I am not convinced that the resolution of the inverted data allows an interpretation as precise (see comments about the resolution of the data after correction).
- Eastern NCC.
  Paragraph 1. "Obvious features […] obvious spatial distributions". It is not so obvious. More explanations are necessary here.
  Paragraph 2. "is consistent with" the authors should explain in what this result is consistent with the earthquake belt.
  Paragraph 3. "the fault belt may have penetrated the lithosphere". The authors should explain why they can make this interpretation. Which is the relation between the fault and the density distribution?
  Paragraph 4. "There are no significant negative density anomalies". I think that the authors mean "There are significant negative density anomalies".
  For this paragraph, the authors should also relate their interpretation to the density distribution described and explain this relation.
- Central NCC.
  Paragraph 1. "N1 is connected to the high-density anomaly area…'I do not see this connection. The author should better explain. For now, the description is too fuzzy.
  "N3 is connected to the high density" In the same way, I do not see that. In this depths range, the sign of the density anomaly changes.
  Paragraph 2. "alternating high and low density". What the reader must understand from this description? What does it mean in term of geological structures?
  I do not know where is N4. N4 is not located on the figure 15.
  Paragraph 3. The amplitude of the density anomaly N5 is compatible with the thermal data in the region? Please quantify.
  Paragraph 4. Again, I do not know where is N6. N6 is not located on the figure 15.

"crustal part" and "mantle part". The authors should use the depth of their maps in the figure 15. It will be much clearer.

"the thinning of the eastern lithosphere" Again, what it the relation between this interpretation and the concerned density anomaly? The mantle is it normal or hotter in this area?

- Western NCC.

Paragraph 2. "the low-density anomaly area at depths of 60-100 km is connected to the negative density anomaly". Are they really connected or are they two juxtaposed independent anomalies? What is the consequence on the interpretation?

Paragraph 4 (end). The amplitude of the low-density anomaly is consistent with the thermal data in the region? Please quantify.

Technical corrections

- Figure 15. Please add HFS, RMF and SP on the maps.
- Figure 11b) replace sedimentary layer interface by Moho layer interface.

---

## Editor Comment (EC1) · Mioara Mandea (Editor) · 19 Feb 2020

I acknowledge the important work realized by the authors. Based on my own reading of the manuscript and my experience in the field, I found that this contribution is of interest to the journal and may be suitable for publication. However, important revisions are required before its publication.

The authors need to consider the very constructive suggestions provided by both reviewers, and the manuscript has to be revised accordingly. In particular, the authors

need to strengthen the discussions about the data quality and robustness of the results, by quantifying the impact of applied corrections on data. More details are needed on the used methodology, and on provided information (e.g. Tables 1 and 2). An important issue is to appropriately address the interpretation of results in terms of geodynamical structure of North China Craton. It is important to provide a point by point response to the reviewers' comments and a new manuscript with marked changes.
* * *

---

## Author Comment (AC1) · 20 Mar 2020

The comment was uploaded in the form of a supplement:
https://www.solid-earth-discuss.net/se-2019-181/se-2019-181-AC1-supplement.pdf

---

## Author Comment (AC2) · 20 Mar 2020

The comment was uploaded in the form of a supplement:
https://www.solid-earth-discuss.net/se-2019-181/se-2019-181-AC2-supplement.pdf

---

## Author Comment (AC3) · 20 Mar 2020

I acknowledge the important work realized by the authors. Based on my own reading of the manuscript and my experience in the field, I found that this contribution is of interest to the journal and may be suitable for publication. However, important revisions are required before its publication. The authors need to consider the very constructive suggestions provided by both

**5 reviewers, and the manuscript has to be revised accordingly**

10

**Response:** Thanks a lot for your affirmation. This is a very important paper for us that we really take much effort on it. We discussed and considered a lot when we did every step in this project. We take all the valuable comments by topical editor and reviewers very seriously. After the carefully and detailed revision, we believe that the paper has the remarkable progress in quality. And we really want to show

our gratitude to you with the real name in the acknowledgement, we have added this part in acknowledgement in the revised version.

**1. In particular, the authors need to strengthen the discussions about the data quality and 15 robustness of the results, by quantifying the impact of applied corrections on data.**

**Response:** This is very good questions we also considered when we did the gravity gradient data processing. We made three revisions in the paper for this comments.

- (1) The amplitude of the topography effects and the underground interface undulation effects is obvious,
   but topographic correction and underground interface undulation corrections balance each other out to a certain degree (Szwillus et al., 2016). In consideration of the gravity gradient effect caused by the far zones (the extended area outside the study area). We extended geographic area within E90°-E130° and N23°-N53° as the calculation area for corrections (Figure 13 e-f). And the obtained results in the study area are cut out from the calculation area. We have added the gravity gradient effects
   (Topographic and Interface undulation) with extended area in Fig. 13(e-f). The effect of the
- counteraction can be more clearly seen in the extended area.

(2) We have added gravity gradient corrections  $T_{zz}$  at two cross sections (Latitude 35° and Latitude 37.5°) in Fig. 13. The effect of the counteraction can be also observed at these two cross sections.

(3) We have added a table about the statistics for each gravity gradient effects in Table 3.

30

35

**Revision: (Page 24, Paragraph 2)**

[revised manuscript text omitted]

**2. More details are needed on the used methodology, and on provided information (e.g. Tables 1 and 2).**

60 **Response:** (1) We have added the section "2.1 Kernel function calculation" in the section "2 Methodology". (Page 5)

(2) We have added the real densities of the true model, and RMS in both Table 1 and Table 2 as followings.

65 Revision: (Page 10)

Table 1 Conditions and results for initial density model one.

|        | True density
at each layer
(g/cm 3 )
(1 st to 4 th layer) | Initial density
at each layer
(g/cm 3 )
(1 st to 4 th layer) | Iterations
(k) | RMS
(E)             | Maximum density
at each layer
(g/cm 3 )
(1 st to 4 th layer) |
|--------|-----------------------------------------------------------------------------------------------------|--------------------------------------------------------------------------------------------------------|-------------------|------------------------|--------------------------------------------------------------------------------------------------------|
| Test 1 | (0.4, 0.7, 0.7, 1.0)                                                                                | (0, 0, 0, 0)                                                                                           | 10                | 7.35×10 -11 | (0.34, 0.42, 0.53, 0.55)                                                                               |
| Test 2 | (0.4, 0.7, 0.7, 1.0)                                                                                | (0.4, 0.4, 0.4, 0.4)                                                                                   | 7                 | 6.94×10 -12 | (0.51, 0.55, 0.66, 0.58)                                                                               |
| Test 3 | (0.4, 0.7, 0.7, 1.0)                                                                                | (0.1, 0.2, 0.2, 0.3)                                                                                   | 6                 | 3.50×10 -12 | (0.36, 0.64, 0.67, 0.64)                                                                               |
| Test 4 | (0.4, 0.7, 0.7, 1.0)                                                                                | (0.2, 0.4, 0.4, 0.7)                                                                                   | 3                 | 2.31×10 -13 | (0.38, 0.72, 0.76, 0.95)                                                                               |
|        | Minimum density                                                                                     | Average density                                                                                        |                   |                        |                                                                                                        |
|        | at each layer                                                                                       | at each layer                                                                                          |                   |                        |                                                                                                        |
|        | $(g/cm^3)$                                                                                          | $(g/cm^3)$                                                                                             |                   |                        |                                                                                                        |
|        | $(1^{st} to 4^{th} layer)$                                                                          | $(1^{st} to 4^{th} layer)$                                                                             |                   |                        |                                                                                                        |
| Test 1 | (0.23, 0.27, 0.34, 0.29)                                                                            | (0.31, 0.34, 0.41, 0.35)                                                                               |                   |                        |                                                                                                        |
| Test 2 | (0.42, 0.43, 0.51, 0.48)                                                                            | (0.47, 0.50, 0.58, 0.55)                                                                               |                   |                        |                                                                                                        |
| Test 3 | (0.32, 0.46, 0.33, 0.55)                                                                            | (0.34, 0.55, 0.59, 0.58)                                                                               |                   |                        |                                                                                                        |
| Test 4 | (0.33, 0.52, 0.57, 0.83)                                                                            | (0.35, 0.59, 0.65, 0.91)                                                                               |                   |                        |                                                                                                        |

**70 (Page 12)**

Table 2 Conditions and results for initial density model two.

|        | True density
(Trapezoid,
Cuboid)
(g/cm 3 ) | Initial density
(Trapezoid,
Cuboid)
(g/cm 3 ) | Iterations
(k) | RMS
(E)             | Maximum density
(g/cm 3 ) | Minimum density
(g/cm 3 ) | Average density
(g/cm 3 ) |
|--------|----------------------------------------------------------------|-------------------------------------------------------------------|-------------------|------------------------|-----------------------------------------|-----------------------------------------|-----------------------------------------|
| Test 5 | (0.8, 1.0)                                                     | (0, 0)                                                            | 8                 | 4.37×10 -12 | 0.6887                                  | 0.1328                                  | 0.4225                                  |
| Test 6 | (0.8, 1.0)                                                     | (0.2, 0.2)                                                        | 7                 | 5.63×10 -13 | 0.7121                                  | 0.1856                                  | 0.4886                                  |
| Test 7 | (0.8, 1.0)                                                     | (0.5, 0.5)                                                        | 5                 | 1.06×10 -13 | 0.7556                                  | 0.4762                                  | 0.6256                                  |
| Test 8 | (0.8, 1.0)                                                     | (-0.2, -0.2)                                                      | 13                | 8.49×10 -11 | 0.6399                                  | 0.0399                                  | 0.3276                                  |

**3. An important issue is to appropriately address the interpretation of results in terms of geodynamical structure of North China Craton.**

75

80

**Response:** As we have put forward the fundamental questions at the beginning of the abstract, the destruction mechanism and geodynamics of the NCC are the fundamental questions, we have added the section "5.4 The destruction mechanism of NCC" at the end of the section "5 Discussion" as followings. And we also interpreted results in terms of geodynamical structure in every section 5.1, 5.2 and 5.3, which in concert with the final discussion section 5.4.

**Revision: (Page 34)**

**5.4 The destruction mechanism of NCC**

- 85 The low-density anomalies in the Qilian block and northern Taihang Orogenic belt are affected by the high heat flux environments. Low-density anomalies in the central Taihang Orogenic belt exist but are not accompanied by continuous high heat flux environments at mantle depth. However, without mantle plumes, the low-density anomalies in Bohai Bay are affected by the extension of the Tancheng-Lujiang fault belt, and mechanical extension destroys the lithosphere in this area. Based on our studies, one 90 theory is that it is hard to explain the destruction phenomena and modes in the whole NCC. The destruction of NCC is not only affected by physical tension but also caused by thermal erosion and delamination. Previous dynamic studies have shown that (Zhu et al., 2012; Zhu, 2018) since the
- Mesozoic, the Pacific Plate subducted westward to the Taihang Orogenic belt in the central part of the NCC. The residual dehydration of the subducted plate in the mantle transition zone promoted an increase in the molten fluid content in the upper mantle beneath the NCC. The delamination and thermal erosion of the lithosphere in the NCC reflect different forms of mantle convection instability. Therefore, through this study, it is believed that the destruction in the NCC is caused by several forms. Several destruction modes of the NCC coexist in different geological structural backgrounds.

**100 4. It is important to provide a point by point response to the reviewers' comments and a new manuscript with marked changes.**

**Response:** We have provided a point by point response to the reviewers' comments, we also attached the new manuscript with marked changes in red after the response.

105

**Sequential inversion of GOCE satellite gravity gradient data and terrestrial gravity data for the lithospheric density structure in the North China Craton**

Yu Tian1,2,3, Yong Wang2,3

- 1Ocean College, Minjiang University, 350108 Fuzhou, China
   2State Key Laboratory of Geodesy and Earth's Dynamics, Institute of Geodesy and Geophysics, Chinese Academy of Sciences, 430077 Wuhan, China
   3University of Chinese Academy of Sciences, 100049 Beijing, China
   *Correspondence to*: Yu Tian (tybgys455429145@163.com)
- 120 Abstract. The North China Craton (NCC) is one of the oldest cratons in the world. Currently, the destruction mechanism and geodynamics of the NCC remain controversial. All of the proposed views regarding the issues involve studying the internal density structure of the NCC lithosphere. Gravity field data are among the most important data in regard to investigating the lithospheric density structure, and gravity gradient data and gravity data each possess their own advantages.

Given the different observational plane heights between the on-orbit GOCE satellite gravity gradient and terrestrial

- 125 gravity and the effects of the initial density model on the inversion results, sequential inversion of the gravity gradient and gravity are divided into two integrated processes. By using the preconditioned conjugate gradient (PCG) inversion algorithm, the density data are calculated using the preprocessed corrected gravity anomaly data. Then, the newly obtained high-resolution density data are used as the initial density model, which can serve as constraints for the subsequent gravity gradient inversion. Several essential corrections are applied to the four gravity gradient tensors ( $T_{xx}$ ,  $T_{yy}$ ,  $T_{zz}$ ) o

---

## Author Response (AR1)

**Before I respond all the comments, I want to show my gratitude to the reviewers. It is my honor to get this constructive comments and suggestions. The reviewer not only pointed out the shortcoming of the paper, but also provided suggestions for revisions, and even recommended several excellent papers to me. I really learned a lot from this comment with comprehensive thinking, precise logic, and excellent formatting. This is a very important paper for us , we really take every piece of valuable comment very seriously, we adopted all the constructive suggestions and responded all the comments as followings. In addition, I will keep this document all the time to inspire myself in my whole academic career. And we really want to show our gratitude to reviewers with the real name in the acknowledgement , please contact us if it is possible.**

Yu Tian (On behalf of all authors)

Email:tybgys455429145@163.com

**General Comments**

**• **Terminology**

**1. The paper denotes the gravity anomaly/gradient data sets that were reduced for multiple effects as "remaining" gravity anomaly/gradient. Using a term that is commonly used in gravimetry would be more appropriate (e.g., "corrected" for the effect of " etc.)**

**Response:** We have used the term "corrected" to replace the term "remaining" throughout the whole paper.

**2. Throughout the paper: "resultant data" → "resulting data"?**

**Response:** We have used the term "resulting" to replace the term "resultant" throughout the whole paper.

**3.** Based on the paper that two referees recommended,we have used the term "terrestrial" to replace the term "surface" ; based on the inversion classification, we have used the term "sequential inversion" to replace the term "joint inversion" ; we have changed the title to " Sequential inversion of GOCE satellite gravity gradient data and terrestrial gravity data for the lithospheric density structure in the North China Craton".

**• Formatting**

**In the text, white spacing is often missing when Figures are mentioned, look for "...(Fig...)"**

40  **Response:** We have already checked the white spacing between the bracket "( )" and the term "Fig"; and the white spacing between the term "Fig" and the following numbers.

**• Figures**

45  **4. Figures - on the top of the colorbars is a slash followed with a unit (probably coming from the panels with densities). Can it be removed (it is confusing)?**

**Response**: We have removed the unit on the top of colorbars and presented the unit (elevation, depth, gravity gradient, density) along with the colorbars as followings. We have updated all the figures
50  throughout the paper.

[Figure]

[Figure]

55   5. **IMPORTANT**: Figures 2,3 have different colorbars for the input and the output. Please, update them to have the same colorbar span.

**Response: (Page 11 and Page 12)**
We have updated five figures in Figures 2 with the same colorbar ($0.2g/cm^3$-$1.0g/cm^3$), we have updated
60   five figures in Figures 3 with the same colorbar ($0.1g/cm^3$-$1.0g/cm^3$) in the paper .

***Detailed comments*** (**We have carefully followed all the comments on words, sentences, grammar, we have marked in red throughout the revised submission.** The order is **important questions** throughout the paper **first**, **then** the **detailed questions**.)

**Important questions**

•1. (1) **167 Reformulate the sentence and see Martinec, Z. (2014). Mass-density Green's functions for the gravitational gradient tensor at different heights. Geophysical Journal International,**
70   **196(3), 1455-1465. ... not all gradients have the kernel with the properties as you describe (what you describe is related to the radial derivatives)**
(2) **176 IMPORTANT. You use some damping function to revert the kernel behaviour. Why do you use the same for all the gradients if their kernels behave differently?**
**Response:**
75       This is a very important part that we did not describe in detail. The kernel functions are calculated under the Cartesian coordinate instead of the spherical coordinate, therefore we added the whole section " 2.1 Kernel function calculation" in Section 2. We have provided the detailed integral and analytical equations, which have the properties as we described in the depth weighting function. And for better understanding,we added the description about the kernel function after the Equation (6). In fact,

we used the different kernel functions corresponding to each gravity gradient components, we have provided the detailed description in the revised version.

The recommended paper (**Martinec Z., 2014**) presents the kernel function calculated in spherical coordinate, this is a very good paper which describes the advantage and disadvantage of the gravitational gradient tensor at different heights,we have cited this paper in the section ”3.2 Downward Continuation” .

**Revision:(Page 5)**
1. We have added section ” 2.1 Kernel function calculation” .

**(Page 6)**
2. The description about the kernel function after the Equation (6) when we first mentioned the Kernel function in ” Section 2.2 Inversion method ”.

”for gravity data, $\boldsymbol{G} = \boldsymbol{G}_z$ , while for gravity gradient components, $\boldsymbol{G} = [\boldsymbol{G}_{xx}, \boldsymbol{G}_{xz}, \boldsymbol{G}_{yy}, \boldsymbol{G}_{zz}]^{\mathrm{T}}$ ”.

● 2. (1) **Table 1 IMPORTANT. Could you add the real densities used in the model. The table gives just the initial values but not true ones so the reader cannot compare the result with the truth.**

**(2) 266 You claim ”that is almost identical to the true model”. If I have not overlooked something, I did not find the true values so the claim cannot be checked by the reader. As said above, updating Table 1 with true values would help. Note ”almost identical” is soft characteristics - it is more convenient to use RMS or so.**

**Response:** We have added the real densities of the true model, and RMS in both Table 1 and Table 2 as following.

**Revision:  (Page 10)**

**Table 1** Conditions and results for initial density model one.

| | True density at each layer $(g/cm^3)$ (1st to 4th layer) | Initial density at each layer $(g/cm^3)$ (1st to 4th layer) | Iterations (k) | RMS (E) | Maximum density at each layer $(g/cm^3)$ (1st to 4th layer) |
|---|---|---|---|---|---|
| Test 1 | (0.4, 0.7, 0.7, 1.0) | (0, 0, 0, 0) | 10 | $7.35 \times 10^{-11}$ | (0.34, 0.42, 0.53, 0.55) |
| Test 2 | (0.4, 0.7, 0.7, 1.0) | (0.4, 0.4, 0.4, 0.4) | 7 | $6.94 \times 10^{-12}$ | (0.51, 0.55, 0.66, 0.58) |
| Test 3 | (0.4, 0.7, 0.7, 1.0) | (0.1, 0.2, 0.2, 0.3) | 6 | $3.50 \times 10^{-12}$ | (0.36, 0.64, 0.67, 0.64) |
| Test 4 | (0.4, 0.7, 0.7, 1.0) | (0.2, 0.4, 0.4, 0.7) | 3 | $2.31 \times 10^{-13}$ | (0.38, 0.72, 0.76, 0.95) |

| | Minimum density at each layer (g/cm$^3$) ($1^{st}$ to $4^{th}$ layer) | Average density at each layer (g/cm$^3$) ($1^{st}$ to $4^{th}$ layer) |
|---|---|---|
| Test 1 | (0.23, 0.27, 0.34, 0.29) | (0.31, 0.34, 0.41, 0.35) |
| Test 2 | (0.42, 0.43, 0.51, 0.48) | (0.47, 0.50, 0.58, 0.55) |
| Test 3 | (0.32, 0.46, 0.33, 0.55) | (0.34, 0.55, 0.59, 0.58) |
| Test 4 | (0.33, 0.52, 0.57, 0.83) | (0.35, 0.59, 0.65, 0.91) |

**(Page 12)**

**Table 2** Conditions and results for initial density model two.

| | True density (Trapezoid, Cuboid) (g/cm$^3$) | Initial density (Trapezoid, Cuboid) (g/cm$^3$) | Iterations (k) | RMS (E) | Maximum density (g/cm$^3$) | Minimum density (g/cm$^3$) | Average density (g/cm$^3$) |
|---|---|---|---|---|---|---|---|
| Test 5 | (0.8, 1.0) | (0, 0) | 8 | $4.37\times10^{-12}$ | 0.6887 | 0.1328 | 0.4225 |
| Test 6 | (0.8, 1.0) | (0.2, 0.2) | 7 | $5.63\times10^{-13}$ | 0.7121 | 0.1856 | 0.4886 |
| Test 7 | (0.8, 1.0) | (0.5, 0.5) | 5 | $1.06\times10^{-13}$ | 0.7556 | 0.4762 | 0.6256 |
| Test 8 | (0.8, 1.0) | (-0.2, -0.2) | 13 | $8.49\times10^{-11}$ | 0.6399 | 0.0399 | 0.3276 |

• 3. (1) **359 IMPORTANT"of the European Space Agency". The webpage is not maintained by ESA and ESA is not responsible for that.**
**(2) 619 at the given website goce.mka.zcu.cz data cannot be browsed (how?), just it can be downloaded? Please change the sentence if true.**

**Response:** I really agree with the comments. The pages are web pages for ESA supported project **Towards a better understanding of the Earth's interior and geophysical exporation research "GOCE+ Geoexplore II"**, instead of ESA website. We have deleted the description and provided the data website http://goce.kma.zcu.cz/data.php, which we have checked can be browsed.

**Revision: (Page 16 Paragraph 1)**
1. This study directly downloaded the preprocessed gravity gradient anomaly data with the spatial resolution of 10 arc-min, which was acquired over 48 months from November 2009 to October 2013 (Sebera et al., 2014), from the website GOCE+ Geoexplore II (http://http://goce.kma.zcu.cz/data.php).

**(Page 35)**
2. The on-orbit GOCE gravity gradient data can be browsed and downloaded from GOCE+ Geoexplore II (http://goce.kma.zcu.cz/data.php).

• 4. **Section 3.6 - the text implies that this correction is due to the masses below the area of interest. This is was recently studied by Szwillus et al. (2016) showing the importance of lateral masses. Please mention this problem.**

135

**Response:** Firstly, thanks a lot for recommending this important paper, which help us to know the exact extension criteria. We really have considered the long wavelength gravity gradient effect caused by the far zones, although we did not confirm the extended area (with 10° extension in all directions for far zone distant effects) is enough when we did this. The recommended paper points out that " For satellite-based gravity gradients, to reduce the remaining rms distant effect to 10% of the global rms, a radius of 10° is sufficient. And for the regional test, the RMS is much more smaller than the global average." Therefore , we confirm that 10° extension for far zone distant gravity gradient effects is enough for our study. We adopted the same method as the paper, the extended geographic area within E90°–E130° and N23°–N53° is defined as the calculation area for corrections (Figure 13 e-f). And the obtained results in the study area are cut out from the calculation area.

We made three revisions according to this problem. In revision 1, we described about the far zone distant gravity gradient effects before we did all the corrections. In revision 2, we added the two figures with extended area in Section 3.7, and these two figures (Topographic effects and underground interface undulation effects) also demonstrate that " Topographic and isostatic balance each other out to a certain degree" with the reference (Szwillus et al., 2016). In revision 3, for better understanding, we have changed the section title " long wavelength correction" to " Correction for the gravity gradient effects of the mantle under 180km".

**Revision**: **(Page 18 Paragraph 1)**

155 1. The gravity gradient anomaly components after downward continuation are a combination of the interface undulation and density heterogeneity. Thus, the topographic correction, underground interface undulation correction and long wavelength correction should be carried out in accordance with the existing precise models. These three corrections are affected by the masses within and outside the study area, the far-field correction outside the study area should be taken into consideration (Szwillus et al.,

160 2016). Therefore, the correction radius in the study area is extended by 10° for all corrections. The same geographic area within E90°–E130° and N23°–N53° is defined as the calculation area for corrections (Figure 13 e-f). And the obtained results in the study area are cut out from the calculation area.

165

2.  **(Page 25)**

[Figure]

170  **Figure 13** Gravity gradient effects with extended area (e) Topographic effects, (f) Interface undulation (Sedimentary and Moho) effects.

**3. (Page 23)**
We changed the section title from " Long wavelength correction" to " Correction for the gravity
175  gradient effects of the mantle under 180km ".

**Detailed questions**

• **14** *Inconsistency* **- do you mean a difference in the data representation (geometry)?**
180

**Response:** We wanted to emphasize the different height between the observation planes, we have revised the sentence as following.

**Revision: (Page 24 Paragraph 2)**
185  Given the different bservation plane height between the on orbit GOCE satellite gravity gradient and terrestrial gravity, ... .

• **19** *performed* → **applied to?**

190  **Revision: (Page 1 Paragraph 2)**

Several essential corrections are applied to the four gravity gradient tensor ... .

• **20** *observation quantity* → **observations?**

195 **Revision: (Page 1 Paragraph 2)**
... after which the corrected gravity gradient anomaly ($T'_{xx}$, $T'_{xz}$, $T'_{yy}$, $T'_{zz}$) are used as the observations.

• **23-24** *high heat flux* **... could add a reference to this sentence?**

200 **Response:** We have added the reference to this sentence. We also added the temperature difference at 140km and 180km in Figure 15 (i, k) for better understanding.

**Revision: (Page 1 Paragraph 3)**
While in the mantle, the presented obvious low density areas are mainly affected by the high heat flux
205 environment (An and Shi, 2007).

•**(1) 36-38 please reformulate or leave this sentence "***The most direct and effective approach...***" because the following one repeats it in other words. Btw, the non-uniqueness of the gravity-based inversions make the gravity data not the easiest means for understanding the subsurface. • (2) 38**
210 ***variations* - this little implies time variations. Please, reformulate. • (3) 38 "***laws***"? Gravimetry does not study the laws behind measurements but it uses the laws to study phenomenas in the region of interest. Please reformulate**

**Response:** The original version is too cucumber, we have deleted the first sentence "*The most direct*
215 *and effective approach...*", and shorten other two sentences as following.

**Revison: (Page 2 Paragraph 1)**
The gravity field data plays an important role in determining and interpreting the lithospheric density structure and state of motion.
220

• **49 Note ... gravity data contains all the frequencies about the whole Earth, but the high frequencies of the deep structures are just strongly attenuated due to the distance. If we would substantially improve S/N of the gravimeters, we could infer hi-freqs from the very deep density contrasts - the signal is there, just masked out by the signal of the closer masses. The point is**
225 **however that the data contains it all but due to the attenuation and the errors we do not have access to it.**

**Response:** Very constructive comments. We have reformulated the original sentences as following.

**Revison: (Page 2 Paragraph 1)**

In the frequency domain, the high frequencies of the deep structures are strongly attenuated due to the distance, and masked out by the signal of the closer masses, the gravity data can be mainly used to provide mid low frequency information of the deep structure.

• **60** *"rare"*? **(please reformulate), search for the topic, e.g.https://scholar.google.cz/scholar?hl=cs&as_sdt=0%2C5&q=joint+gravity+and+gradient+inversion& btnG=**

**Response:** We wanted to emphasize the inversion realized in two steps, as we have changed the title from the *"joint"* to *"sequential"*, we revised sentences as following.

**Revison: (Page 2 Paragraph 2)**

Currently, most researches based on the joint inversion of the gravity and gravity gradient data instead of the sequential inversion.

• **(1) 120 the sentence "Since" does not make sense, please update**
• **(2) 121** *undetermined* **is related to a number of data but you rather want to emphasize that the problem is ill-conditioned right?**

**Response:** We revised the sentence as following, we emphasize the non-unique problem in the first sentence, and the ill-conditioned problem in the second sentence.

**Revision: (Page 6)**

The quantity of the unknowns m greatly exceeds the acquired data vector, the solutions of the equations are non-unique. Moreover, the inversion is an ill-conditioned problem, the appropriate constraints upon the objective function are required to narrow the range of solutions.

• **143 "is comprehensive" not clear what you mean (please reformulate)**

**Response:** The word "comprehensive" is misleading, we used the word "complex" to replace comprehensive.

**Revision: (Page 7 Paragraph 1)**
265 "calculation of the Lagrangian multiplier is complex."

**• 146 ?"...while a very small value may lead to a large ...the model and the data, thus providing unrealistic ..."??**

270 **Response:** The sentence is misleading, we changed the sentences as following.

**Revision: (Page 7 Paragraph 1)**
Since the regularization parameter serves to balance the data fitting function and the model fitting function, an excessively large value will result in substantial differences between the inversed results
275 response and the observation, while an overwhelmingly small value leads to the ineffectiveness of the model fitting function.

**• 147 introduce the L-curve method before you write it is used**

280 **Response:** We have reformulated the paragraph as following when we first introduced the L-curve.

**Revision: (Page 7 Paragraph 2)**
Given these problems, the L-curve method was developed for the selection of regularization parameters in the solution of ill-posed problems (Hansen, 1992). The L curve is a criterion that is based on a
285 comparison between the actual data fitting function and the model objective function, which is applicable to solving problems with large scales. The value corresponding to the inflection point of the L curve is assigned to the regularization parameter. The effectiveness of this method has been validated in previous studies (Tian et al., 2018; 2019).

290 **• 278 ".... a more precise initial model" and/or more data, right?**

**Response:** Right, we did not input the initial model in Figure 2b, but we input the different initial models in Figure 2c-2e. We have revised sentences as following.

295 **Revison: (Page 11 Paragraph 1)**
" ... With the help of a more precise initial model and more data, Fig. 3b–d demonstrate that ...".

**• 300 ...*decomposed into two integrated ...* → is composed of two steps?**

300 **Revison: (Page 12)**

The sequential inversion is realized in two steps, namely, the gravity inversion and the gravity gradient inversion.

**• (1) 335 What is meant by "outer" iterations. Are there any inner iterations so that these two**
**have to distinguished in the paper ? (2) 455: "outer iterations"?    (3) 202-228 Consider leaving the**
**whole paragraph on PCG since this method is well described and established in the community.**
**The paper is already quite long, here is the potential to shorten it.**

**Response:** The PCG algorithm consists of inner iteration and outer iteration. However, as we have left the whole paragraph on PCG algorithm and just provided the reference, we will not discriminate inner iteration and outer iteration in the revised version. We have used the term "iteration" to replace "outer iteration" throughout the paper.

**• Figure 5: the red line is hardly visible, maybe not needed in this figure.**

**Response:** We have updated both Figure 5 (b) and Figure 14, and deleted all the red line as following.

**Revision: (Page 14)**

[Figure]

**• Figure 6: density distribution variation (it is not a full value right?)**

**Response:** We have revised the Figure 6 caption as following.

**Revison: (Page 16)**

**Figure 6** Lithospheric density distribution of NCC by gravity at different depths.

**• Figure 9: update the caption, it is not clear what the panel (e) is showing (a 3D**

**map or Tzz as written in the title?)**

330   **Response:** In Figure 9、Figure 10 and Figure 11,as most figures are 2D, we want to try some new attempts for showing figures in 3D form intuitively for better understanding. We want to show the topography/interface undulation and gravity gradient effects in one figure. The undulation of the figure represents topography or interface undulation , and the color reflects the corresponding gravity gradient effects $T_{zz}$. We have updated the figure with unit as following.

335   **Revison: (Page 19)**

[Figure]

**Figure 9 (e)** The 3D gravity gradient effect $T_{zz}$ of topography.

•   **Figure 10 the same for (g), update unit in the colorbar of this panel (g). Shorten the caption as**
340   **the caption repeats all the same (you can put "anomalous gravity gradients Txx, Tyy ...etc" into one sentence)**

**Response:** We have checked all the figure captions. We have deleted the number in the panel and shortened all the captions, especially when we did not describe figures in detail. We have shortened the
345   caption as following.

**Revison: (Page 21)**
Figure 10 (**a**) Sedimentary layer interface of the NCC. (**b**) Sedimentary layer interface undulation of the NCC. The anomalous gravity gradient component caused by the relief at the sedimentary layer interface undulation, (**c**) $T_{xx}$ , (**d**) $T_{xz}$ , (**e**) $T_{yy}$ , (**f**) $T_{zz}$ . (**g**) The 3D gravity gradient effect $T_{zz}$ of sedimentary
350   layer interface undulation.

•   **421: You probably mean the gravity effect of the layer bounded by topo and Moho, right? The**
**sentence implies to calculate the gravity effect of the Moho surface.**

355

**Response: Right.** We calculated the gravity gradient induced by Moho undulation instead of Moho surface. We have revised the paper as following.

**Revison: (Page 21)**

360 The Moho undulation after correcting for the average depth is illustrated in Fig. 11b. Similarly, the gravity gradient effects induced by the Moho undulation can be calculated.

**• 442: what is meant by "*relatively minor*"? - the magnitudes are up to 1.5 E, which quite large**

365 **Response:** Right. We have deleted this sentence, and we have provided the statistics for all the corrections in Figure 13 (g-h) and Table 4, we really cannot ignore this correction based on the value.

**• Figure 12: remove "(a)...(d)" ...clear from the panels**

370 **Response:** We have updated all the figures in the paper, we only numbered the panel when we discussed some figures.

**• 445: not clear what the sentence tries to tell (what official website, why this is linked with the accuracy/reliability of the data?)**

375

**Response:** We wanted to emphasize the spatial resolution of several models we used are different. We have revised paper as following.

**Revision: (Page 24)**

380 *"Since models with different spatial resolutions are used in several corrections, the spatial resolutions of the obtained results are different."*

**• Figure 14: unit is missing in the y-axis**

385 **Response:** We have updated both Figure 5 (b) and Figure 14 with the unit "RMS ($10^{-1}$E)".

**Revision: (Page 14 and Page 27)**

[Figure]

390    **Figure 14** With the iterative calculation, residual mean square between the forward calculated theoretical gravity gradient and the gravity gradient measurements versus iteration number in the PCG inversion algorithm.

• **470: "*negative density anomalies*" → negative density contrast/residuals?**

395    **Response:**

When we mentioned the specific value, we revised all the "*negative density anomalies*" → "negative density residuals", and "*positive density anomalies*" → "positive density residuals".

400

405

410

**Before I respond all the constructive and valuable suggestions and comments,I want to show our gratitude to reviewers. It is my honor in my academic career to get this inspiring and leading comments, which really carry heavy weight for us to improve this manuscript. This is a very important paper for us that we really take much effort on it. We discussed and considered a lot when we did every step in this project. We really take every piece of suggestion or comment very seriously. This is the longest response with 24 pages that I have ever written, but it really deserves. And we really want to show our gratitude to the reviewer with the real name in the acknowledgement , please contact us if it is possible. We will be very glad about that.**

Yu Tian (On behalf of all authors)
Email:tybgys455429145@163.com

1. **One of the major issue of the paper is that the authors never discuss and quantify the impact of each correction applied on the data before the inversion on the resulting density models in terms of resolution and amplitude.**

**Response:** This is very good questions we also considered when we did the gravity gradient data processing. We made three revisions in the paper for this comments.

(1) The amplitude of the topography effects and the underground interface undulation effects is obvious, but topographic correction and underground interface undulation corrections balance each other out to a certain degree (Szwillus et al., 2016). In consideration of the gravity gradient effect caused by the far zones (the extended area outside the study area).  We extended geographic area within E90°–E130° and N23°–N53° as the calculation area for corrections (Figure 13 e-f). And the obtained results in the study area are cut out from the calculation area. We have added the gravity gradient effects (Topographic and Interface undulation) with extended area in Fig. 13(e-f). The effect of the counteraction can be more clearly seen in the extended area.

(2) We have added gravity gradient corrections $T_{zz}$ at two cross sections (Latitude 35° and Latitude 37.5°) in Fig. 13. The effect of the counteraction can be also observed at these two cross sections.

(3) We have added a table about the statistics for each gravity gradient effects in Table 3.

**Revision: (Page 24, Paragraph 2)**

The amplitude of the topography effects and the underground interface undulation effects is obvious, but the topographic correction and underground interface undulation correction balance each other to a

certain degree (Szwillus et al., 2016). This counteraction is obvious in the extended study area, as shown in Fig. 13e (topographic correction) and Fig. 13f (underground interface undulation correction). The detailed statistical amplitude for each gravity gradient correction are summarized in Table 3. To present the different corrections $T_{zz}$ at different cross sections of latitude 35° and latitude 37.5° are presented in Fig.13 (g-h).

[Figure]

**Figure 13** Gravity gradient effects with extended area (e) Topographic effects $T_{zz}$, (f) Interface undulation $T_{zz}$ (Sedimentary and Moho) effects. Several corrections $T_{zz}$ at different cross sections (g) Latitude 35°, (h) Latitude 37.5°.

450

**Table 3** Statistics for each gravity gradient correction.

| Gravity Gradient Effects (E) | Downward Continuation Maximum (E) | Downward Continuation Minimum (E) | Topographic Effects Maximum (E) | Topographic Effects Minimum (E) | Interface Undulation Effects Maximum (E) | Interface Undulation Effects Minimum (E) |
|---|---|---|---|---|---|---|
| $T_{xx}$ | 5.3 | -5.7 | 6.3 | -9.4 | 10.1 | -4.9 |
| $T_{xz}$ | 7.2 | -5.6 | 11.9 | -5.6 | 5.2 | -8.9 |

| | | | | | |
|---|---|---|---|---|---|
| $T_{yy}$ | 4.8 | -3.6 | 7.2 | -6.8 | 6.3 | -6.7 |
| $T_{zz}$ | 7.5 | -6.5 | 13.8 | -6.3 | 7.7 | -14.1 |

| Gravity Gradient Effects (E) | Long Wavelength Effects Maximum (E) | Long Wavelength Effects Minimum (E) | Corrected Effects Maximum (E) | Corrected Effects Minimum (E) |
|---|---|---|---|---|
| $T_{xx}$ | 1.1 | -0.8 | 8.7 | -6.8 |
| $T_{xz}$ | 1.2 | -0.7 | 8.5 | -4.7 |
| $T_{yy}$ | 1.4 | -0.7 | 6.7 | -6.6 |
| $T_{zz}$ | 1.5 | -1.5 | 11.5 | -12.3 |

**2. Another major issue concerns the inversion methodology of the gravity gradients. In this paper, the authors focus on the 4-high accuracy GOCE satellite gravity gradient tensor components ($T_{xx}$, $T_{xz}$, $T_{yy}$, $T_{zz}$). The authors do not explain if their method inverts these components separately or simultaneously and do not discuss the contribution of each component in the inversion results.**

**Response:** (1) We inverted four GOCE satellite gravity gradient tensor components simultaneously, and we did not add the weight on the four GOCE satellite gravity gradient tensor components. We have added a whole section "2.1 Kernel function calculation".

(2) We have tested the contribution of each component in the inversion results by the model test in our previous paper (Tian et al., 2019) as followings, we compared all the independent components inversion results and the joint inversion results, we also compared the information contained in each components by forward calculation. As there is too much content in this paper and words limited by the journal, we added the corresponding reference in the paper.

References

Tian, Y., Ke, X., and Wang, Y.: Inversion of three-dimensional density structure using airborne gradiometry data in Kauring test site, Geomatics and Information Science of Wuhan University, 44 (4), 501-509, doi: 10.13203/j.whugis20160503, 2019b.

[Figure]

**Figure 2** (Tian et al. 2019) Inversion of independent component and joint inversion. (a) True model, (b) Txx, (c) Txz, (d) Tyy, (e) Tzz, (f) Joint inversion.

480

**Revision:** (1) (**Page 4-5**)

Please referred to section "2.1 Kernel function calculation".

(2) (**Page 8-9**)

Eq. (15) can be simplified as:

485    $A\Delta m = b$                                                          (16)

for the gravity data,we take the $T_z$ as the observation,$A = \left[ \ G_z \ , \sqrt{\mu}W_i \ \right]^{\mathrm{T}}$ and $b = \left[ T_z \ , 0 \right]^{\mathrm{T}}$, for the gravity gradient data, we selected four processed components $T'_{xx}$、$T'_{xz}$、$T'_{yy}$ and $T'_{zz}$ simultaneously as the observation,which implies the Jacobian matrix $A = \left[ G_{xx}, G_{xz}, G_{yy}, G_{zz}, \sqrt{\mu}W_i \right]^{\mathrm{T}}$ and $b = \left[ T'_{xx} \ , \ T'_{xz} \ , \ T'_{yy} \ , \ T'_{zz}, 0 \right]^{\mathrm{T}}$. The contribution of each component can be referred to our previous studies

490    (Tian et al., 2019).

**3. The last major issue regards the discussion of the results in terms of geological structures and geodynamical processes in the North China Craton. The authors should remind what is/are the fundamental question(s) in this region and in what this study brings answers or at least new constraints. This is not clearly specified in the current version of the paper.**

**Response:** As we have put forward the fundamental questions at the beginning of the abstract, the destruction mechanism and geodynamics of the NCC are the fundamental questions, we have added the section " 5.4 The destruction mechanism of NCC" at the end of the section" 5 Discussion"   as followings.

**Revision: (Page 34-35)**

**5.4   The destruction mechanism of NCC**

The low-density anomalies in the Qilian block and northern Taihang Orogenic belt are affected by the high heat flux environments. Low-density anomalies in the central Taihang Orogenic belt exist but are not accompanied by continuous high heat flux environments at mantle depth. However, without mantle plumes, the low-density anomalies in Bohai Bay are affected by the extension of the Tancheng-Lujiang fault belt, and mechanical extension destroys the lithosphere in this area. Based on our studies, one theory is that it is hard to explain the destruction phenomena and modes in the whole NCC. The destruction of NCC is not only affected by physical tension but also caused by thermal erosion and delamination. Previous dynamic studies have shown that (Zhu et al., 2012; Zhu, 2018) since the Mesozoic, the Pacific Plate subducted westward to the Taihang Orogenic belt in the central part of the NCC. The residual dehydration of the subducted plate in the mantle transition zone promoted an increase in the molten fluid content in the upper mantle beneath the NCC. The delamination and thermal erosion of the lithosphere in the NCC reflect different forms of mantle convection instability. Therefore, through this study, it is believed that the destruction in the NCC is caused by several forms. Several destruction modes of the NCC coexist in different geological structural backgrounds.

**4. Finally, the authors interpret some density anomalies as thermal variations in the mantle without any quantification. In a general way, in their interpretation, the authors should systematically and clearly relate the density anomaly to the geological structure or to the geodynamic process which is not the case in this current version.**

**Response:** (1) Based on the collected thermal data, we calculated the temperature difference at the depth of 140kn-180km and made Figures 15 (i-j), which are better to present the temperature level in different regions. The detailed quantification and interpretation can be referred to the corresponding "Response" in "Discussion".

(2) With every detailed piece of constructive comments in discussion, we really systematically and clearly relate the density anomaly to the geological structure or to the geodynamic process in this revised version.

**5. The authors choose to use gravity and gravity gradient data obtained from measurements and not derived from the gravity field models in order to preserve their high precision, which is indeed relevant. However, the applied corrections imply the use of models not well constrained as the CRUST 1.0 model which highly compromise the high quality of the data and thus the resolution of the inversion results.**

**Response:** This comment is a very important question that we have considered a long time when we did the data processing. In fact, we also collected other regional and global crustal models (Zheng et al., 2011; Shen et al., 2016; Abrehdary et al., 2017), and discussed this question with article authors. **Most models provide the higher spatial resolution of the depth at points,but without any information about the density, which are not sufficient for calculation. As we used the software "Tesseroids" to calculate interface undulation effects. We used the depth and real density at every point, instead of the depth and the empirical density.** Although the spatial resolution improved, but the empirical density value will also bring uncertainties. The model we used have to satisfy three conditions: (1) The model has to cover the whole calculation area. (2) The model has to provide the depth of Moho

and sediment. (3) The model has to provide the density of Moho and sediment. The other collected models only satisfies one or two conditions. This is the reason why we selected the CRUST 1.0 finally.

**Response:** This is a very important part that we have ignored in the original version. For better understanding, we have added the section "2.1 Kernel function calculation" at the beginning of the section "2 Methods". We calculated the kernel function under the Cartesian coordinate system, we provided the integral expressions and analytical expressions of the gravity kernel function and gravity gradient kernel function.

**Methods/Joint inversion (section 2.3).**

**8. Can we talk about joint inversion when the inversion is realized in two steps? The term of "sequential inversion" would be more appropriate here (see Lines et al., 1988 Cooperative inversion of geophysical data).**

**Response:** (1) We have revised the sentence as followings. (2) We have changed the term "joint inversion" to "sequential inversion", and and checked all the terms throughout the paper.

**Revision: (Page 12, Paragraph 2)**
The sequential inversion is realized in two steps, namely, the gravity inversion and the gravity gradient inversion.

**9. How the four GOCE satellite gravity gradient tensor components are inverted? Separately? Simultaneously? Which is the contribution of each component in the inversion results?**

**Response:** The detailed information can be referred to comments 2.

**10. What about the two others components not used in this study? Despite the errors on these components, they really cannot bring any useful constraints? How the kernel function is calculated in the gravity gradient inversion? One kernel function by component or one for the all of them?**

**Response:** (1) This is really very good question that we really have discussed between authors when we carried out this study. We did not use two others components, because there are errors on the observation instead of noise, the inversion algorithm has the anti-noise ability, but the gravity gradient components with errors will bring much more uncertainties on the inversion results, and we also referred some published papers (Rummel et al., 2011; Yi et al., 2013) that recommended the use of four accuracy components.

(2) We calculated all the kernel functions of four gravity gradient components, we have added the section "2.1 Kernel function calculation", and we have provided detailed information about kernel function when first mentioned it in section "2.2 Inversion method"

**Response:** We have modified the color bar, we changed fixed value bar to the gradient bar as followings in Figure 5(a).

**Revision: (Page 14)**

[Figure]

Figure 5 (a) The corrected gravity anomalies after several corrections.

**Data processing/correction for the underground interface undulation effects (section 3.5).**

**17. The resolution of the CRUST 1.0 model used for the data correction is 1°x1°. The resolution of the inversion results is 0.25° x 0.25°. I have the same question: How is possible? The resolution of inversion results has not to be higher than the resolution of the inverted data. The author should adapt the resolution of the inversion in function of the resolution of corrected data.**

**Response**:Thank again for reminding this problem. We have adapted the resolution of the inversion in function of the resolution of corrected data. And we described the information about how the the results for each calculation are homogenized the same spatial resolution of  0.5° × 0.5°. We have revised the paper as followings.

**Revision: (Page 14, Paragraph 2)**
Therefore, before calculating the corrected gravity gradient component, the resolutions of the results for each calculation are homogenized to the same spatial resolution of 0.5° × 0.5°. For the higher spatial resolution of the data (e.g., gravity gradient data after downward continuation and topographic correction), we extracted data from the calculated results. For the gravity gradient correction caused by the underground interface undulation, the common kriging interpolation method was adopted to obtain the data needed for the spatial resolution.

**18. What is the impact of these corrections (sedimentary layers and crust) on the final inversion results? The amplitude of these corrections is much greater than the residual signal used for the inversion.**

**Response**:This comments can be referred to comments 1.

**19. Data processing/correction for the long wavelength effects (section 3.6).**

**The authors say that this effect is minor. They should directly quantify and clearly state how**
740   **many it is. For the Tzz, I compute 10% of the total signal. It is not so minor. Why the authors use**
**the EGM 2008 model for this correction and not a model based on the GOCE data only or even**
**better their own data developed in spherical harmonic? I really do not understand this step of**
**treatment.**

745   **Response**:(1) The original description is not accurate and rigorous. We have revised the sentence from
"The gravity gradient effects induced by the long wavelength are relatively minor, as is shown in Fig.
12." to "The gravity gradient effects induced by the long wavelength are presented in Fig. 12."

(2) The impact of each correction applied on the data before the inversion on the resulting density
models can be referred to comment 1.

750   (3) We also considered the other GOCE models at first, but for two reasons, we chose the EGM2008
model. Firstly, the agreement between EGM2008 and the GOCE-models up to degree and order 200 is
good (Yi and Rummel, 2014), the differences exist in the higher order terms. But the long wavelength
correction is for the lower order terms in our calculation. And the relative reliability distribution of
EGM2008 is high in the Eastern of China (Zhang, 2013), where North China Craton locates. Secondly,
755   we used different models and several programs or software in the data processing, we have to ensure the
calculations are correct in every step, we compared our calculation results with existing results. The
EGM2008 model is widely used with more published results, which is more favourable for the
comparison and analysis.

**Response:** We discussed about the comparison of density distribution in words, and we presents the comparison of density amplitude in Table 4. Based on the above two parts , we figure that the difference between gravity gradient data inversion and gravity data inversion is significant.

**Revision: (Page 27, Paragraph 1)**

In comparison (Fig. 6), the inversion based on the gravity gradient provides more local and detailed information about the density anomaly distribution within the entire NCC. The maximum and minimum values of the inversion results based on the gravity gradient have a larger range, and the detailed data statistics are summarized in Table 4. The center of the anomalies is more concentrated with the regional anomaly features, which is more favorable for the discussion about the stability and destruction in different regions of the whole NCC area. In the eastern NCC, the boundary of density differences on both sides of the Tancheng-Lujiang fault belt zone is more obvious; the extreme value of low-density anomalies is continuously present in the Bohai Bay area at depths of 60 km-80 km. In the central NCC, it is easier to determine the center of the density anomaly distribution in the southern, middle and northern parts of the Taihang Orogenic belt at 42 km-80 km, as these areas have different regional block features of the density anomaly. In the western NCC, the gravity inversion results are connected overall; however, the result of gravity gradient inversion shows the southeastern trend of the Qilian block, which is more favorable for a geodynamic analysis in the western NCC.

**Table 4** Comparison of gravity and gravity gradient inversion results.

| Depth / Inversion Results | 10km $(g/cm^3)$ | 25km $(g/cm^3)$ | 42km $(g/cm^3)$ | 60km $(g/cm^3)$ | 80km $(g/cm^3)$ | 100km $(g/cm^3)$ | 140km $(g/cm^3)$ | 180km $(g/cm^3)$ |
|---|---|---|---|---|---|---|---|---|
| Maximum density of gravity inversion | 0.023 | 0.021 | 0.011 | 0.035 | 0.053 | 0.056 | 0.058 | 0.079 |

| | | | | | | | | |
|---|---|---|---|---|---|---|---|---|
| Maximum density of gradient inversion | 0.025 | 0.026 | 0.015 | 0.043 | 0.058 | 0.070 | 0.077 | 0.091 |
| Minimum density of gravity inversion | -0.033 | -0.026 | -0.022 | -0.020 | -0.024 | -0.061 | -0.066 | -0.064 |
| Minimum density of gradient inversion | -0.035 | -0.028 | -0.025 | -0.023 | -0.028 | -0.063 | -0.078 | -0.069 |

**General comments:**

795

**21. The authors should remind what is/are the fundamental question(s) in this region and in what this study brings answers or at least new constraints in each study area.**

**Response:** This comment can be referred to comments 3.

800

**22. In their interpretation, the authors should systematically and clearly relate the density anomaly to the geological structure or to the geodynamic process. For example, they should explain why a fault which penetrates in the lithosphere produces a density anomaly. The reason is not necessarily obvious for the reader.**

805

**Response:** (1) We made two responses based on this comment. We introduced common geological structure about the Tancheng-Lujiang fault belt when we mentioned this area firstly in the paper. (2) We then explained why a fault which penetrates in the lithosphere produces a density anomaly as followings.

810

**Revision: (Page 30-31)**

1. The Tancheng-Lujiang fault belt is one of the major north-northeast fault zones in the East Asia continent. This fault belt extends more than 2400 km in China and cuts through different structural units (Huang et al., 2011). The geological structure of the Tancheng-Lujiang fault belt is complicated.

815 2. On both sides of the Tancheng-Lujiang fault belt, there are no connected large-scale density anomalies, and the density anomalies are distributed along the Tancheng-Lujiang fault belt with

different features. This distribution represents the density differences between different tectonic units. Therefore, the Tancheng-Lujiang fault belt is obvious as the boundary of the tectonic units. In addition, based on the study of seismic receiver functions (Chen et al., 2006; Li et al., 2011), the Tancheng-Lujiang fault belt zone and its extension are the most obvious areas of lithospheric thinning in the eastern NCC, and the thickness of the lithosphere is only 60-70 km. The significantly different density distributions on both sides of the Tancheng-Lujiang fault belt indicate that the fault belt may have penetrated the lithosphere.

**Eastern NCC.**

**23. Paragraph 1. "Obvious features […] obvious spatial distributions". It is not so obvious. More explanations are necessary here.**

**Response:** We have discussed the spatial distribution features with detailed topography and annotations as followings.

**Revision: (Page 30)**

The eastern NCC is characterized by obvious features with connections to Bohai Bay. In the eastern NCC, the distribution of positive and negative densities is always alternately characterized within depths 0-180 km. From the Tancheng-Lujiang fault belt (E2 and E3) in the south, to Bohai Bay (E5) and North China Basin (E1) in the middle, and the Songliao Basin (E6) in the north, the density always has the features of a regional distribution instead of large-scale distribution in the central NCC. The obvious spatial distributions of the density anomalies are consistent with the theory that this area has experienced strong deformation (Tian and Zhao, 2011).

**24. Paragraph 2. "is consistent with" the authors should explain in what this result is consistent with the earthquake belt.**

**Revision: (Page 30)**

At the depth interval of 10-180 km in the eastern NCC, there are positive density anomalies in the E1 area of the North China Basin, which are mainly distributed along the Tangshan-Xingtai earthquake belt. This result is consistent with the positive P-wave velocity along the earthquake belt (Huang and Zhao, 2004; 2009), which implies a positive density distribution affected by the Tangshan-Xingtai earthquake belt.

**25. Paragraph 3. "the fault belt may have penetrated the lithosphere". The authors should explain why they can make this interpretation. Which is the relation between the fault and the density distribution?**

**Response:** Please referred to general comments 22.

**26. Paragraph 4. "There are no significant negative density anomalies". I think that the authors mean "There are significant negative density anomalies". For this paragraph, the authors should also relate their interpretation to the density distribution described and explain this relation.**

**Response:** (1) Good reminder. We have revised the sentence as followings.

(2) We have added the introduction and analysis about Tancheng-Lujiang fault belt cutting effect. As the extension of the Tancheng-Lujiang fault belt, we have related and explained the interpretation to the density distribution in Bohai Bay as followings.

**Revision: (Page 31 Paragraph 2)**

(1) In the E5 area of the Bohai Bay, there are significant negative density anomalies at depths of 60-80 km.

(2) According to previous studies (Teng et al., 1997; Su et al., 2009), these density anomalies are ascribed to the extension of the Tancheng-Lujiang fault belt and the development of the mantle plume in the deep Bohai Bay area. However, the distribution of density anomalies in this region does not conform to the features of the mantle plume with continuous negative density residuals in the mantle.

Based on this study, these density anomalies are mainly ascribed to the extension of the Tancheng-Lujiang fault belt. Negative density residuals (E4) continuously exist at the extension of the Tancheng-Lujiang fault belt (100-180 km), which implies that the region at the extension of the fault zone has penetrated the crust to the mantle. The cutting effect of the Tancheng-Lujiang fault belt and lithospheric thinning result in asthenospheric upwelling with negative density residuals in Bohai Bay.

**Central NCC.**

**27. Paragraph 1. "N1 is connected to the high-density anomaly area…'I do not see this connection. The author should better explain. For now, the description is too fuzzy. "N3 is connected to the high density" In the same way, I do not see that. In this depths range, the sign of the density anomaly changes.**

**Response:** For better understanding, we have added the annotations when we described the detailed density distribution. We also have checked all the annotations that we mentioned in Figure 15.

**Revision: (Page 32 Paragraph 1)**
The Taihang Orogenic belt is generally northeast-southwest oriented, and this belt can be divided into three blocks, namely, southern block N1, middle block N2, and northern block N3. From 42 km to 100 km, N1 is connected to the high-density anomaly area of the western NCC (W4), while N3 is connected to the high-density anomaly area of the eastern NCC (E1) and the low-density anomaly area in the Yinshan-Yanshan blocks (N5). Blocks N3 and N5 are connected, forming low-density anomalies in large-scale areas.

**28. Paragraph 2. "alternating high and low density". What the reader must understand from this description? What does it mean in term of geological structures?**

**Response:**

(1) For better understanding, we have revised this paragraph with specific depth as followings.

(2) We reorganized the paragraph, the detailed description and analysis in terms of the geological structures can be referred to comment 30, we discussed the area N4 and N5 altogether.

**Revision: (Page 32 Paragraph 2)**

N4 in the northern part of the Linfen-Weihe Graben block is mainly distributed in the Datong volcanic area. As the depth increases, the density anomalies become significant high-density residuals at 25 km (Fig. 15b) and then become obvious low-density residuals from 42 km to 140 km. Although the Datong volcano is no longer active (Tian et al., 2009), its surrounding area still exhibits low-density residuals.

**29. I do not know where is N4. N4 is not located on the figure 15.**

**Response:** We have added N4 in figure 15, and we also have checked all the annotations in figure 15 when we mentioned relative annotations in the paper.

**30. Paragraph 3. The amplitude of the density anomaly N5 is compatible with the thermal data in the region? Please quantify.**

**Response:** (1) We really made major revisions based on this comment and general comment 3. Based on the collected temperature data, we made the temperature difference Figure 15(i-j) in order to illustrate the temperature level in different areas. (2) Based on the density inversion results and temperature differences at mantle, we figured that the central NCC is a typical area. We extended the discussion, especially the comparison between northern part and central-southern part of Taihang Orogenic belt. These discussions also in concert with the final discussion in section "5.4 The destruction mechanism of NCC".

**Revision: (Page 32-33)**

[Figure]

**Figure 15** Temperature differences at (i) 140 km and (j) 180 km. E represents the eastern NCC, N represents the central NCC, and W represents the western NCC; HFS is the Haiyuan fault system, RMF is the Riyue mountain fault, and SP is the Songliao plain.

From the distribution of the temperature difference (An and Shi, 2007; Yang et al. 2013) in Fig. 15 i-j, N5 is located in an area with a continuous high heat environment. Based on previous studies using the magnetotelluric imaging method (Zhang et al., 2016), melting occurs in the mantle of the Datong volcanic area and north of the Taihang Orogenic belt. According to the seismic receiver function, the Poisson's ratio in the northern Taihang Orogenic belt is as high as 0.3, while in the southern Taihang Orogenic belt, the value is approximately 0.25-0.26 (Ge et al., 2011). The Poisson's ratio of the continent is generally between 0.25-0.27, and although the temperature and material composition seem to have a dominant influence on Poisson's ratio (Zandt and Ammon, 1995), it is difficult to increase the Poisson's ratio to 0.3 only by changing the material composition. Therefore, the obvious negative density residuals in this area are mainly affected by the high heat environment. The upwelling of the thermal materials from the deep asthenosphere formed the magma migration pathway, which apparently transforms the lithosphere and upper mantle.

In contrast, with the northern region, negative density residuals are observed in the central and southern parts of the Taihang Orogenic belt from 60 km to 180 km, with an extreme value of -0.048 g/cm$^3$ at a depth of 180 km. However, at the depth of the mantle, the central southern part does not show features of a high continuous heat environment (Fig. 15i-j). At a depth of 140 km, the temperature in the southern block of the Taihang Orogenic (N1) is lower than the average temperature, and the

temperature is higher than the average temperature only in the middle block of the Taihang Orogenic (N2). As the depth increases, the temperatures in the southern and middle blocks of the Taihang Orogenic (N1 and N2 areas) are all lower than the average temperature at a depth of 180 km. The Poisson's ratio of this area accords with the typical continental features (Ge et al., 2011). Thermal erosion is always accompanied by a high heat flow environment, which is not consistent with this feature in the central and southern regions of the Taihang Orogenic belt. Therefore, it is inferred that the impact of temperature is limited, and the obvious negative density anomalies may be caused by delamination.

**31. Again, I do not know where is N6. N6 is not located on the figure 15. What it the relation between this interpretation and the concerned density anomaly? The mantle is it normal or hotter in this area?**

**Response:** (1) We have added N6 in figure 15.

(2) We have added detailed relation between the concerned density anomaly and the geological interpretation.

**Revision: (Page 33 Paragraph 2)**

N6 in the central NCC exhibits significant positive density residuals at depths of 140-180 km, which are in contrast to the low-density residuals over a large area of the central NCC. The mantle part of N6 is connected to the positive density anomaly area in the W1 area, Ordos block, and western NCC (Fig. 15e-f). In Fig. 15i-j, the temperature in N6 is normal at depths of 140-180 km, and the temperature boundary exists between the N6 and W1 areas. Based on previous studies (Ai et al., 2019), the Taihang Orogenic belt in the mantle part experienced a blocking effect from the rigid Ordos block during the expansion of the orogenic belt. The stable Ordos block area presents continuous large-scale high-density residuals. It is inferred that the Ordos block's blocking effect creates the positive density residuals in the central NCC (N6) and connects with the Ordos block.

Western NCC.

**32. Paragraph 2. "the low-density anomaly area at depths of 60-100 km is connected to the negative density anomaly". Are they really connected or are they two juxtaposed independent anomalies? What is the consequence on the interpretation?**

**Response:** As the low density anomalies in Ordos block (W1) are connected with low density anomalies at different sides, at a depth of 80 km, the low density anomaly of Ordos block (W1) is connected at east side, however, at a depth of 100km, the low density anomaly of Ordos block (W1) is connected at west side. And the locations of the density anomalies are different. Therefore, it is inferred the density anomalies in Ordos block (W1) are affected by central NCC (N2) and Qilian block (W3), the density anomalies distribution is connected instead of independent.

**Revision: (Page 33 Paragraph 3)**

Although Ordos is a relatively stable block with high-density features, a low-density block invasion appears at depths of 80-100 km. At a depth of 80 km, the low-density anomaly of the Ordos block (W1) on the east side is connected with the low-density anomaly in the central NCC (N2), which indicates that Ordos is affected to some extent by the destruction of the central NCC. At a depth of 100 km, the low-density anomaly of the Ordos block (W1) is connected to the Qilian block (W3) and Qaidam block (W5), and the distributions of the density anomalies are consistent with the theories that the Ordos block is affected by the northeast compression of the Qinghai-Tibet Plateau and substantial deep expansion (Sheng et al., 2015).

**33. Paragraph 4 (end). The amplitude of the low-density anomaly is consistent with the thermal data in the region? Please quantify**

**Response:** We have added the detailed figures, which can be referred to comment 30. We also illustrated the description with the geodynamics and geological background as followings.

**Revision: (Page 34 Paragraph 2)**

As the depth increases in the range of 140-180 km, persistent obvious negative density residuals are increasingly dominant in the Qilian block (W2 areas) of the western NCC. According to a previous study (Teng et al., 2010), the lithosphere in the western NCC is the thickest, with an average depth of 140-150 km. However, within this depth range, obvious low-density residuals are present in the orogenic belts of the western NCC, which eliminates the possibility that lithospheric thinning is the main cause of the density anomalies at depths of 140-180 km. According to the temperature differences (Figure 15 i-j), the Qilian block in the western NCC has a high temperature at depths of 140 km-180 km. Furthermore, according to research on the terrestrial heat flux (An and Shi, 2007), the Qilian block has an average heat flux value of up to 68.340 mW/m². Therefore, it can be concluded that the high heat flux environments lead to negative density residuals in the Qilian block in the western NCC, accompanied by an upwelling of deep asthenosphere materials and the subsequent transformation of the mantle above the lithosphere.

**Technical corrections**

34.  **Figure 15. Please add HFS, RMF and SP on the maps.**

**Response:** We have added all the annotations in Figure 15 **(Page 29)** .

**35. Figure 11b) replace sedimentary layer interface by Moho layer interface.**

**Response:** We have revised the caption as followings.

**Revision: (Page 23)**

[revised manuscript text omitted]

where $G_z^{(P_1)}, G_{xx}^{(P_1)}, G_{xz}^{(P_1)}, G_{yy}^{(P_1)}, G_{zz}^{(P_1)}$ represents the corresponding kernel function matrix at the point $P_1$, $\rho = (\rho_1, \rho_2, ......, \rho_k)^T$ represents the density of each cube. The gravity gradient anomalies generated by all the n points on the observation surface meet the linear relationship from the k discrete prisms in the underground space. The relationship can be expressed as the equation:

$$T_{ij} = G_{ij}\rho \; ,$$
$$T_{ij} = [T_{ij}^{(P_1)}, T_{ij}^{(P_2)}, ......, T_{ij}^{(Pn)}]^T \; , \tag{3}$$
$$G_{ij} = [G_{ij}^{(P_1)}, G_{ij}^{(P_2)}, ......, G_{ij}^{(P_n)}]^T \; , \; ij = z, xx, xz, yy, zz$$

[revised manuscript text omitted]
=\left[\boldsymbol{G},\sqrt{k}W_i\right]^{\mathrm{T}}$ and defining $\boldsymbol{b}=\left[\Delta\boldsymbol{d},0\right]^{\mathrm{T}}$, Eq. (15) can be simplified as:

$$A\Delta\boldsymbol{m}=\boldsymbol{b},\qquad(16)$$

for the gravity data,we take the $\boldsymbol{T}_z$ as the observation,$A=\left[\ \boldsymbol{G}_z,\sqrt{\mu}W_i\ \right]^{\mathrm{T}}$ and $\boldsymbol{b}=\left[\boldsymbol{T}_z,0\right]^{\mathrm{T}}$, for the gravity gradient data, we selected four processed components $\boldsymbol{T}'_{xx}$、$\boldsymbol{T}'_{xz}$、$\boldsymbol{T}'_{yy}$ and $\boldsymbol{T}'_{zz}$ simultaneously as the observation,which implies

the Jacobian matrix $A = \left[ G_{xx}, G_{xz}, G_{yy}, G_{zz}, \sqrt{\mu}W_i \right]^T$ and $b = \left[ T'_{xx}, T'_{xz}, T'_{yy}, T'_{zz}, 0 \right]^T$. The contribution of each component and its differences from joint inversion can be referred to our previous studies (Tian et al., 2019).

[revised manuscript text omitted]
 value of inversion results based on the gravity gradient has larger range, the detailed data statistics is summarized in Table 4. The center of anomalies are more concentrated with regional anomaly features, which is more favourable for the discussion about the stability and destruction in different regions of the whole NCC area. In the eastern NCC, the boundary of density differences on both sides of Tancheng-Lujiang fault belt zone are more obvious; the extreme value of low density anomalies present continuously in Bohai Bay area at the depth of 60km-80km. In the central NCC, it is easier to determine the centre of density anomaly distribution in the southern, middle and northern part of Taihang Orogenic belt at 42km-80km, which has different regional block features of density anomaly. In the western NCC, the results of gravity inversion is connected as a whole, however, the result of gravity gradient inversion shows the Southeast trend of Qilian block, which is more favorable for the geodynamic analysis in the western NCC.

[Figure]

[revised manuscript text omitted]
 with each other forming low density anomaly in large scale area. These features may indicate the that Taihang Orogenic belt have experienced.different geological processes from south to north.

N4 in the northern part of the Linfen-Weihe Graben block is mainly distributed in the Datong volcanic area. As the depth increases, the density anomalies significantly become high density residuals at 25 km (Fig. 15b), and then becoming obvious low density residuals from 42 km to 140 km. Although the Datong volcano is no longer active (Tian et al., 2009), its surrounding area still exhibits low density residuals. Similarly, N5 at the junction of Yinshan-Yanshan and the north of the Taihang Orogenic belt, where is featured by obvious low density residuals at depths of 60-180 km. When the depth increases to 60 km (Fig. 15d), these anomalies become connected with N3 in the north of the Taihang Orogenic belt and with N4 at the junction of the Datong volcanic area, forming a large low density anomaly zone. As the depth further increases, the low density anomaly area covering N5 presents persistently negative density residuals that extend down to a depth of 180 km (Fig. 15h). Moreover, the negative density anomaly value is more significant.

From the distribution of temperature difference (An and Shi, 2007; Yang et al. 2013) in Fig 15 i-j, the N5 is located at the area with continuous high heat environment. Based on previous studies by magnetotelluric imaging method (Zhang et al., 2016), melting occurs at the mantle of Datong volcanic area and north of the Taihang Orogenic belt. According to the seismic receiver function, the Poisson's ratio in the north of Taihang Orogenic belt is as high as 0.3, while in the south of Taihang Orogenic belt is just about 0.25-0.26 (Ge et al., 2011). The Poisson's ratio of continent is generally between 0.25-0.27, although the temperature and material composition seem to have a dominant influence on Poisson's ratio (Zandt and Ammon, 1995), it is difficult to increase the Poisson's ratio to 0.3 only by the change of material composition. Therefore, the obvious negative density residuals in this area is mainly affected by the high heat environment. The upwelling of the thermal materials from deep asthenosphere formed the magma migration pathway, which transforms the lithosphere and the upper mantle apparently.

In contrast with north regions (N3 and N5), negative density residuals is observed at the central (N2) and southern part (N1) of Taihang Orogenic belt from 60km to 180km, with the extreme value -0.048g/cm$^3$ at a depth of 180km. But at the depth of mantle, the central south part does not show the features with high continuous heat environment. At a depth of 140km, the

temperature at southern block of Taihang Orogenic is lower than the average temperature, the temperature is higher than the average temperature only at middle block of Taihang Orogenic. With the depth increases, the temperature at southern and middle block of Taihang Orogenic all lower than the average temperature at the depth of 180km. And the Poisson's ratio of this area accords with the typical continental features. The thermal erosion is always accompanied by the high heat flow environment, which is not consistent with this feature in the central and southern region of Taihang Orogenic belt. Therefore, it is inferred that the impact of temperature is limited, the obvious negative density anomalies may be caused by the delamination.

N6 in the central NCC exhibits significant positive density residuals at the depth of 140-180km, which contrast the low density residuals over a large area of the central NCC. The mantle part of N6 is connected to the positive density anomaly area in the W1 area, Ordos block, western NCC (Fig. 15e-f). While in Fig. 15i-j, the temperature in N6 is normal at the depth of 140-180km, and the temperature boundary exists between N6 and W1 areas. Based on previous studies (Ai et al., 2019), Taihang Orogenic belt in the mantle part experiences the blocking effect by the rigid Ordos block during expansion of the orogenic belt. And the stable Ordos block area presents continuous large scale high density residuals. It is inferred that the Ordos block blocking effect makes the positive density residuals exist in central NCC (N6) and connect with the Ordos block.

**5.3 Western NCC**

[revised manuscript text omitted]

**5.4 The destruction mechanism of NCC**

The low density anomalies in the Qilian Block and northern Taihang Orogenic belt are affected by the high heat flux environments. The low density anomalies in cental Taihang Orogenic belt exist but not accompanied by the continuous high heat flux environments at mantle depth. However, without mantle plume found, the low density anomalies in the Bohai Bay are affected by the extension of the Tancheng-Lujiang fault belt, the mechanical extension makes the lithosphere destruction

in this area. Based on our studies, one theory is hard to explain the destruction phenomena and modes in the whole NCC. The destruction of NCC is not only affected by the physical tension, but also caused by the thermal erosion and delamination.

1700 The former dynamic studies show that (Zhu et al., 2012; Zhu, 2018), since Mesozoic, the Pacific plate subducted westward to the Taihang Orogenic belt in the central part of the NCC. The residual dehydration of subducted plate in the mantle transition zone promoted the increase of the molten fluids content in the upper mantle beneath the NCC. The delamination and thermal erosion of the lithosphere in NCC just reflect different forms of mantle convection instability. Therefore, through this study, it is believed that the destruction in NCC is caused by several forms. The several destruction modes of the

1705 NCC coexist with different background of geological structures.

[revised manuscript text omitted]

---

## Author Response (AR2)

**ABSTRACT**

**Comment1 Make the sentence "The inversion results show followings" broader, something as:"This study clearly illustrates that GOCE data are helpful in understanding the geological settings and tectonic structures at regional scale, here the NCC. Indeed, we show that… "**

**Response:** We have revised the sentence as followings.

**Revision:** This study clearly illustrates that GOCE data are helpful in understanding the geological settings and tectonic structures in the NCC with regional scale. The inversion results show that in the crust, ... .

**Comment2 Avoid references in your abstract, in order to keep it self-consistent.**

**Response:** We have deleted references in abstract, we have revised the manuscript as followings.

**Revision:** In the mantle, the presented obvious negative density areas are mainly affected by the high heat flux environment.

**MAIN TEXT**

**Comment3 For some figure captions there are some extra spaces, e.g. Fig 13. (a) Txx , (b) Txz , (c) Tyy , (d) Tzz.**

**Response:** We have checked all the figure captions and deleted all the extra spaces. We revised the manuscript as followings.

**Revision:** The corrected gravity gradient components (a) $T_{xx}$ , (b) $T_{xz}$ , (c) $T_{yy}$ , and (d) $T_{zz}$ .

**Comment4 For some figures: note that different panels have different color scales. You need to indicate this (e.g. "Note the color scales are different"), and in the text to motivate this choice. It will be nicer to have on each color-scale a clear min/max (e.g. Fig 6 for 80 km, the max is exactly at 0.06, but for 100km is > 0.06 and < 0.08 (considering the 0.02 tick intervals)**

**Response:** We have indicated this in the paper, and presented the reason for this choice (Revision 1). We also annotated the min and max along the color scale as followings (Revision 2).

**Revision1:** In order to highlight the density anomaly at different depths, the color scales of figures are different in Fig. 6. For further comparison of gravity inversion results (Fig. 6) and the gravity gradient inversion results (Fig. 15), the color scales are unified in Fig. 6 and Fig. 15.

**Revision2:**

[Figure]

**Comment5** "E represents eastern NCC, N represents central NCC, W represents western NCC" – using N for center is somehow confusing (one might think at northern NCC). Why not to use "C" (center) or "M" (middle). Attention – if you agree to implement this change, it has to be done in figures as well.

**Response:** We have adopted your constructive suggestions. We have updated the figure 15 with "C" (center), and we also have updated this notation throughout the paper.

[revised manuscript text omitted]

$$\boldsymbol{R} = \left\|\partial_x \Delta \boldsymbol{m}\right\|^2 + \left\|\partial_y \Delta \boldsymbol{m}\right\|^2 + \left\|\partial_z \Delta \boldsymbol{m}\right\|^2 = \int\left(\frac{\partial \Delta \boldsymbol{m}}{\partial x}\right)^2 dv + \int\left(\frac{\partial \Delta \boldsymbol{m}}{\partial y}\right)^2 dv + \int\left(\frac{\partial \Delta \boldsymbol{m}}{\partial z}\right)^2 dv, \tag{8}$$

by meshing the model and replacing the partial differential form with the finite difference form, the above equation is converted into the following matrix:

$$\boldsymbol{R} = \Delta \boldsymbol{m}(\boldsymbol{R}_x^{\mathrm{T}} \boldsymbol{R}_x + \boldsymbol{R}_y^{\mathrm{T}} \boldsymbol{R}_y + \boldsymbol{R}_z^{\mathrm{T}} \boldsymbol{R}_z) \Delta \boldsymbol{m}, \tag{9}$$

[revised manuscript text omitted]

---

## Author Response (AR3)

**Before I responded two questions, I acknowledged for the editor and referees to help us improve the manuscript many a time, I really learned a lot from your rigorous academic attitude and high standard for paper, I really improved a lot during the whole revision process, and I figure this is the reason why we chose to submit our important work to the journal Solid Earth. This is really a memorable experience in my whole academic career.**

Yu Tian

tybgys455429145@163.com

**Comment 1 My initial concern is to quantify the impact of these corrections on the final density models, notably the impacts of the crust and sedimentary thicknesses not always very well constrained. For example, it will be interesting to know what is the impact of a few kilometer error in the CRUST Moho and sedimentary depths on the final results. This a good way to estimate the robustness (with error bars) of the final density values obtained in the study area.**

**Response:**

1. Very constructive comments. I really agreed with referee on this important point. As there are too much content in previous response, we did not respond this question in detail. In fact, we have considered this question in our previous studies (Tian and Wang, 2018), in which we only did inversion by GOCE data. We calculated anomalous gravity gradient effect component caused by the sedimentary layer uncertainty depth and Moho uncertainty depth, the statistics of the calculation can be used to test the anti-noise ability of the algorithm. The uncertainties of crust and sedimentary thicknesses have some impacts on the final corrected gravity gradient components (Rs and Rm), but the inversion algorithm also has the anti-noise ability. Based on the anomalous gravity gradient effect caused by the sedimentary layer uncertainty depth and Moho uncertainty depth, we have tested the anti-noise ability of the inversion algorithm as followings. The notable anomalous gravity gradient effect component caused by the sedimentary layer uncertainty depth and Moho uncertainty depth is Tzz (-4.51%), and we tested the anti-noise ability of the inversion results with 8% Gaussian spatially correlated noise by two sets model tests, the algorithm presents the strong anti-noise ability based on the inversion results (Figure 10).

**Reference**

Tian, Y., and Wang, Y.: Inversion of the density structure of the lithosphere in the North China Craton from GOCE satellite gravity gradient data, Earth Planets Space, 70, 173, doi:10.1186/s40623-018-0942-1, 2018.

**2.** In consideration of the inversion is an ill-conditioned problem, we did the anti-noise ability tests on the complex models instead of the study area, as we can compare the inversion results with the true model and estimate inversion results, which is more favourable to demonstrate the effectiveness of the inversion results.

**Table 2 The sedimentary layer interface correction and uncertainty analysis of the layer depth**

| Gravity Gradient Tensors | Maximum of gravity gradient tensors by sedimentary layer(E) | Minimum of gravity gradient tensors by sedimentary layer(E) | Maximum uncertainty of gravity gradient tensors by sedimentary layer(E) | Minimum uncertainty of gravity gradient tensors by sedimentary layer(E) | Mean uncertainty of gravity gradient tensors by sedimentary layer(E) | Resultant change in gravity gradient tensors ($R_s$) |
|---|---|---|---|---|---|---|
| $T_{xx}$ | 6.4258 | -1.7022 | 0.7810 | -1.2032 | 0.038 | -1.35% |
| $T_{xz}$ | 0.8060 | -0.5912 | 0.1908 | -0.2594 | -0.006 | 0.32% |
| $T_{yy}$ | 2.9595 | -1.8646 | 1.2278 | -1.4064 | 0.057 | -1.88% |
| $T_{zz}$ | 1.6786 | -7.3107 | 1.5172 | -1.5437 | -0.148 | -2.18% |

**Table 3 The Moho layer interface correction and uncertainty analysis of the layer depth**

| Gravity Gradient Tensors | Maximum of gravity gradient tensors by Moho layer(E) | Minimum of gravity gradient tensors by Moho layer(E) | Maximum uncertainty of gravity gradient tensors by Moho layer(E) | Minimum uncertainty of gravity gradient tensors by Moho layer(E) | Mean uncertainty of gravity gradient tensors by Moho layer(E) | Resultant change in gravity gradient tensors ($R_m$) |
|---|---|---|---|---|---|---|
| $T_{xx}$ | 10.099 | -4.924 | 4.862 | -2.468 | -0.036 | 1.28% |
| $T_{xz}$ | 4.514 | -8.584 | 1.9071 | -2.758 | 0.044 | -2.35% |
| $T_{yy}$ | 5.805 | -6.260 | 2.4803 | -3.250 | 0.062 | -2.04% |
| $T_{zz}$ | 8.916 | -14.587 | 3.7330 | -3.5636 | -0.158 | -2.33% |

**Anti-noise ability test of PCG algorithm**

The uncertainty of the gravity gradient effect due to the total uncertainty sedimentary and Moho interface depth are the -0.07% ( $T_{xx}$ ), -2.03% ( $T_{xz}$ ), -3.92% ( $T_{yy}$ ), -4.51% ( $T_{zz}$ ) respectively. To ensure the reliability of the inversion results, we designed two sets of complex models (M1 and M2), and the 8% Gaussian noise was added to the four gravity gradient components in the synthetic model tests. To test the feasibility of the inversion method for several anomalies, model M1 consists of four anomaly blocks with different sizes, depths and densities. To demonstrate the effectiveness of the inversion method for large blocks, we designed two large blocks with density anomalies in Model M2.

70 Compared with Figures 10b and 10e, the Figures 10c and 10f show that there are only some same anomalies appear around the anomaly, and the density at the upper layer is a little higher than the anomalies in Figures 10b and 10e, the inversion result reveals that the algorithm has robust anti-noise ability.

[Figure]

**Fig. 10** Anti-noise ability test of PCG algorithm. **a** Synthetic model M1. **b** inversion results (M1) of four components without noise. **c** inversion results (M1) of four components with 8% Gaussian noise. **d** Synthetic model M2. **e** inversion results (M2) of four components without noise. **f** inversion results (M2) of four components with 8% Gaussian noise.

**Table 4 Conditions and results for two different synthetic inversion models**

|  |  | Iteration | Maximum value of results($10^3$kg/m³) | Minimum value of results($10^3$kg/m³) | Mean value of results($10^3$kg/m³) |
|---|---|---|---|---|---|
|  | Observation ($T_k$) |  |  |  |  |
| Model 1 | $T_1$ | 5 | 1.0649 | 0.0831 | 0.489 |
|  | $T_1$+8%noise | 6 | 1.1134 | 0.0956 | 0.512 |
| Model 2 | $T_2$ | 6 | 0.8527 | 0.0654 | 0.445 |
|  | $T_2$+8%noise | 8 | 0.9033 | 0.0872 | 0.478 |

$T_1$ and $T_2$ represent the synthetic gravity gradient data $T_k$ for M1 and M2, respectively

**Revision:** (Page 21) The gravity gradient effect caused by the sedimentary depth uncertainty and Moho depth uncertainty has been considered in our previous studies(Tian and Wang, 2018), and the anti-noise ability of the inversion results is tested based on the underground layer depth uncertainty, the PCG algorithm has the strong robustness in anti-noise ability to ensure the reliability of the inversion results.

**Comment 2 You extended the geographic area by 10° for the corrections computation. Why 10°?**

**Response:**

Based on the recommended paper (Szwillus et al., 2016) by referee 1, the recommended paper points out that " For satellite-based gravity gradients, to reduce the remaining rms distant effect to 10% of the global rms, a radius of 10° is sufficient. And for the regional test, the RMS is much more smaller than the global average, only one third of the global rms." Therefore , we confirm that 10° extension for far zone distant gravity gradient effects is enough for our study area. For better understanding, we have revised the manuscript as followings.

[revised manuscript text omitted]

(2)

where $G_z^{(P_1)}, G_{xx}^{(P_1)}, G_{xz}^{(P_1)}, G_{yy}^{(P_1)}, G_{zz}^{(P_1)}$ represents the corresponding kernel function matrix at point $P_1$ and $\rho = (\rho_1, \rho_2, \ldots, \rho_k)^T$ represents the density of each cube. The gravity gradient anomalies generated by all n points on the observation surface meet the linear relationship from the k discrete prisms in the underground space. The relationship can be

255 expressed as the following equation:

$$T_{ij} = G_{ij}\boldsymbol{\rho} \; ,$$

$$T_{ij} = [T_{ij}^{(P_1)}, T_{ij}^{(P_2)}, \cdots\cdots, T_{ij}^{(Pn)}]^T \; ,$$

$$G_{ij} = [G_{ij}^{(P_1)}, G_{ij}^{(P_2)}, \cdots\cdots, G_{ij}^{(P_n)}]^T \; , \;\; ij = z, \; xx, \; xz, \; yy, \; zz \tag{3}$$

where $T_{ij}$ represents the observation and $G_{ij}$ represents the corresponding kernel function matrix for the observation surface.

**2.2 Inversion method**

Essentially, the inversion of the gravity and gravity gradient tensors is a process of solving a system of linear equations. The quantity of the unknown m greatly exceeds the acquired data vector, and the solutions of the equations are nonunique. Moreover, inversion is an ill-conditioned problem, and appropriate constraints on the objective function are required to narrow the range of solutions. Therefore, in the linear inversion theory, the objective function mostly consists of the data fitting function and the model objective function (Constable et al., 1987). Under such circumstances, solving the inversion problem is equivalent to finding a model vector $\boldsymbol{m}$ that can minimize the objective function while satisfying the data fitting function condition.

The objective function can be expressed as follows:

$$\text{minimize}: \phi = \phi_d + \mu\phi_m \; , \tag{4}$$

where $\mu$ represents the regularization parameter, which represents the weight factor that balances the data fitting function $\phi_d$ and the model objective function $\phi_m$.

The data fitting function is defined as follows:

$$\phi_d = \sum_{i=1}^{N} \left( \frac{\Delta d_i - G_i \Delta m}{\sigma_i} \right)^2 = \left\| W_d (\Delta d - G\Delta m) \right\|^2 \; , \tag{5}$$

$$W_d = diag\left\{ 1/\sigma_1, 1/\sigma_2, ..., 1/\sigma_N \right\} \; , \tag{6}$$

where $\Delta\boldsymbol{m} = \boldsymbol{m} - \boldsymbol{m}_0$ is the correction between the model parameter vector $\boldsymbol{m}$ and the initial model $\boldsymbol{m}_0$; $\boldsymbol{G}$ is the kernel function, namely, the linear projection operator from the model element to the observation (for gravity data, $\boldsymbol{G} = \boldsymbol{G}_z$, while for gravity gradient components, $\boldsymbol{G} = [\boldsymbol{G}_{xx}, \boldsymbol{G}_{xz}, \boldsymbol{G}_{yy}, \boldsymbol{G}_{zz}]^T$); $\Delta\boldsymbol{d}$ is the correction of the corresponding measurement; and $\boldsymbol{W}_d$ is a diagonal matrix, with $\sigma_i$ representing the standard deviation of the $i$-th data. The objective function of the model is constructed according to the minimization model function.

The Lagrangian multiplier is used as the regularization parameter in the PCG inversion algorithm. In the process of solving a large-scale matrix, the calculation of the Lagrangian multiplier is complex; therefore, an empirical value is often adopted as the relative optimal value for the regularization parameter. Since the regularization parameter serves to balance the data fitting function and the model fitting function, an excessively large value will result in substantial differences between the inverse results response and the observation, while an overwhelmingly small value leads to ineffectiveness of the model fitting function.

Given these problems, the L-curve method was developed for the selection of regularization parameters in the solution for ill-posed problems (Hansen, 1992). The L-curve is a criterion that is based on a comparison between the actual data fitting function and the model objective function, which is applicable to solving large-scale problems. The value corresponding to the inflection point of the L-curve is assigned to the regularization parameter. The effectiveness of this method has been validated in previous studies (Tian et al., 2018; 2019). The curvature of the L-curve can be expressed as follows (Hansen, 1992):

$$k = \frac{\hat{\rho}'\hat{\eta}'' - \hat{\rho}''\hat{\eta}'}{\left[ (\hat{\rho}')^2 + (\hat{\eta}')^2 \right]^{3/2}}, \tag{7}$$

where $\hat{\rho} = \log(\phi_d)$ and $\hat{\eta} = \log(\phi_m)$; and the superscripts ' and " represent the first-order and second-order derivatives of the function, respectively. Accordingly, during the inversion process, the algorithm seeks the maximum curvature of the L-curve based on the function constructed by actual data.

To constrain the spatial structure of the model and achieve a continuous variation in the inversion image along the three axis directions, a roughness matrix is introduced into the model objective function (Constable et al., 1987), with reference to the minimization model function. The three-dimensional model vector $\boldsymbol{R}$ is the quadratic sum of the first-order partial difference of the model vector $\boldsymbol{m}$ along the x, y and z directions.

$$\boldsymbol{R} = \left\| \partial_x \Delta \boldsymbol{m} \right\|^2 + \left\| \partial_y \Delta \boldsymbol{m} \right\|^2 + \left\| \partial_z \Delta \boldsymbol{m} \right\|^2 = \int \left( \frac{\partial \Delta \boldsymbol{m}}{\partial x} \right)^2 dv + \int \left( \frac{\partial \Delta \boldsymbol{m}}{\partial y} \right)^2 dv + \int \left( \frac{\partial \Delta \boldsymbol{m}}{\partial z} \right)^2 dv, \tag{8}$$

by meshing the model and replacing the partial differential form with the finite difference form, the above equation is converted into the following matrix:

$$\boldsymbol{R} = \Delta \boldsymbol{m} (\boldsymbol{R}_x^{\mathrm{T}} \boldsymbol{R}_x + \boldsymbol{R}_y^{\mathrm{T}} \boldsymbol{R}_y + \boldsymbol{R}_z^{\mathrm{T}} \boldsymbol{R}_z) \Delta \boldsymbol{m}, \tag{9}$$

where $\boldsymbol{R}_x$, $\boldsymbol{R}_y$ and $\boldsymbol{R}_z$ are the roughness matrices of the model along the x, y and z directions, respectively.

Because the gravity data and gravity gradient data have no fixed depth resolution, the kernel function declines rapidly with increasing depth, and the inversion results are limited near the surface, which results in difficulties in capturing the true position of the anomaly. By introducing the depth weighting function into the model objective function, the kernel function is optimized to reflect the true weighted value of the anomaly element at each depth. The depth weighting function designed by Li and Oldenburg (1996), especially for the inversion of gravity data and gravity gradient data, is adopted as follows:

$$W(z) = \frac{1}{(Z + Z_0)^{\beta/2}},$$
(10)

where $Z$ is the burial depth of the center of the grid cell and $Z_0$ and $\beta$ are constants. For gravity data, these values are used to counteract the decline in the kernel function $G$, with $\beta$ often set to 2 and the function written as $(Z+Z_0)^{-1}$, while for the gravity gradient data, these values are used to compensate for the decline in the kernel function $G$, with $\beta$ often set to 3 and the function written as $(Z+Z_0)^{-3/2}$.

In accordance with the minimization model function, the model objective function can be constructed as shown below, in reference to the roughness matrix and depth weighting function:

$$\phi_m(\boldsymbol{m}) = \alpha_s \int_V (\partial W(z) \Delta \boldsymbol{m})^2 dv + \alpha_x \int_V \left( \frac{\partial W(z) \Delta \boldsymbol{m}}{\partial x} \right)^2 dv + \alpha_y \int_V \left( \frac{\partial W(z) \Delta \boldsymbol{m}}{\partial y} \right)^2 dv + \alpha_z \int_V \left( \frac{\partial W(z) \Delta \boldsymbol{m}}{\partial z} \right)^2 dv,$$
(11)

The model objective function can be converted into a matrix form by replacing the differential form with the finite difference method:

$$\phi_m(\boldsymbol{m}) = \Delta \boldsymbol{m}^T (W_S^T W_S + W_x^T W_x + W_y^T W_y + W_z^T W_z) \Delta \boldsymbol{m} = \Delta \boldsymbol{m}^T W_i^T W_i \Delta \boldsymbol{m},$$
(12)

$$W_i = \alpha_i R_i D, \quad i = s, x, y, z,$$
(13)

where $\alpha_i$ is the weight coefficient for each term in the objective function; $R_i$ is the difference operator for each component; and $D$ is the discretized depth weighting function matrix. Substituting the model objective function into the objective function yields the following expression:

$$\phi = (\Delta \boldsymbol{d} - G \Delta \boldsymbol{m})^T (\Delta \boldsymbol{d} - G \Delta \boldsymbol{m}) + \mu \Delta \boldsymbol{m}^T W_i^T W_i \Delta \boldsymbol{m},$$
(14)

Eq. (11) can be rearranged into the following matrix form:

$$\begin{bmatrix} G \\ \sqrt{k} W_i \end{bmatrix} \Delta \boldsymbol{m} = \begin{bmatrix} \Delta \boldsymbol{d} \\ 0 \end{bmatrix}$$
(15)

by replacing the condition matrix of the objective function with $A = \left[ G, \sqrt{k} W_i \right]^T$ and defining $\boldsymbol{b} = \left[ \Delta \boldsymbol{d}, 0 \right]^
[revised manuscript text omitted]